# Human assembloids recapitulate periportal liver tissue in vitro

Lei Yuan[1,8], Sagarika Dawka[1,8], Yohan Kim[1,2,8], Anke Liebert[1,8], Fabian Rost[1], Robert Arnes-Benito[1], Franziska Baenke[3,4], Christina Götz[5], David Long Hin Tsang[1], Andrea Schuhmann[1], Anna Shevchenko[1], Roberta Rezende de Castro[1], Seunghee Kim[6], Aleksandra Sljukic[1], Anna M. Dowbaj[1], Andrej Shevchenko[1], Daniel Seehofer[5], Dongho Choi[6], Georg Damm[5], Daniel E. Stange[3,4] & Meritxell Huch[1,7 ✉]

The development of complex multicellular human in vitro systems holds great promise for modelling disease and advancing drug discovery and tissue engineering[1]. In the liver, despite the identification of key signalling pathways involved in hepatic regeneration[2,3], in vitro expansion of human hepatocytes directly from fresh patient tissue has not yet been achieved, limiting the possibility of modelling liver composite structures in vitro. Here we first developed human hepatocyte organoids (h-HepOrgs) from 28 different patients. Patient-derived hepatocyte organoids sustained long-term expansion of hepatocytes in vitro and maintained patient-specific gene expression and bile canaliculus features and function of the in vivo tissue. After transplantation, expanded h-HepOrgs rescued the phenotype of a mouse model of liver disease. By combining h-HepOrgs with portal mesenchyme and our previously published cholangiocyte organoids[4–6], we generated patient-specific periportal liver assembloids that retain the histological arrangement, gene expression and cell interactions of periportal liver tissue, with cholangiocytes and mesenchyme embedded in the hepatocyte parenchyma. We leveraged this platform to model aspects of biliary fibrosis. Our human periportal liver assembloid system represents a novel in vitro platform to investigate human liver pathophysiology, accelerate drug development, enable early diagnosis and advance personalized medicine.

Each year, chronic and end-stage liver diseases account for over 2 million human deaths worldwide[7]. Rodent models have advanced understanding of liver biology. However, species-specific differences (for example, in metabolism and toxicity) impact understanding of which concepts are universal and which are species-specific, making the translation of potential therapeutic targets into effective human therapies a substantial challenge. Human liver single-cell and spatial transcriptomics have unveiled human cellular heterogeneity[8–13]. However, the static nature of these analyses does not provide information about the highly dynamic processes occurring in disease initiation and progression. Primary hepatocytes cannot be expanded in culture[14], and, although cancer cell lines have been informative, they suffer from genetic drift. Reprogrammed hepatocytes (ProliHHs) are proliferative but have bi-phenotypic and progenitor features[15]. Additionally, none of these models recapitulates the three-dimensional (3D) bile canaliculus structures (thin and elongated lumina) observed in tissue[16], making it difficult to model complex disease states or recapitulate patient-specific traits, both of which are essential for precision medicine and early diagnosis.

Organoids have emerged as promising models to better predict therapeutic outcomes[1]. Human intestinal organoids effectively model the structure and function of human tissue[17]. However, recapitulating in vitro the architecture and cellular interactions of complex tissues such as the human liver remains an unmet challenge. We have described liver organoid models[4–6] (recently renamed as intrahepatic cholangiocyte organoids[18]) in which cholangiocytes/ductal cells can be expanded long term in culture. We[19,20] and others[21,22] have demonstrated that these models enable the study of mouse liver regeneration in vitro. Small modifications to this system allowed the generation of branching organoids[23], akin to the morphogenesis of the developing tissue[24,25], and organoids could be transplanted into animals to reconstruct the bile duct in vivo[26]. Mouse adult hepatocyte organoids have been developed[27,28]. Additionally, mouse[29] and human[27,30] liver hepatoblast organoids were successfully generated from fetal tissue. However, expanding human adult hepatocytes from patient tissue has remained a challenge[31]. Regrettably, all these models consist only of epithelial cells and lack the ability to fully replicate the cellular interactions and architecture of in vivo adult human liver tissue. Similarly, liver organoids

[1]Max Planck Institute of Molecular Cell Biology and Genetics, Dresden, Germany. [2]Department of MetaBioHealth, Sungkyunkwan University, Suwon, South Korea. [3]Department of Visceral, Thoracic and Vascular Surgery, Medical Faculty and University Hospital Carl Gustav Carus, Technische Universität Dresden, Dresden, Germany. [4]National Center for Tumor Diseases Dresden (NCT/UCC), a partnership between DKFZ, Faculty of Medicine and University Hospital Carl Gustav Carus, TUD Technische Universität Dresden and Helmholtz-Zentrum Dresden–Rossendorf (HZDR), Dresden, Germany. [5]Department of Hepatobiliary Surgery and Visceral Transplantation, Clinic for Visceral, Transplant, Thoracic and Vascular Surgery, Leipzig University Medical Center, Leipzig, Germany. [6]Department of Surgery, Hanyang University College of Medicine, Seoul, South Korea. [7]Center for Systems Biology (CSBD), Dresden, Germany. [8]These authors contributed equally: Lei Yuan, Sagarika Dawka, Yohan Kim, Anke Liebert. ✉e-mail: huch@mpi-cbg.de

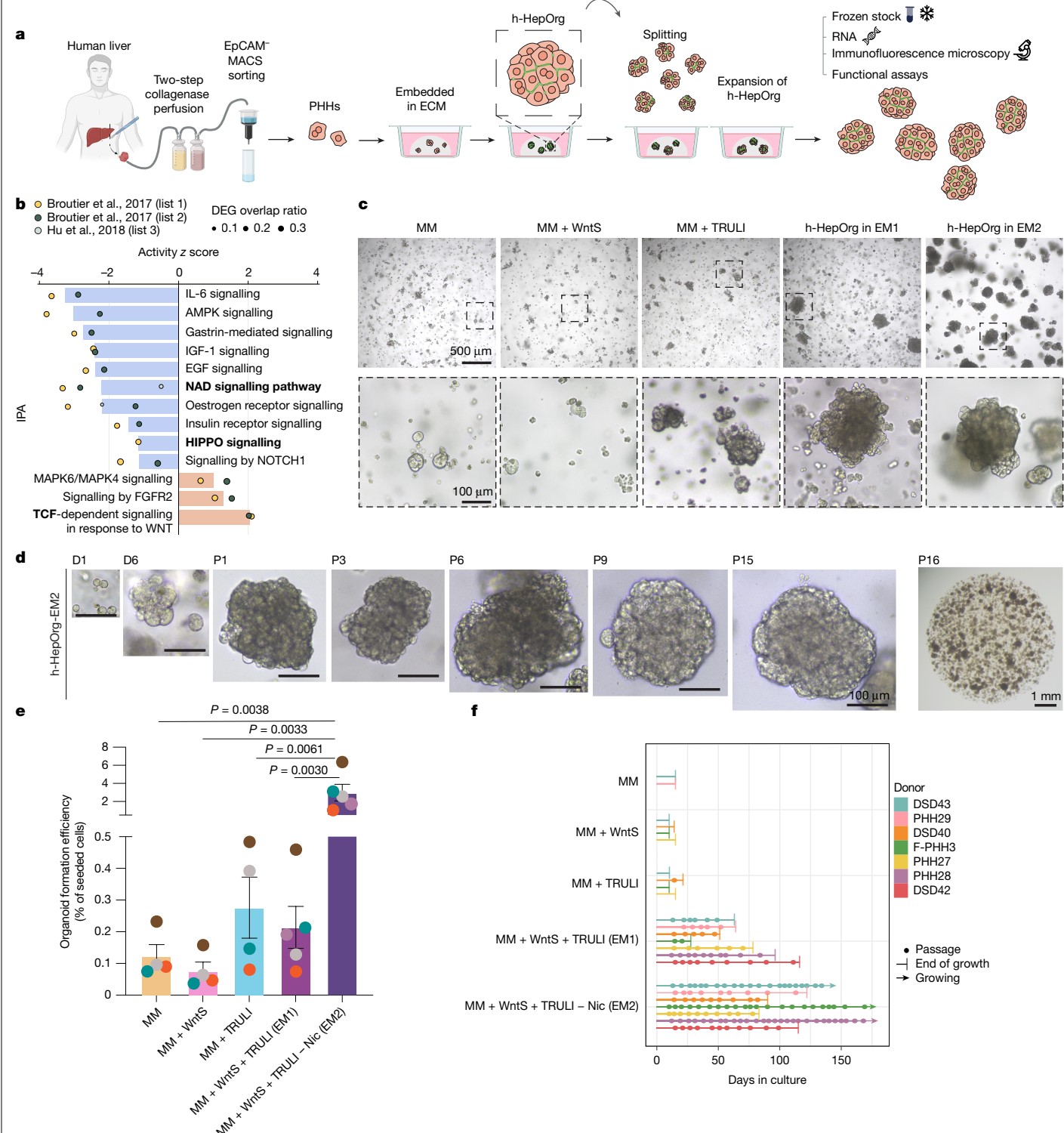

**Fig. 1** | See next page for caption.

derived from human pluripotent stem cells, although they contain stromal and epithelial populations, do not replicate native adult liver periportal cell interactions or architecture[18,32]. By co-culturing mouse cholangiocyte organoids with mouse liver portal mesenchyme, we obtained cholangiocyte–portal mesenchyme organoids that retain the binary cell–cell interactions present in the mouse liver[20]. Chimeric epithelial co-cultures of mouse cholangiocyte and two-dimensional (2D) human hepatocyte-like cells have been reported[33]. However, a complex 3D multicellular model that captures human liver portal cellular interactions does not yet exist for liver tissue from human adults.

Here we developed an adult human hepatocyte organoid model (h-HepOrgs) that allows the long-term serial expansion (passaging for >3 months at 1:2 splits) of human adult hepatocytes directly from fresh patient liver tissue. h-HepOrgs retained the gene expression and function of in vivo human adult hepatocytes in a patient-specific manner and formed bile canaliculus structures akin to the ones in human

**Fig. 1 | Primary human hepatocytes can be expanded long term when grown as hepatocyte organoids. a–f**, Liver tissues were obtained from patients undergoing surgery and processed for cell isolation as described in the Methods. Isolated PHHs were used to generate h-HepOrgs that would self-renew in vitro and could be expanded long term. **a**, Schematic depicting the protocol for generating h-HepOrgs. See Methods for details. **b**, IPA of several publicly available datasets (lists 1–3; generated from refs. 27,35) used to identify signalling pathways involved in hepatocyte proliferation. Bar plots show the IPA pathway activity *z* score for each selected pathway. Circles represent the different datasets. See Supplementary Data 1_S5 for details. Bold represents pathways related to EM1 and EM2 conditions. DEGs, differentially expressed genes. **c**, Representative bright-field images of primary h-HepOrgs cultured in the indicated media (Methods) at day 10 of culture (passage 0). Scale bars: 500 μm (top); magnification, 100 μm (bottom). **d**, Representative bright-field images of patient-derived primary h-HepOrgs serially expanded and cultured long term in EM2.

*n* = 3 independent donors. Scale bars: 100 μm (left images), 1 mm (right-most image). P, passage; D, day. **e**, Organoid formation efficiency of h-HepOrgs cultured with the indicated media. Bars represent mean ± s.e.m. from *n* = 4 (MM, MM + WntS, MM + TRULI) or *n* = 5 (MM + WntS + TRULI, MM + WntS + TRULI − Nic) independent donors. Dots are coloured by donor. One-way analysis of variance (ANOVA) followed by Tukey's multiple-comparison test. **f**, Serial expansion (with 1:2 splits) of h-HepOrgs from the indicated donors. The graph shows the expansion potential of h-HepOrgs in the indicated media. We checked the potential for organoids to be expanded beyond ten passages. As detailed in the graph and in Supplementary Table 2, under EM1 the cultures exhibited lower expansion potential, with none of them expanding beyond passage 10. For the donors for whom h-HepOrgs were expanded in EM2 and reached passage 10 (PHH29, DSD40, PHH27), we stopped culturing the h-HepOrgs at the time of submission. Panel **a** created in BioRender. Yuan, L. (2025) https://BioRender.com/hem14cv.

tissue. As we expand and cryopreserve organoids from fresh tissue, we have been able to generate a living biobank of hepatocyte organoids from 28 donors. We combined these novel patient-derived hepatocyte organoids with primary human portal mesenchyme and human cholangiocyte organoids (h-CholOrgs) from the same patient to generate human periportal assembloids that recapitulate the functional cell interactions and architecture of the in vivo tissue. Finally, we exploited the potential of this system to model aspects of human biliary fibrosis.

## YAP and WNT promote h-HepOrg growth

To recapitulate the epithelial–stromal interactions and architecture of human liver, we sought to obtain an expandable source of adult hepatocytes, cholangiocytes and mesenchyme from the same individual. A prerequisite was to first identify methods to expand human adult hepatocytes. Hence, we obtained hepatocytes from patient tissue by perfusion[34] (Methods) and cultured them in our reported hepatoblast organoid culture medium (MM)[29]. However, the cultures rapidly filled with cholangiocyte organoids, preventing further analysis (Extended Data Fig. 1a, no MACS). We adapted the isolation protocol by including a step of EpCAM magnetic activated cell sorting (MACS), which allowed us to obtain viable hepatocytes from the EpCAM-negative fraction while generating cholangiocyte organoids by culturing the EpCAM-positive cholangiocyte fraction in our reported cholangiocyte medium[4] (Fig. 1a, Extended Data Fig. 1a–c, Methods and Supplementary Table 1). Although we detected minimal growth for 7–14 days (Fig. 1c, MM), hepatocytes rapidly died thereafter.

Hence, we sought to identify culture conditions for the long-term expansion of human adult hepatocytes as hepatocyte organoids. We hypothesized that signalling pathways involved in cancer progression or tissue regeneration could activate hepatocytes and promote their exit from quiescence. To explore this, we analysed expression profiles from human liver cancer organoids[35] and mouse partial hepatectomy[27] and compared them to those for human healthy and cancer tissues (Supplementary Data 1 and Methods). Ingenuity Pathway Analysis (IPA) revealed that several pathways, such as AMPK, EGF, mTOR and IGF-1, were consistently differentially expressed across at least two datasets (Extended Data Fig. 1d and Supplementary Data 1). WNT, MAPK and FGFR2 signalling pathways were active, whereas IL-6, HIPPO and NOTCH pathways appeared inactive (Fig. 1b). Among the predicted active upstream regulators, we found YAP and CTNNB1, suggesting YAP and WNT activation (Extended Data Fig. 1e).

Both WNT and YAP are established drivers of liver regeneration[3,36,37] and cancer[38,39]. Therefore, we activated WNT and YAP signalling by supplementing our MM hepatoblast medium[29] with WNT surrogate (WntS)[40] and a LATS1/LATS2 (LATS1/2) inhibitor (TRULI or TDI-011536)[41]. Combining these enabled serial passaging (5–6 passages) of h-HepOrgs as solid structures with no lumina (Fig. 1c–f and Extended Data Fig. 1g). TRULI-treated cultures showed superior morphology compared

with TDI-011536 (Extended Data Fig. 1h), so we continued with the MM + WntS + TRULI combination, hereafter termed h-HepOrg expansion medium 1 (EM1). The other tested pathways did not result in consistent or quantifiable organoid growth (Extended Data Fig. 1f).

We further optimized the EM1 medium by testing the need for each component. Notably, removing nicotinamide improved the efficiency of organoid formation nearly tenfold and enabled long-term culture for over 3 months (>10 passages with a 1:2 split each week) (Extended Data Fig. 1i and Supplementary Table 2). These results were in line with our IPA analysis showing inactivity of NAD signalling (Fig. 1b) and previous reports of nicotinamide hepatotoxicity in humans[42]. Using these optimized conditions (EM1 without nicotinamide, or MM + WntS + TRULI − Nic, hereafter called EM2), we successfully generated expandable h-HepOrgs from 28 patients (11–85 years old, 30% female) with 100% efficiency, including 5 samples from cryopreserved hepatocytes (Supplementary Table 2). No other tested conditions supported robust expansion (Extended Data Fig. 1j, Supplementary Table 3 and source data for Extended Data Fig. 1). h-HepOrgs maintained stable chromosome numbers over time and could be frozen and thawed without loss of expansion capacity, enabling the creation of a living biobank from a total of 28 different donors (Extended Data Fig. 1k,l).

Together, these results demonstrate that combination of WNT and YAP activation allows the long-term expansion of adult h-HepOrgs.

## h-HepOrgs mimic liver structure and function

To characterize the expanded h-HepOrgs, we first performed RNA sequencing (RNA-seq) analysis on early (passages 1–3) and late (passage 10) cultures and compared their expression patterns with those of freshly isolated primary human hepatocytes (PHHs) and h-CholOrgs from the same donors (when possible). Gene expression in expanded h-HepOrgs closely correlated with that in PHHs, whereas h-CholOrgs segregated separately (Extended Data Fig. 2a,b). Gene expression and enrichment analyses revealed that, compared with PHHs, the matching h-HepOrgs exhibited a proliferative signature that was maintained until late passages (greater than passage 10) and resembled regenerating tissue after hepatectomy (Extended Data Fig. 2c–e). These results were in agreement with positive staining for the proliferation marker Ki-67 (Fig. 2a, top) and negligible staining for the apoptosis marker cleaved caspase 3 (Extended Data Fig. 2f). h-HepOrgs exhibited elevated WNT and YAP target gene expression, compared with PHHs, consistent with WNT pathway activation and LATS1/2 inhibition (Extended Data Fig. 2g,h). Immunofluorescence confirmed nuclear YAP localization in TRULI-treated cultures (Fig. 2b and Extended Data Fig. 2i). Quantitative PCR (qPCR) confirmed these results (Extended Data Fig. 2j). However, we cannot exclude the possibility that off-target effects may also contribute to h-HepOrg growth, as TRULI can inhibit kinases other than LATS1/2.

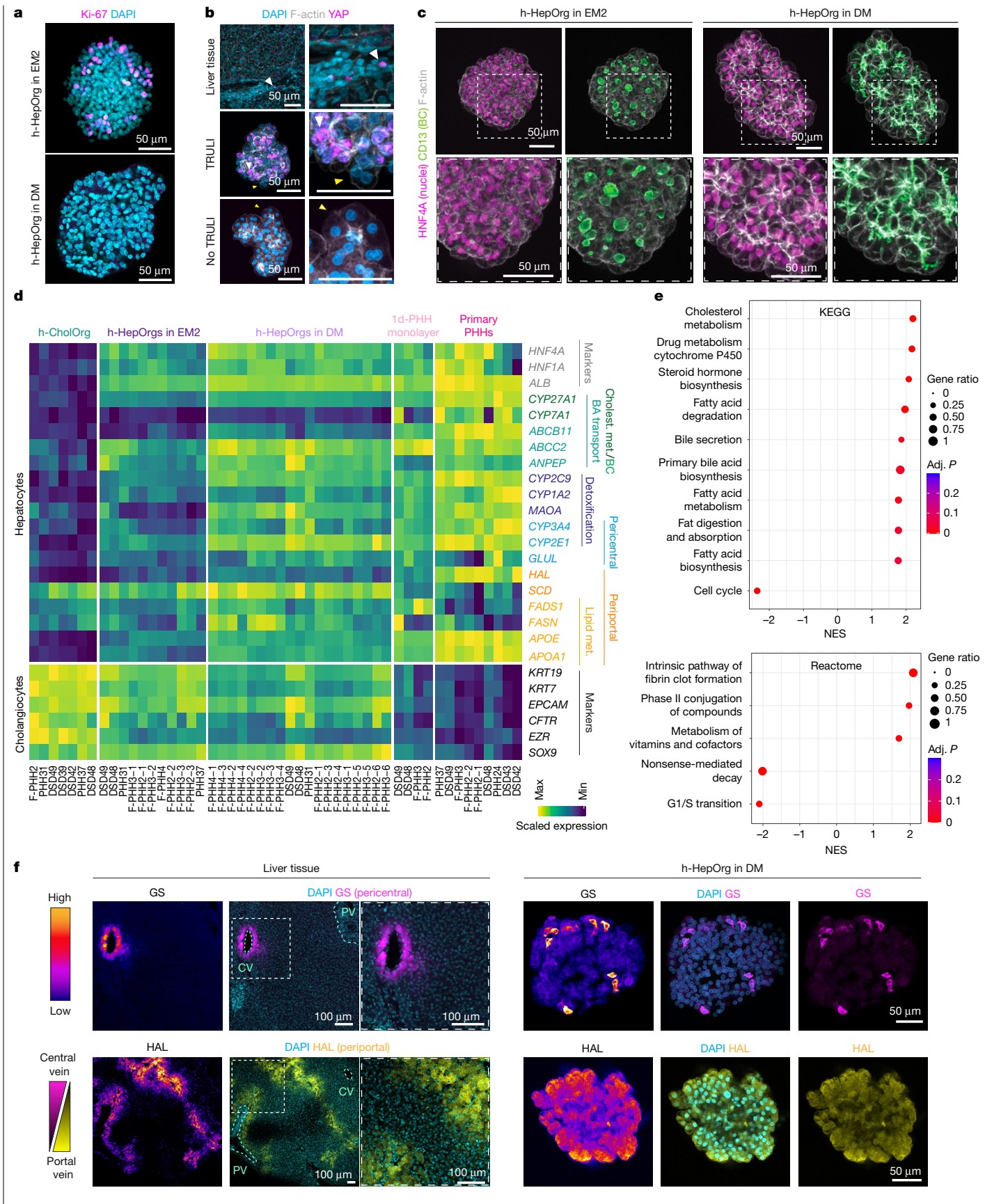

**Fig. 2** | See next page for caption.

**Fig. 2 | h-HepOrg gene expression resembles that of in vivo hepatocytes.**
**a**–**c**, h-HepOrgs expanded in EM2 or differentiated in DM or liver tissue was analysed for expression of the indicated markers. Representative images are shown from at least two independent donors from three independent experiments. **a**, Immunofluorescence staining for Ki-67 (magenta) in h-HepOrgs in EM2 (top) and DM (bottom). DAPI (cyan) stained nuclei. Scale bars, 50 μm. **b**, Immunofluorescence staining for YAP (magenta), nuclei (DAPI; cyan) and cell borders (F-actin; grey) in liver tissue (top) and h-HepOrgs in EM2 with (middle) and without (bottom) TRULI. Scale bars, 50 μm. White arrowheads indicate cells with nuclear YAP. Yellow arrowheads indicate YAP-negative cells. **c**, HNF4A (magenta), bile canaliculus (BC) marker CD13 (green) and F-actin (grey) immunofluorescence staining of organoids in EM2 (left) or DM (right). h-HepOrgs in DM exhibit thinner and more interconnected bile canaliculi. Scale bars: 50 μm. **d**,**e**, RNA-seq analysis of h-HepOrgs in EM2 or DM, h-CholOrgs and PHHs either freshly isolated (primary PHHs) or cultured for 1 day (1d-PHH monolayer). **d**, Heat map representing scaled gene expression. BA, bile acid; Cholest., cholesterol; met., metabolism. **e**, GSEA of h-HepOrgs in DM compared with EM2. Full list in Supplementary Data 2. Results are shown as a dot plot, where dot colour corresponds to the adjusted $P$ value (permutation test implemented in clusterProfiler, adjusted using the Benjamini–Hochberg method). Dot size corresponds to the gene ratio (the number of core enrichment genes divided by the total number of genes in the pathway). KEGG, Kyoto Encyclopaedia of Genes and Genomes; NES, normalized enrichment score. **f**, Human liver tissue (left) and an h-HepOrg in DM (right) stained for pericentral (GS; magenta, top) and periportal (HAL; yellow, bottom) markers. Nuclei were stained with DAPI (cyan). Fire LUT images (first column) show fluorescence intensities for GS and HAL. CV, central vein; PV, portal vein. Scale bars, 100 μm (tissue) and 50 μm (organoids). Representative images from three independent experiments.

Analysis of marker gene expression showed that the expanded h-HepOrgs expressed hepatocyte markers such as *HNF4A* and *ALB*, several apolipoproteins (*APOC2* and *APOA4*) and cytochromes (*CYP3A4* and *CYP3A7*), albeit at lower levels than in freshly isolated hepatocytes (Fig. 2d, Extended Data Fig. 2c and Supplementary Data 2). Expression of cholangiocyte markers such as *SOX9*, *KRT19* and *KRT7* was markedly reduced compared with in h-CholOrgs, whereas expression of the embryonic liver marker *AFP* suggested incomplete maturation (Fig. 2d, Extended Data Fig. 2k and Supplementary Data 2). qPCR and immunofluorescence analyses confirmed high expression of *HNF4A* and the apical and polarity marker CD13 (*ANPEP*) (Fig. 2c,d, Extended Data Fig. 2k and Supplementary Data 2). However, detailed analysis of the distribution of CD13 expression showed the presence of wide, disconnected round lumina, which does not reflect the morphology of the bile canaliculus network formed by hepatocytes in vivo[43] (Fig. 2c, compare CD13 in h-HepOrgs in EM2 to the tissue panel in Fig. 3b). Taken together, these results indicate that expanding h-HepOrgs in EM2 may represent an immature hepatocyte state. Therefore, we sought to define a differentiation medium.

LATS1/2 inhibition was recently shown to promote cholangiocyte growth[44], while it is well established that YAP activation drives hepatocyte de-differentiation and its inactivation facilitates re-differentiation[45]. Therefore, we reasoned that reducing YAP activation would facilitate the maturation of h-HepOrgs. Following several iterations, we developed a hepatocyte differentiation medium (referred to as DM) that removed YAP and FGFR2 activation, maintained WNT signalling and added dexamethasone (Extended Data Fig. 3a and Methods). Under DM, the cellular morphology improved: hepatocytes (HNF4A+) had reduced proliferation, acquired a significantly higher cytoplasm to nucleus ratio and had improved bile canaliculi (CD13+), which presented with a thinner and more elongated morphology (Fig. 2a,c, compare EM2 with DM, and Extended Data Fig. 3b). Combined, these features suggested enhanced hepatocyte maturation. To assess the extent of the maturation, we performed RNA-seq analysis. In principal-component analysis (PCA), differentiated h-HepOrgs were closer to freshly isolated hepatocytes and farther away from h-CholOrgs, when compared with h-HepOrgs in EM (Extended Data Fig. 3c). Differentiated cells had increased expression of many markers of mature cells, some to similar levels as in freshly isolated human hepatocytes, including *ALB*, several apolipoproteins (*APOE* and *APOA1*), bile acid transporters (*ABCC2* (MRP2)), and cholesterol and bile acid metabolic genes (*ABCG8* and *CYP27A1*). Additionally, several detoxifying enzymes, including *CYP2C9*, *CYP3A5*, *CYP3A4* and *MAOA*, some of which are pericentrally zonated[10,11,46,47], were also upregulated (Fig. 2d and Extended Data Fig. 3f–i). In line with this, we observed strong positive enrichment for gene signatures related to hepatocyte functions, including cholesterol, fatty acid and drug metabolism, phase II conjugation, clot formation and bile secretion, among others, whereas cell cycle and proliferation signatures were negatively enriched (Fig. 2e and Extended Data Fig. 3e). Similarly, expression of the embryonic marker *AFP* and cholangiocyte makers *KRT7* and *KRT19* was reduced (Fig. 2d and Extended Data Fig. 3i).

Notably, some pericentrally zonated genes, such as *CYP2E1* and *GLUL* (encoding glutamine synthase, GS), as well as some periportally zonated genes, such as *ALDOB* and *SCD*, were highly upregulated (Fig. 2d and Extended Data Fig. 3h). In immunofluorescence analysis for pericentral (GS) or periportal (histidine ammonia lyase, HAL) markers, some organoids had a gradient of expression, with some cells positive and others negative for the markers (Fig. 2f). Dual staining for CYP2E1 (pericentral marker) and E-cadherin (enriched in the periportal region) highlighted the heterogeneity and spatial distribution of hepatocyte function within the same h-HepOrg, at least for the genes tested (Extended Data Fig. 3j).

Upon differentiation, h-HepOrgs recapitulated the complex cell polarity of in vivo hepatocytes[14], with the tight junction and apical polarity marker ZO-1 localized to the apical surface of adjacent hepatocytes, resembling the morphology of bile canaliculi in human liver tissue (Fig. 3a). Immunofluorescence staining for the apical marker CD13 followed by image analysis and reconstruction revealed that differentiated h-HepOrgs had longer and more branched bile canaliculus networks within each organoid, when compared with the same organoids in EM, and resembled in vivo tissue (Fig. 3b). Additionally, the connectivity of the bile canaliculi network was also significantly improved, coming closer to that of tissue (Fig. 3c). Notably, we observed that different patients had fine-detailed differences in bile canaliculus morphology, with some patients having thin and homogenous bile canaliculi, some having wider and inhomogeneous bile canaliculi and others having bile canaliculi full of branchlets (Extended Data Fig. 4a–c). We found similar variation in bile canaliculi architecture across our different organoid cultures, suggesting that our model could capture the different types of bile canaliculus morphology observed in human tissue (Extended Data Fig. 4d).

Given that our h-HepOrgs are derived directly from patient tissue, we next assessed whether they retain patient-to-patient variability in culture, thus enabling patient-specific modelling of hepatocyte-related liver diseases. For this, we analysed the transcriptomes of primary hepatocytes at the time of isolation and their matching h-HepOrgs under DM to identify the specific gene signatures of each patient. We found strong correlation ($R^2 = 0.7$–$0.9$) between the organoids and the original primary hepatocytes from which they were derived (Extended Data Fig. 4e). Interestingly, many of the patient-specific genes we found expressed in organoids and their source cells had been associated with susceptibility to several liver diseases, including hepatitis virus infection (*IL1RL1* and *ERAP2*), liver cancer (*GPC3*) and cholestasis during pregnancy (*GABRP*). More notably, some genes were involved in metabolic pathways, including the glutathione-related gene *GSTM3*, the lactate dehydrogenase *LDHC* and the lipid metabolism-related genes *APOA4*, *FAR2* and *ACSM1*, among others (Fig. 3d and Supplementary Table 4). These results indicated that h-HepOrgs could preserve patient-specific signatures, with important implications for modelling human liver diseases.

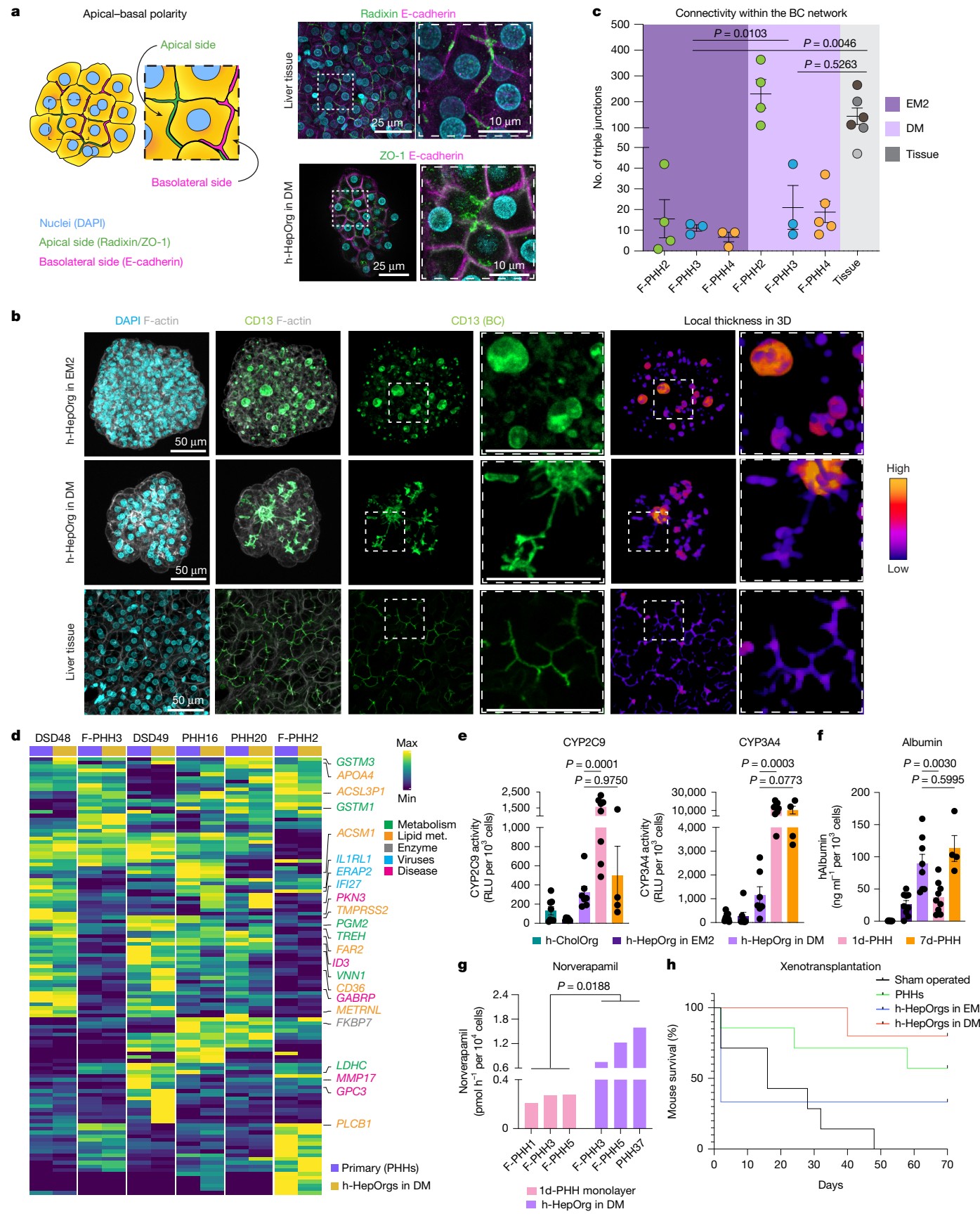

**Fig. 3 | See next page for caption.**

**Fig. 3 | h-HepOrgs maintain in vivo function and patient-specific features.**
**a**, Illustration of the apical–basal polarity of hepatocytes. Right, representative images of E-cadherin (magenta), radixin (tissue; green, top) and ZO-1 (h-HepOrg in DM; green, bottom) staining ($n = 1$ donor). Nuclei are stained with DAPI (cyan). Scale bars: 25 µm; magnification, 10 µm. **b**, Representative images ($n = 3$ donors) of CD13 (bile canaliculi; green) and F-actin (grey) in an h-HepOrg in EM2 (top), an h-HepOrg in DM (middle) and human liver tissue (bottom). DAPI marks nuclei (cyan). Right, segmented bile canaliculi in 3D, depicting local thickness in Fire LUT (blue, thinner; red, thicker). Scale bars for all images and magnifications, 50 µm. **c**, Total number of triple junctions (a proxy for connectivity). For tissue, dots correspond to one field of view. For organoids, dots correspond to one organoid and colours correspond to donors. The graph shows the mean ± s.e.m. Two-way ANOVA (factors: condition and donor) with Tukey's multiple-comparison test; $P$ values shown in the figure. **d**, Heat map showing scaled expression of donor-specific genes computed across primary PHHs (purple) and matching h-HepOrgs in DM (yellow). Hierarchical clustering was performed on both samples and genes. Full list in Supplementary Table 4. **e**,**f**, Cytochrome activity (relative luciferase units, RLU) (**e**) and albumin secretion (ng ml⁻¹) (**f**) for h-HepOrgs in EM2 (dark purple) or DM (light purple), h-CholOrgs (green) and freshly isolated PHHs cultured in 2D for 1 (light pink) or 7 (orange) days. Results are normalized by cell count. Graphs show the mean ± s.e.m. from the indicated number of independent donors: 7d-PHH ($n = 4$), h-HepOrgs in EM2 ($n = 7$ for **e** and 9 for **f**), h-HepOrgs in DM ($n = 7$ for **e** and 8 for **f**), h-CholOrgs ($n = 8$) and 1d-PHH ($n = 9$). One-way ANOVA followed by Tukey's multiple-comparison test. **g**, Norverapamil production (pmol h⁻¹ per 10⁴ cells) detected by mass spectrometry in h-HepOrgs in DM and PHHs. Unpaired two-tailed Student's $t$ test with Welch's correction ($n = 3$ donors). **h**, Kaplan–Meier survival curves for $Fah^{-/-}Rag2^{-/-}Il2rg^{-/-}$ mice sham operated or intrasplenically injected with 500,000 cells from h-HepOrgs in EM2 or DM or fresh PHHs. Log-rank test, $P = 0.0127$. Scheme from panel **a** adapted from ref. 51, Springer Nature Limited.

Next, we compared the functional performance of differentiated h-HepOrgs to that of PHHs. Differentiated h-HepOrgs exhibited hepatic functions, including robust albumin secretion and moderate cytochrome P450 activity, comparable to 7-day PHHs (Fig. 3e,f). Specifically, differentiated h-HepOrgs displayed CYP2C9 activity equivalent to that of 7-day PHHs and modestly reduced CYP3A4 activity, whereas 1-day PHHs had superior activity for both enzymes. Mass spectrometry analysis revealed that differentiated h-HepOrgs significantly outperformed 1-day PHHs in converting the antiarrhythmic and antihypertensive drug verapamil into its primary metabolite norverapamil (Fig. 3g). This suggests more robust or sustained expression and coordination among multiple CYP enzymes relevant to verapamil metabolism, including the metabolizing enzymes *CYP2C8*, *CYP3A4* and *CYP3A5*, all of which are responsible for verapamil *N*-demethylation and were highly expressed in h-HepOrgs in DM (Extended Data Fig. 3g,i). Furthermore, we observed inter-donor variability in verapamil metabolism among h-HepOrg lines (Fig. 3g), reflecting patient-specific metabolic phenotypes and underscoring the potential of this platform for personalized drug metabolism studies.

Notably, both expanded and differentiated hepatocyte organoids readily engrafted and maintained their hepatic function in vivo, following xenotransplantation in the mouse model of tyrosinemia type I liver disease ($Fah^{-/-}Rag2^{-/-}Il2rg^{-/-}$ mice)[48], with grafts distributed throughout the liver parenchyma. Importantly, the engrafted cells were able to rescue the lethal phenotype (Fig. 3h and Extended Data Fig. 4f).

In summary, we have developed a novel h-HepOrg model that enables the expansion of functional adult human hepatocytes directly from patient tissue and preserves hepatocyte polarity and bile canaliculus organization while retaining some aspects of patient-to-patient variability.

## Assembloids model periportal tissue

We next aimed to reconstruct the periportal region of the liver lobule by reproducing the cellular interactions among hepatocytes, cholangiocytes and portal mesenchyme, specifically portal fibroblasts. *PDGFRA*, which is exclusively expressed in liver mesenchyme and absent in other stromal cells[8,10,11,13,49], was used to isolate liver mesenchymal cells (Extended Data Fig. 5a–f and Methods). To enrich for portal fibroblasts, we examined publicly available datasets[8,49] and found that CD90 (*THY1*) is exclusively expressed in human portal fibroblasts (Extended Data Figs. 5g and 6a). Immunofluorescence analysis confirmed its restricted expression in the periportal region (Extended Data Fig. 6c). Thus, by sorting for CD90⁺PDGFRA⁺ cells and culturing them in defined conditions, we enriched for human portal fibroblasts (Extended Data Fig. 6b,d). RNA-seq and qPCR analysis confirmed that CD90⁺PDGFRA⁺ cells expressed portal fibroblast markers (for example, *DCN* and *THY1*), some VSMC markers (*MYH11*) and several growth factors (*HGF* and *WNT5A*, among others), all enriched in portal mesenchyme in vivo, and

were negative for markers of other liver stromal populations, such as hepatic stellate cells (for example, *LRAT*) and mesothelia (for example, *OGN*) (Extended Data Fig. 6e–h). Immunofluorescence for vimentin (mesenchyme) and CD90 (portal mesenchyme) confirmed that the majority of the expanded cells were portal fibroblasts (Extended Data Fig. 6i).

Next, we aimed to generate human periportal assembloids using cells from healthy human tissue. To facilitate visualization of integration of cells in the structures, cholangiocyte organoids and portal mesenchymal cells were tagged with nuclear fluorescent proteins (Extended Data Fig. 6j) while leaving hepatocytes unlabelled. To determine the proportions of cells to assemble into composite structures, we first quantified the physiological proportions of the three cell types in vivo, finding an average of 15% cholangiocytes, 8% portal fibroblasts and 77% hepatocytes (Extended Data Fig. 7a). Following several iterations to induce self-assembly of the three populations into a single structure, we found that mixing one h-HepOrg structure with a defined number of portal fibroblasts and cholangiocytes (from h-CholOrgs), from the same donor when possible, in 96-well low-adhesion U-bottom plates, readily generated structures in which the three cell types were together, with cholangiocytes and portal fibroblasts embedded inside the h-HepOrg structure. We called these structures periportal assembloids (Fig. 4a,b). The ratio of 1 h-HepOrg to 25 portal fibroblasts and 100 cholangiocytes better captured the tissue cell ratios at day 6 after assembloid formation and was selected for further experiments (Extended Data Fig. 7b,c). To upscale the assembloids generated, we used AggreWell plates (Fig. 4b and Extended Data Fig. 7d). Notably, both methods generated assembloids with high efficiency (approximately 80%) (Fig. 4c) and the resulting assembloids closely mirrored the cellular composition and proportions of tissue (Fig. 4e). Therefore, we only used AggreWell plates from this point on. Notably, by day 3, assembloids reproducibly established key periportal architectural features, with cholangiocytes (KRT19⁺nuclear green fluorescent protein (nGFP)⁺) forming bile duct-like structures containing open lumina in close proximity to portal fibroblasts (nuclear red fluorescent protein (nRFP)) and both cell types embedded within the hepatocyte (HNF4A⁺) parenchyma. This architectural organization, in which ductal cells form an apical lumen, basally contacted by mesenchymal cells and embedded in the hepatocyte structure, was observed in approximately 80% of the assembloids and across donors. These results were independent of the donor source of healthy liver mesenchyme, indicating minimal impact of mesenchyme origin under healthy conditions (Fig. 4d–f and Extended Data Fig. 7d–h). Portal fibroblasts consistently extended long cellular processes towards the basal side of cholangiocytes, leading to physical contacts and reminiscent of the interactions observed in human tissue, although these processes did not completely wrap cholangiocytes as in portal tracts in vivo (Fig. 4f and Extended Data Fig. 7h). Under these conditions, the assembloids could be maintained

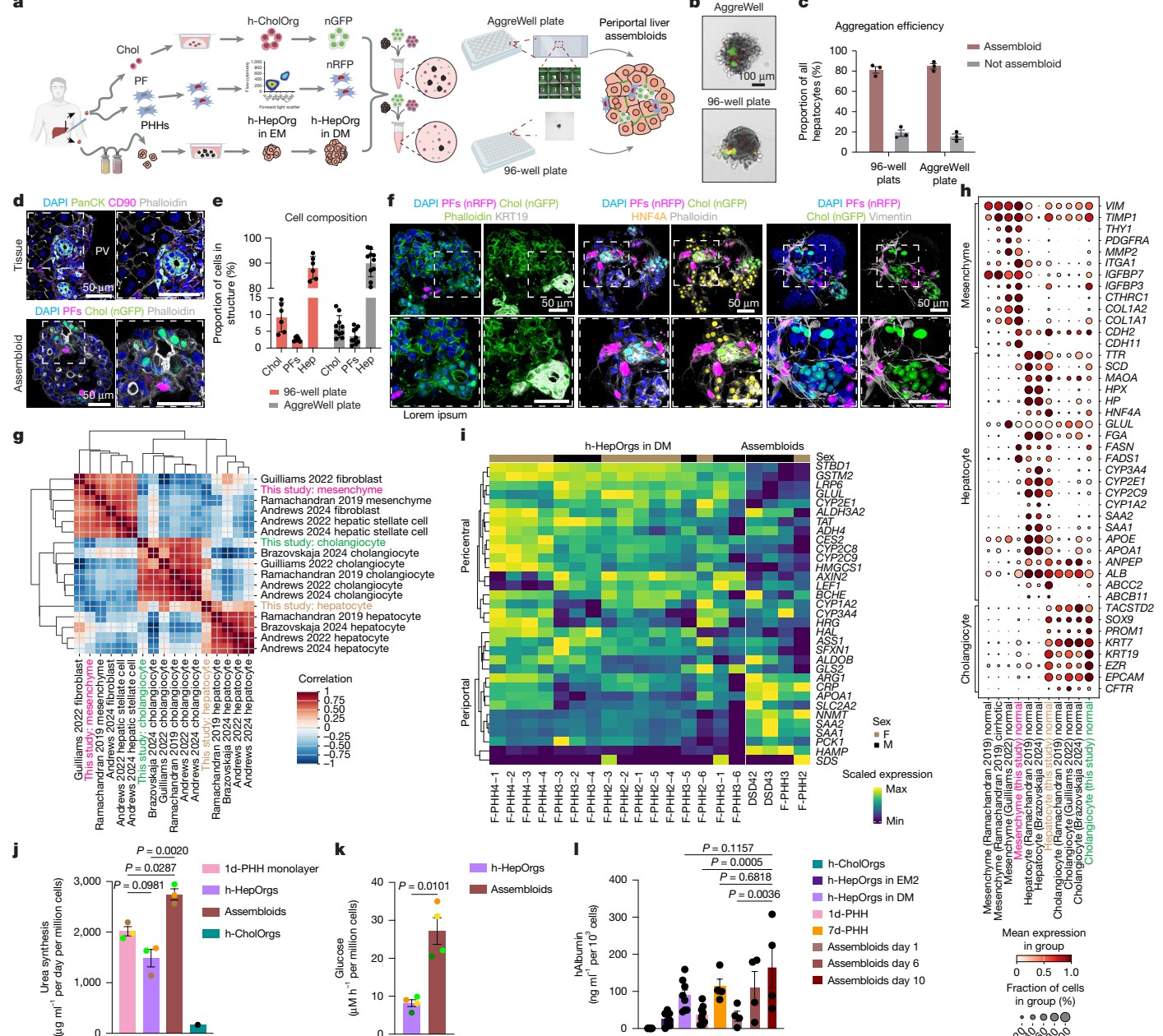

**Fig. 4 | Human periportal assembloids recapitulate in vivo liver periportal tissue. a**, Experimental approach. Chol, cholangiocytes; PFs, portal fibroblasts. **b**, Day 6 periportal assembloids. Scale bar, 100 μm. **c**, Aggregation efficiency at 24 h. Mean ± s.e.m. of *n* = 3 independent experiments. Results are presented relative to the number of HepOrgs and are expressed as a percentage. Not assembloid, structures that do not containe all 3 cell types or structures with 2 or more hepatocyte organoids per structure. **d**, Representative images of AggreWell-derived assembloids (bottom) and liver tissue (top). Portal fibroblasts (magenta), cholangiocytes (green), nuclei (blue) and membranes (grey) were visualized using the indicated markers. Scale bars, 50 μm. PanCK, pan-cytokeratin. **e**, Cellular composition at day 6. Mean ± s.e.m. from ≥3 independent experiments. Dots correspond to the percentages of hepatocytes (Hep), cholangiocytes and portal fibroblasts per structure. **f**, Representative images of day 3 assembloids showing cholangiocytes (nGFP) and portal fibroblasts (nRFP). Staining was performed for KRT19 (white, left), HNF4A (yellow, middle), vimentin (white, right), nuclei (DAPI; blue) and membranes (phalloidin; green (left) or grey (middle)). Scale bars, 50 μm. **g,h**, scRNA-seq analysis of assembloids. **g**, Correlation between assembloid cells (this study) and matching cells in liver cell atlases.

References used for comparison were refs. 8,11–13,49. **h**, Dot plot showing marker expression in assembloids and liver atlases. **i**, Heat map of liver zonation genes in h-HepOrgs in DM and hepatocytes from assembloids (pseudobulk; Methods). **j**, Urea synthesis by day 5 assembloids (brown), h-HepOrgs in DM (purple), PHHs (1-day culture; pink) and control h-CholOrgs (green). Mean ± s.e.m., *n* = 3 donors. Dot colours correspond to donors. One-way ANOVA with Tukey's test for multiple comparisons. **k**, Gluconeogenesis by day 6 assembloids (brown) and h-HepOrgs in DM (purple). Mean ± s.e.m., *n* = 4 independent donors. Dot colours correspond to donors. Two-tailed paired *t* test. **l**, Human albumin secretion by assembloids at 1, 6 or 10 days (brown). Data for h-HepOrgs reproduced from Fig. 3f and shown for comparison: data are also shown for h-CholOrgs (green), PHHs (1-day culture, light pink; 7-day culture, orange), and h-HepOrgs in DM (light purple) or EM2 (dark purple). Bars show the mean ± s.e.m. from *n* = 4 (7d-PHH, assembloids), *n* = 8 (h-CholOrgs, h-HepOrgs in DM) or *n* = 9 (h-HepOrgs in EM2, 1d-PHH) donors. One-way ANOVA followed by Tukey's multiple-comparison test. In **j**–**l**, results are normalized to total cell number. Panel **a** created in BioRender. Yuan, L. (2025) https://BioRender.com/nf4r0g6.

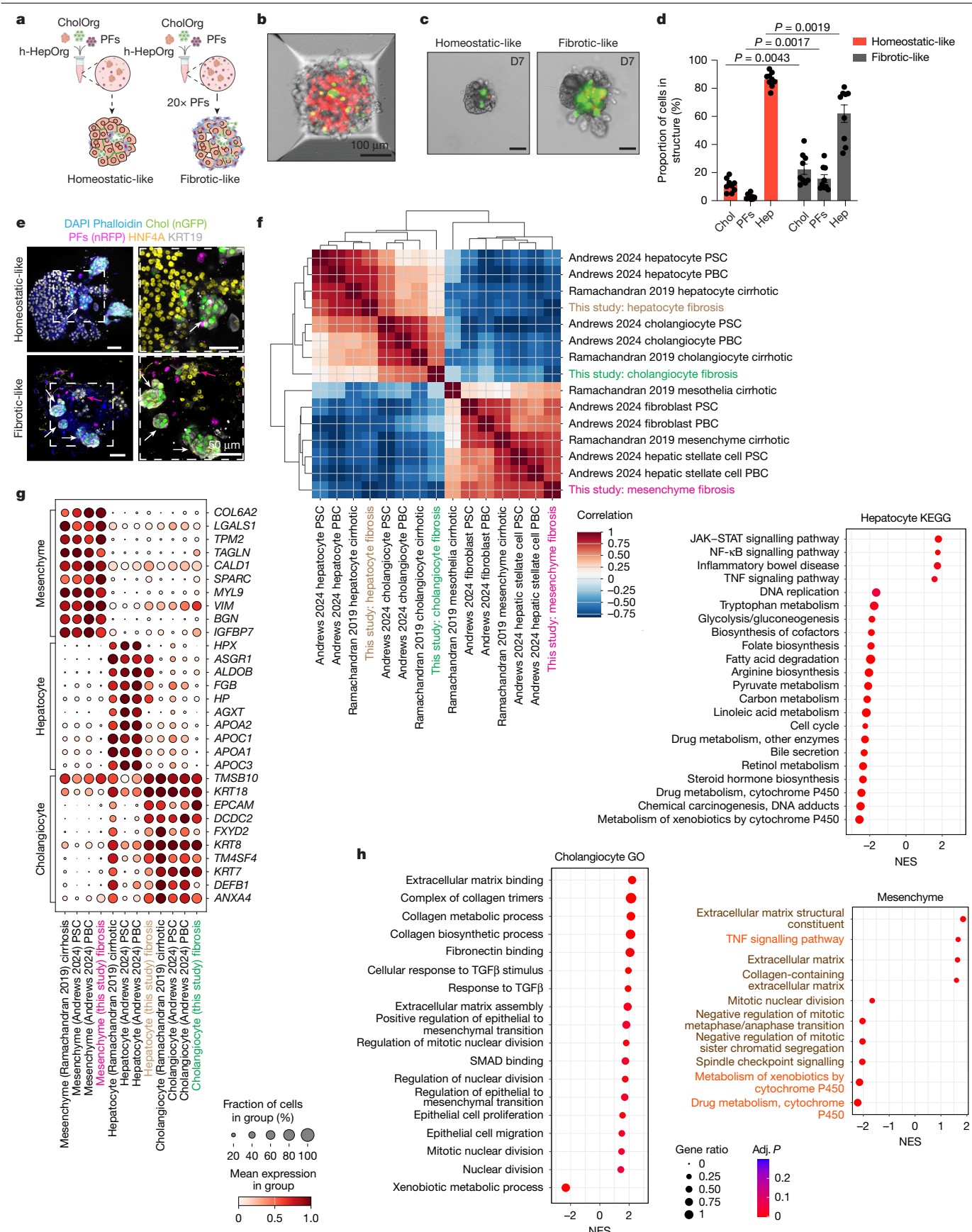

**Fig. 5 |** See next page for caption.

**Fig. 5 | Periportal assembloids mimic aspects of human biliary fibrosis.**
**a**–**h**, Assembloids were generated by assembling h-HepOrgs, cholangiocytes/
ductal cells derived from h-CholOrgs (nGFP$^+$) and portal fibroblasts (nRFP$^+$)
at a ratio of 1 h-HepOrg to 25 portal fibroblasts and 100 cholangiocytes
(homeostatic-like assembloids) or 1 h-HepOrg to 500 portal fibroblasts
(20 times more mesenchymal cells) and 100 cholangiocytes (fibrotic-like
assembloids) (**a**); 24 h or 7 days later, the cultures were collected and processed
for immunofluorescence (**b**–**e**) or RNA-seq (**f**–**h**) analysis. **a**, Experimental
design. **b**, Representative bright-field image of an assembloid at 24 h in an
AggreWell plate. Scale bar, 100 µm. **c**, Representative bright-field images of a
homeostatic-like assembloid (left) and a matching fibrotic-like assembloid
with 20-fold excess mesenchymal cells (right) at 7 days. Scale bars, 100 µm.
**d**, Cell composition of homeostatic-like and fibrotic-like assembloids. Dots
correspond to the percentages of hepatocytes, cholangiocytes and portal
fibroblasts in each structure. The graph shows the mean ± s.e.m. for
assembloids from three independent experiments. Two-sided paired (by donor)
Student's *t* test; *P* values are shown in the figure. **e**, Representative images

of homeostatic-like (top) and fibrotic-like (bottom) assembloids stained for
hepatocyte (HNF4A; yellow, magenta arrow) and cholangiocyte (KRT19; white
and nGFP, white arrow) markers. Mesenchyme is marked by nRFP (magenta).
The low-magnification views also show phalloidin (membrane; blue) and DAPI
(nuclei; blue) channels. Scale bars, 50 µm. **f**, Correlation between the three
different populations in fibrotic assembloids (this study) and corresponding
cells in diseased human liver cell atlases. **g**, Dot plot for hepatocyte,
cholangiocyte and mesenchyme markers in fibrotic-like assembloids (this
study) and liver tissue datasets. References used for comparison are refs. 8,13.
**h**, GSEA comparing fibrotic and homeostatic assembloids. Dot plots show
selected enriched terms in fibrotic versus homeostatic assembloids. Dot
colour corresponds to the adjusted *P* value (permutation test in clusterProfiler,
adjusted using the Benjamini–Hochberg method). Dot size, gene ratio (the
number of core enrichment genes divided by the total number of genes in
the pathway). Orange, KEGG pathways; brown, Gene Ontology (GO) terms.
Full list in Supplementary Data 4. Panel **a** created in BioRender. Yuan, L. (2025)
https://BioRender.com/ejby6iy.

---

for at least 2 weeks with no evidence of cell death or proliferation
(Extended Data Fig. 7i–l).

Next, we used single-cell RNA-seq (scRNA-seq) analysis to benchmark
our model to in vivo human liver tissue. In clustering, PCA and correla-
tion analysis, the assembloid cells mostly overlapped with the corre-
sponding cells in human liver cell atlases[8,11–13,49] (Fig. 4g and Extended
Data Fig. 8a). Hepatocytes, cholangiocytes and mesenchymal cells from
assembloids expressed classical markers of their in vivo counterparts
(hepatocytes: *ALB*, *HNF4A*; cholangiocytes: *KRT7*, *KRT19*; mesenchy-
mal cells: *VIM*, *THY1*) (Fig. 4h). Gene set enrichment analysis (GSEA)
confirmed that mesenchymal cells were highly enriched for signatures
of extracellular matrix (ECM) organization and cell adhesion, cholan-
giocytes were enriched for cytoskeleton and cell–cell communication,
and hepatocytes were enriched for fatty acid metabolism, comple-
ment and drug metabolism, similar to human liver tissue (Extended
Data Fig. 8c,d).

Interestingly, we observed heterogeneous expression of classical
zonated hepatocyte markers, with a fraction of hepatocytes express-
ing periportal markers (*SAA1*, *SAA2* and *APOA1*) and others expressing
pericentral markers (*CYP2E1*) (Fig. 4h and Extended Data Fig. 8b). To
investigate whether the periportal assembloid microenvironment and
the interaction with portal ductal and mesenchymal populations could
promote a more portalized identity, we compared the gene expres-
sion profile of hepatocytes from h-HepOrgs cultured in DM with that
of assembloids (also cultured in DM). Notably, hepatocytes within
assembloids exhibited higher expression of periportal markers, includ-
ing *SAA1*, *SAA2*, *HAMP* and *APOA1*, whereas pericentral genes such as
*CYP2E1*, *CYP3A4* and *GLUL* were downregulated relative to h-HepOrgs
cultured alone (Fig. 4i). Staining for SAA1 and SAA2 confirmed the spa-
tially heterogenous expression of these portal markers, with positive
cells overlapping with regions of E-cadherin-high cells (Extended Data
Fig. 9a), in agreement with our scRNA-seq results (Fig. 4h,i).

Notably, periportal assembloids outperformed differentiated
h-HepOrgs in both urea production and gluconeogenesis (both of which
are portal functions), while the drug-metabolizing capacity associated
with pericentral hepatocytes was less pronounced than in hepato-
cyte organoids, in line with their more portal-like nature (Fig. 4j,k and
Extended Data Fig. 9b). As expected, periportal assembloids retained
core hepatocyte functions, with albumin secretion increasing over time
to levels matching those for hepatocyte organoids and exceeding those
of 1-day 2D primary hepatocyte cultures (Fig. 4l).

These findings suggest that the periportal microenvironment within
assembloids could promote acquisition of a more portal-like hepato-
cyte identity. In line with this hypothesis, we noted that some hepato-
cyte membranes joined the lumen of the bile ducts, similar to what
we observed in tissue in vivo and suggestive of physiological cell–cell
contact between these cell types (Extended Data Fig. 9c,d).

Taking these data together, our human liver periportal assembloid
model captures the gene expression, the cell interactions and aspects
of the tissue architecture of the native human liver periportal region.

## Assembloids model features of biliary fibrosis

Portal mesenchyme often contributes to myofibroblast populations
in human fibrosis[50]. Hence, we next investigated whether we could
use our human assembloid model containing portal fibroblasts to
recapitulate aspects of human liver disease in vitro, specifically biliary
fibrosis. Interestingly, increasing mesenchymal cell numbers (20-fold)
while keeping the other cell numbers constant (even from the same
source tissue) altered the assembloids' composition, with increased
cholangiocyte (GFP$^+$KRT19$^+$) numbers while hepatocyte (HNF4A$^+$)
numbers decreased (Fig. 5a–e). Ki-67 staining indicated that cholan-
giocytes exhibited early proliferative responses to fibrotic cues, and
cleaved caspase 3 staining revealed that the reduction in hepatocyte
numbers was associated with increased cell death occurring, at least in
part, through apoptosis (Extended Data Fig. 10a–c). This finding was
consistent with our observations in mouse assembloids[51], suggesting
a conserved mechanism across species.

scRNA-seq clustering and correlation analyses revealed that the cells
from assembloids with excess mesenchyme recapitulated the transcrip-
tional signatures of diseased human livers[8,13] (Fig. 5f). The top markers
identifying the three cell populations in the corresponding patient
datasets were also highly expressed in the corresponding assembloid
cells (Fig. 5g), and GSEA revealed that mesenchyme and cholangiocytes
from fibrotic, but not homeostatic, assembloids had increased expres-
sion of proteins involved in collagen and matrix deposition processes
(Fig. 5h, Extended Data Fig. 10f,g and Supplementary Data 4). Similarly,
cholangiocytes, but not mesenchyme, exhibited signatures of pro-
liferation (Fig. 5h and Extended Data Fig. 10f), in agreement with the
increased number of GFP$^+$ cholangiocytes detected (Fig. 5c,d). These
gene signatures (increased matrix and cholangiocyte numbers) are
reminiscent of the fibrotic tissue from human patients with biliary
fibrosis and primary sclerosing cholangitis (PSC)[13,50]. We therefore refer
to assembloids with excess mesenchyme as 'fibrotic-like', to distinguish
them from the 'homeostatic-like' assembloids with homeostatic num-
bers of mesenchymal cells.

Notably, hepatocytes from fibrotic-like assembloids were positively
enriched for gene sets related to inflammatory reactions, including
tumour necrosis factor (TNF) signalling, several interleukins (IL-4 and
IL-6), and NF-κB, JAK–STAT and Toll-like receptor cascades (Fig. 5h and
Extended Data Fig. 10d,e). Conversely, cell cycle signatures and hepato-
cyte functions such as bile secretion and lipid and drug metabolism were
negatively enriched (Fig. 5h). Both hepatocytes and cholangiocytes
from fibrotic assembloids were also highly enriched in transforming

growth factor-β (TGFβ) signalling signatures (Fig. 5h and Extended Data Fig. 10d–f), mirroring the transcriptional changes in patients with biliary fibrosis[13].

Morphologically, we observed that fibrotic-like assembloids, but not matching homeostatic assembloids, exhibited a cystic-like phenotype reminiscent of cholangiocyte organoids (Extended Data Fig. 10h,i). This observation was in line with the immunofluorescence analysis, which indicated that in fibrotic-like assembloids some hepatocytes (HNF4A⁺GFP⁻) were positive for the cholangiocyte marker KRT19, and opened lumina, resembling the polarity of simple ductal epithelium, suggestive of potential hepatocyte-to-duct transdifferentiation (Extended Data Fig. 10j). Interestingly, all these phenotypes, including (1) enrichment of gene signatures for TNF, IL-4 and IL-6 signalling; (2) increased hepatocyte apoptosis; and (3) increased expression of cholangiocyte markers, have been reported in patients with fibrosis as well as in recent liver cell atlases of patients with PSC and primary biliary cirrhosis (PBC)[13,52]. These results combined suggest that our assembloid model with excess mesenchyme mimics some aspects of human biliary fibrosis as seen in cholangiopathies, including PSC and PBC.

## Discussion

Failure in maintaining the intricate cellular organization and multidirectional interactions of the cells within the liver leads to chronic disease, often presenting with cholestasis and fibrosis, which progresses to cirrhosis and cancer[53,54]. Despite being reductionist by nature, ex vivo systems offer powerful tools to dissect disease mechanisms. We recently showed that mouse periportal assembloids model key architectural features of the in vivo tissue and can serve as a tractable in vitro model to investigate universal principles of bile canaliculi formation, cholestatic injury or fibrogenesis[51]. However, species-specific differences in drug metabolism, toxicity or pathophysiology necessitate the development of complementary human models that capture patient-specific features to better understand disease mechanisms and identify therapeutics.

Recent advances in human liver models underscore the ongoing efforts and broad interest in developing physiologically relevant in vitro systems. These include induced pluripotent stem cell-derived hepatocyte organoids harbouring liver sinusoidal endothelial cell (LSEC)-like cells[55] or exhibiting dual zonation[56], functional hepatocyte organoids derived from cryopreserved hepatocytes[57], mass generation of hepatobiliary organoids[58], co-cultures of dermal fibroblasts with hepatocyte spheroids[59], and mouse fibroblasts aggregated with hepatocyte spheroids and cholangiocyte organoids[60]. However, a model capable of recapitulating the multicellular periportal liver tissue organization and cellular interactions ex vivo—while it would enable inter-individual comparative studies and investigation of patient-specific disease traits—has not yet been developed.

Here we overcome this challenge by establishing long-term-expandable h-HepOrgs from adult patient liver tissue and combining them with h-CholOrgs and human portal mesenchyme to form complex periportal liver assembloids. These assembloids recapitulate essential structural and functional features of the native human periportal region and, upon manipulation, model aspects of human biliary fibrosis. Our h-HepOrg model enables long-term expansion while preserving functional drug-metabolizing capabilities and capturing patient-to-patient variability, including differences in metabolic enzymes and disease-predisposing genes. At both the cellular and mesoscale levels, h-HepOrgs mimic fine architectural features such as bile canaliculus morphology and display heterogeneous expression of zonated hepatocyte genes. Although we observed variability in bile canaliculi morphology among organoids derived from different donors, whether this reflects true patient-to-patient differences will require further investigation.

Interestingly, assembloids exhibited increased portal-region functional features. Whether direct interactions between hepatocytes, cholangiocytes and portal mesenchyme are sufficient to instruct portal-specific hepatocyte identity remains an open question. Likewise, the possibility that hepatocyte subpopulations at the onset of culture influence differential responses to microenvironmental cues cannot be excluded. Our modular, 'self-organized Lego-like' assembloid platform provides a unique system to systematically manipulate individual cellular components and begin to dissect, in a controlled setting, how specific microenvironmental signals or cell–cell interactions contribute to human hepatocyte identity and zonation.

Of note, by increasing the number of portal mesenchymal cells, we generated assembloids that recapitulated several aspects of human cholestatic disease and biliary fibrosis. One caveat, though, is the lack of other mesenchymal cells, immune cells and portal vasculature (portal vein and hepatic artery), which limits the formation of a true periportal triad. Incorporating these will be crucial to reproduce all aspects of liver disease.

In summary, the patient-derived hepatocyte organoids and periportal assembloid models we present here hold the potential to initiate a new era in diverse areas of liver research, including in diagnostics, toxicology, personalized drug efficacy screening and cellular transplantation therapy.

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

## Methods

### Human specimens

All human liver tissues used in this study were obtained after informed consent was obtained from patients undergoing operations at either the Department of Visceral, Thoracic and Vascular Surgery (VTG), University Hospital Carl Gustav Carus Dresden (UKD) or Leipzig University Medical Center. Informed consent was obtained from all participants. Use of the human samples for this study was approved by the corresponding institutional review boards of either the University Hospital Carl Gustav Carus Dresden (ethical vote BO-EK-57022020, ratified on 10 March 2020) or the Leipzig University Hospital (ethical vote: registration number 322/17-ek, date 10 June 2020 ratified 30 November 2021 and registration number 450/21-ek, date 21 November 2021 ratified on 4 October 2024). Five samples (F-PHH1–F-PHH5) were obtained from cryopreserved hepatocytes from Lonza Pharma&Biotech-Bioscience Solutions. Resected liver specimens were obtained from patients undergoing partial hepatectomy for benign or malignant conditions (for example, colorectal liver metastases, hepatocellular carcinoma or benign focal lesions). Only histologically normal, non-tumorous tissue adjacent to the resection site was used for organoid derivation. Clinical background information (sex, age, diagnosis/surgical indication) is provided in Supplementary Tables 1 and 2. Commercially obtained cryopreserved PHHs were derived from the livers of healthy donors deemed unsuitable for transplantation. Commercial number and supplier are given in Supplementary Table 2.

All procedures involving human material were conducted in accordance with the Declaration of Helsinki and institutional ethical guidelines.

### Isolation of primary human hepatocytes and cholangiocytes

PHHs were isolated using a two-step collagenase perfusion method as described in refs. 34,61. The human liver tissue received from UKD was perfused with solution A (composed of 10 mM HEPES and 2.5 mM EGTA in HBSS) at 39 °C for at least 20 min, with a rate of 15 ml per 20 s. Subsequently, the perfusion solution was switched to solution B (containing 100 mM HEPES, 4.8 mM $CaCl_2$ and 1 g $l^{-1}$ collagenase P, in HBSS) and perfused at 37 °C for 5–15 min, also at a rate of 15 ml per 20 s. The digestion process was halted by adding cold William's E medium supplemented with 1% HEPES, 1% GlutaMAX and 1% penicillin/streptomycin. PHHs were detached from the tissue by shaking using forceps and combing the cells out of the tissue. Afterwards, they were filtered through a 100-μm nylon cell strainer. Cells were then spun at 50$g$ for 5 min, and the resulting pellet was resuspended in cold William's E medium supplemented with 1% HEPES, 1% GlutaMAX and 1% penicillin/streptomycin. The cell suspension was kept cold and centrifuged again at 50$g$ for 5 min.

For samples obtained from Leipzig University Hospital, the perfusion procedure differed slightly: solution A (composed of 10 mM HEPES (Carl Roth), 143 mM NaCl, 6.7 mM KCl, 2.4 mM EGTA, 5 mM $N$-acetyl-L-cysteine, 11 mM D-glucose (all provided by Sigma-Aldrich) and 32 U $l^{-1}$ human insulin (Eli Lilly) in double-distilled water (pH 7.4)) at 39 °C with a rate of 25 ml per minute for at least 20 min. The perfusion solution was then switched to solution B (composed of 67 mM NaCl, 6.7 mM KCl, 10 mM HEPES, 0.5% BSA, 4.8 mM $CaCl_2 × 2H_2O$ (all provided by Sigma-Aldrich), and 1 g $l^{-1}$ collagenase P (Roche) in $ddH_2O$ (pH 7.6), diluted 1:2 in stop solution (composed of DPBS with $Ca^{2+}$, $Mg^{2+}$ (Gibco), supplemented with 16.7% FBS (Merck)) and perfused at 39 °C for 5–15 min at a rate of 25 ml $min^{-1}$. The digestion was stopped by adding cold stop solution. Hepatocytes were filtered through a funnel with gauze (Hartmann) and centrifuged at 51$g$ for 5 min. Cell pellets were washed in DPBS with $Ca^{2+}$, $Mg^{2+}$, centrifuged at 51$g$ for 5 min and resuspended in William's E medium supplemented with 10% FBS (Merck), 15 mM HEPES, 1 mM sodium pyruvate, 1% penicillin/streptomycin, 1% MEM NEAA (all provided by Gibco), 1 μg $ml^{-1}$ dexamethasone

(Jenapharm) and 32 U $l^{-1}$ human insulin (Eli Lilly). The isolated PHHs were shipped overnight in ChillProtec plus medium (Biochrom).

Cryopreserved hepatocytes (F-PHH1–F-PHH5; Supplementary Table 2), commercially available from Lonza, were defrosted using human hepatocyte thawing medium (Lonza) following the manufacturer's instructions.

The isolated PHH preparations (either from fresh tissue from Dresden or Leipzig Hospital or commercially available frozen hepatocytes) were enriched for both EpCAM-negative (hepatocytes) and EpCAM-positive (cholangiocytes) by MACS using an anti-human CD326 antibody (BioLegend) and anti-biotin microbeads (Ultra Pure, Miltenyi) following the manufacturer's instructions. The EpCAM-negative fraction with a viability of >50% (Supplementary Table 1) was used to generate hepatocyte organoids as described below (see 'Hepatocyte organoid culture'). The EpCAM-positive fraction, formed by human cholangiocytes, was used to generate h-CholOrgs as described previously[4,5] and in 'Cholangiocyte organoid culture'. A digestion method without perfusion, as the one detailed in ref. 4, only generated h-CholOrgs. h-HepOrgs were not formed under non-perfused protocols.

The complete list of patients used and the comparative between digestion and perfusion are provided in Supplementary Tables 1 and 2.

### Flow cytometry validation of PHH purity following MACS enrichment

Freshly isolated PHHs and MACS-enriched EpCAM-negative PHHs (as described above) were centrifuged at 80$g$ for 5 min. Pellets were resuspended in HBSS containing 2% FBS and incubated on ice for 10 min (blocking). After centrifugation (80$g$, 5 min), cells were resuspended in HBSS with 1% FBS, stained with EpCAM-Alexa 488 (5 μl per test; BioLegend), and incubated for 45 min on ice. Cells were then washed twice with HBSS containing 1% FBS, centrifuged and resuspended in 200 μl HBSS with 1% FBS, DAPI (1:1,000) and DNase I (1:1,000) for flow cytometry analysis.

### Cholangiocyte organoid culture

For cholangiocyte organoid cultures, EpCAM-positive cholangiocytes were mixed with Matrigel growth factor reduced (Matrigel, Corning) or Cultrex basement membrane extract 2 (BME2) (Cultrex-RGF basement membrane extract type 2, BME2 (AMSBIO) at 50,000 cells per 50 μl in each well of a 24-well plate and cultured at 37 °C and 5% $CO_2$ in h-CholOrg EM medium as described in refs. 4,5: AdDMEM/F12 medium containing 1% HEPES, 1% penicillin/streptomycin, 1% GlutaMAX, 1× B27 and 1.25 mM $N$-acetylcysteine (Sigma) supplemented with 10 nM gastrin (Merck/Sigma), 50 ng $ml^{-1}$ hEGF (Peprotech), 10% RSPO1 conditioned medium (homemade), 100 ng $ml^{-1}$ FGF10 (Peprotech), 10 mM nicotinamide (Merck/Sigma) and 25 ng $ml^{-1}$ HGF (Peprotech)], 5 μM A8301 (Tocris) and 10 μM forskolin (Tocris, 1099). For the first 3–5 days in culture, this medium was supplemented with 30% WNT3a conditioned medium (Wnt-CM) (homemade), 25 ng $ml^{-1}$ Noggin (Peprotech) and 10 μM ROCK inhibitor (Ri) (Y-27632, Merck/Sigma). The grown cholangiocyte organoids were passaged at a 1:3 ratio once a week as described in ref. 4. Organoid lines were routinely tested for mycoplasma.

### Hepatocyte organoid culture

For hepatocyte organoid cultures, the isolated PHHs (EpCAM-negative fraction) were mixed with Matrigel (Corning) or BME2 (AMSBIO), and 12,500–50,000 cells were seeded in 50-μl domes per well in 24-well plates and incubated at 37 °C and 5% $CO_2$. After gel solidification, culture medium was added. The culture medium was based on the medium from ref. 29 for hepatoblasts (MM) with modifications and the addition of WNT and YAP activation. The medium was composed of AdDMEM/F12 (Invitrogen) supplemented with 1% HEPES, 1% GlutaMAX (ThermoFisher), 1% penicillin/streptomycin (ThermoFisher), 1× B27 without retinoic acid (Gibco), 1.25 mM $N$-acetylcysteine (Sigma), 10 nM gastrin (Sigma) and the following growth factors: 50 ng $ml^{-1}$

hEGF (Peprotech), 15% RSPO1 conditioned medium (home-made), 100 ng ml$^{-1}$ FGF10 (Peprotech), 100 ng ml$^{-1}$ FGF7 (Peprotech), 50 ng ml$^{-1}$ HGF (Peprotech), 10 mM nicotinamide (Sigma, for EM1 medium only), 2 µM A83-01 (Tocris), 3 µM CHIR99021 (Tocris), 10 µM Y-27632 (Tocris), 0.5 nM Wnt surrogate Fc fusion protein as in ref. 40 (IPA, N001) and 10 µM TRULI (Axon) or 10 µM TDI-011536 (Selleckchem).

After 1 week to 10 days, the organoids were removed from the Matrigel or BME2, mechanically dissociated into small fragments using TrypLE Express (Gibco) and transferred to fresh Matrigel or BME2. Passaging was performed once per week at a 1:2 split ratio for at least 3 months. For preparation of frozen stocks, the organoid cultures were dissociated, mixed with Recovery cell culture freezing medium (Gibco) and frozen following standard procedures.

For the optimization of culture conditions, medium component screening experiments were performed in which each of the components Amphiregulin (AREG; 100 ng ml$^{-1}$; R&D Systems), dexamethasone (1.6 µM; Sigma), G-CSF (50 ng ml$^{-1}$; R&D Systems), IL-6 (2 ng ml$^{-1}$; R&D Systems), M-3m3FBS (phospholipase C activator; 25 µM; Tocris), TGFα (100 ng ml$^{-1}$) and TRULI (Axon) was added to our previously published mouse hepatoblast medium (MM[29]) with minor modifications: AdDMEM/F12 (Invitrogen) supplemented with 1% HEPES, 1% GlutaMAX, 1% penicillin/streptomycin, 1× B27 without retinoic acid, 1.25 mM N-acetylcysteine, 10 nM gastrin, 50 ng ml$^{-1}$ hEGF, 15% RSPO1 conditioned medium, 100 ng ml$^{-1}$ FGF10, 100 ng ml$^{-1}$ FGF7, 50 ng ml$^{-1}$ HGF, 10 mM nicotinamide, 2 µM A83-01, 3 µM CHIR99021, 10 µM Y-27632 and 0.5 nM Wnt surrogate Fc fusion protein. Note that addition of TRULI alone resulted in a significant increase in organoid formation efficiency (Fig. 1c,e). However, after 1–2 splits, the cultures rapidly deteriorated and could not be expanded further (Fig. 1f).

For h-HepOrg hepatic differentiation, h-HepOrgs were expanded in EM2 medium above, split, seeded and cultured for 2–5 days under EM1 culture medium, after which the medium was changed to DM medium composed of AdDMEM/F12 supplemented with 1% HEPES, 1% GlutaMAX, 1% penicillin/streptomycin, 1× B27 without retinoic acid, 1.25 mM N-acetylcysteine, 50 ng ml$^{-1}$ hEGF, 15% RSPO1 conditioned medium, 50 ng ml$^{-1}$ HGF, 2 µM A83-01, 3 µM CHIR99021, 10 µM Y-27632, 0.5 nM Wnt surrogate Fc fusion protein, 100 ng ml$^{-1}$ FGF19 (R&D Systems) and 1.6 µM dexamethasone (Sigma). DM was changed every 2–3 days for 7 days.

For organoid formation efficiency, primary hepatocytes were isolated and cultured in different media as described above. To prevent organoids from fusing, 25,000 (for EM2 medium) or 50,000 (all other media) viable hepatocytes (viability of >80%) were plated in 50 µl Matrigel or BME2 and cultured as described above. After 12–14 days, organoid numbers were counted and results were expressed as a percentage relative to the initial seeding cell numbers. Organoid lines were routinely tested for mycoplasma.

## Isolation of human liver portal fibroblasts

Human liver portal fibroblasts were isolated from human liver tissues by collagenase digestion. In brief, human liver tissue was minced and rinsed with cold DMEM (Gibco) supplemented with 1% HEPES, 1% GlutaMAX, 1% penicillin/streptomycin and 1% FBS. Minced tissues were incubated with a collagenase solution consisting of 2.5 mg ml$^{-1}$ collagenase D (Roche), 0.1 mg ml$^{-1}$ DNase I (Sigma), 1× B27 without retinoic acid, 1.25 mM N-acetylcysteine, 5% RSPO1 conditioned medium and 10 µM Y-27632 in DMEM supplemented with 1% HEPES, 1% GlutaMAX and 1% penicillin/streptomycin. Incubation was carried out for 30–60 min at 37 °C on a shaker set at 120 rpm. The digestion was halted by adding cold DMEM supplemented with 1% HEPES, 1% GlutaMAX, 1% penicillin/streptomycin and 1% FBS. The suspension was then filtered through a 70-µm cell strainer and centrifuged for 5 min at 300g. After removing the supernatant, the cell pellet was resuspended in cold DMEM supplemented with 1% HEPES, 1% GlutaMAX, 1% penicillin/streptomycin and 1% FBS. The suspension was centrifuged again for 5 min at 300g, and

the resulting pellet was resuspended in cold DMEM supplemented with 1% HEPES, 1% GlutaMAX, 1% penicillin/streptomycin and 20% FBS. For sorting, portal fibroblasts were stained with anti-human CD90 (THY1) conjugated to APC, anti-human CD140a (PDGFRα) conjugated to PE, anti-CD11b/CD31/CD45 conjugated to PECy7 and anti-EpCAM conjugated to Alexa 488 for 30 min on ice and washed twice. THY1-positive portal fibroblasts were sorted using a BD FACSAria Fusion and cultured in DMEM supplemented with 1% HEPES, 1% GlutaMAX, 1% penicillin/streptomycin and 20% FBS at 37 °C and 5% CO$_2$ until used for assembloid formation or frozen for biobanking. Portal fibroblast cultures were routinely tested for mycoplasma.

## Viral infection

For portal fibroblast infections, cultures (passage 0 or 1) grown in DMEM$^{+++}$ supplemented with 20% FBS (Merck/Sigma, F7524) were washed with PBS and dissociated to single cells by incubation with 1× TrypLE for 6 min at 37 °C. The cell concentration was determined by manual counting in a haemocytometer, and 10,000 cells were plated into each well of a 48-well plate and the medium mixed with nRFP- or nGFP-encoding lentivirus (LVP360-R and LVP360-G, GenTarget) to achieve a multiplicity of infection (MOI) of 10–30, then replaced after 12 h and the solution was changed after 72 h.

For cholangiocyte organoid infection, duct cells (passage 0 or 1) were extracted from Matrigel and digested with TrypLE to prepare single-cell suspensions as described in ref. 5, which were then manually counted using a haemocytometer to determine cell concentration. In a 48-well plate, 150 µl of cells and 50 µl of virus suspension from nRFP- or nGFP-encoding lentivirus (LVP360-R and LVP360-G, GenTarget) were added to achieve a MOI of 10–30, mixed thoroughly, centrifuged at 600g for 60 min at 32 °C and incubated for 6 h at 37 °C in 5% CO$_2$. Cells were collected in 1.5-ml tubes and centrifuged at 600g for 5 min, the virus-containing medium was discarded and cells were resuspended in 25 µl of Matrigel, followed by the addition of cholangiocyte medium (supplemented with 30% WntCM, 25 ng ml$^{-1}$ noggin and 10 µM Y-27632 for the first 3 days).

## Periportal assembloids

To generate liver periportal assembloids comprising hepatocytes, cholangiocytes and portal fibroblasts, we first prepared the cellular components as follows: nGFP-labelled cholangiocyte organoids (passage 5–11), grown in cholangiocyte expansion medium (h-CholOrg-EM) as detailed above, were collected from Matrigel using cold AdDMEM/F12 (Invitrogen, 12634010) containing 1% HEPES (ThermoFisher, 15630-056), 1% penicillin/streptomycin (ThermoFisher, 15140-122) and 1% GlutaMAX (ThermoFisher, 35050038). Matrigel was removed and organoids were dissociated to single cells using prewarmed 1× TrypLE (Gibco) for 7–12 min at 37 °C. nRFP-labelled portal fibroblast cultures (passage 5–12) grown in DMEM$^{+++}$ with 20% FBS (Merck/Sigma, F7524) were washed with PBS and dissociated to single cells by incubation with 1× TrypLE for 6 min at 37 °C. Both single-cell suspensions were spun at 200g for 5 min, resuspended in DM medium as described above but without A8301, and then manually counted with a haemocytometer to determine cell concentration. Cultured h-HepOrgs from EM2 medium were split and transferred to EM1 medium for 2 days and then to DM medium for 3 days. Hepatocyte organoids were then collected and washed using cold AdDMEM/F12 supplemented with 1% HEPES, 1% penicillin/streptomycin and 1% GlutaMAX and incubated for 10 min on ice using cold cell recovery solution (Corning, 354253) to remove the ECM. h-HepOrgs were then resuspended using DM without A8301 and placed into a low-attachment six-well plate; differentiated organoids (with bubbly morphology) were selected and hand-picked under a stereomicroscope.

To define an approach for human periportal liver assembloid formation, several iterations were performed. First, we sought to identify a medium that would support assembloid formation, that is, the culture of all three cell types: hepatocytes, cholangiocytes/ductal cells and

portal mesenchyme without overgrowth of any of them, we tested several media and found that a minor adaptation of the DM medium used for h-HepOrgs differentiation without A8301 (assembloid medium) supported culture of the three cell types while preventing their overgrowth. To determine the optimal quantities of the three cell types required for periportal assembloid formation, we first investigated the proportions of portal fibroblasts and ductal cells in healthy human periportal liver tissue. We observed that the ratio varies from donor to donor from 1:1 to 4:1 ductal cells per fibroblast. Therefore, we tested this range of ratios in vitro by varying the proportions of mesenchyme and ductal cells that were mixed with a single h-HepOrg (~200-μm diameter). In short, in 96-well low-attachment U-bottom plates (Corning), we assembled (as described below) 1 h-HepOrg with 25 portal fibroblasts and 25, 50, 100 or 200 cholangiocytes, or with 100 cholangiocytes and 50 or 100 portal fibroblasts. We selected the proportion of 25 portal fibroblasts per 100 cholangiocytes/ductal cells. In AggreWell plates (AggreWell 800, Stem Cell Technologies), we scaled up proportionally, taking into account that the AggreWell 800 plate has 300 microwells in each well and used 7,500 portal fibroblasts, 30,000 cholangiocytes and 100 h-HepOrgs (proportion of 1 h-HepOrg to 75 portal fibroblasts and 300 cholangiocytes).

For non-healthy/non-physiological ratios, we used 500 portal fibroblasts, 100 cholangiocytes and 1 h-HepOrg for 96-well low-attachment U-bottom plates, and 15,0000 portal fibroblasts, 30,000 cholangiocytes and 50 h-HepOrgs for AggreWell plates.

For assembly in MW96, we mixed fibroblasts and cholangiocytes in 96-well low-adhesion U-bottom plates using 150 μl DM (without A8301) with 2.4 mg ml$^{-1}$ methylcellulose (MeC; Sigma, M6385) and spun at 50$g$ for 5 min. Individual h-HepOrgs were then added to the well and the mixture was incubated for 18–24 h at 37 °C and 5% CO$_2$. For assembly in AggreWell plates, plates were first pretreated as recommended by the manufacturer. Then, ductal and mesenchymal cells together with h-HepOrgs were mixed in 1.5 ml DM (without A8301) with 2.4 mg ml$^{-1}$ methylcellulose, spun down for 5 min at 50$g$ and incubated for 18–24 h at 37 °C and 5% CO$_2$. After 18–24 h in suspension in the 96-well/AggreWell plate, the cell suspension was collected with a 1-ml pipette and transferred to a low-attachment 6-well plate. The structures were manually picked under a stereomicroscope and seeded in 25 μl Matrigel dome in prewarmed 48-well plates. The Matrigel was allowed to solidify for 30 min at 37 °C in 5% CO$_2$, and the wells were overlayed with an additional 300 μl of DM (without A8301). The medium was changed every 3–4 days. Under these conditions, 70% of the initial cholangiocytes formed a lumen. Raw data were incorporated into the quantification of periportal-like spatial organization in assembloids (source data for Extended Data Fig. 7e).

## Immunostaining of organoids and assembloids

For immunofluorescence staining, organoids and assembloids were first extracted from Matrigel with ice-cold Cell Recovery solution and then fixed for 30 min with 4% paraformaldehyde (PFA) at 4 °C. Fixed organoids were washed and transferred to μ-Slide 8-well chamber slides (glass bottom; Ibidi). Blocking and permeabilization were performed for 1 h at room temperature in PBS containing 2% BSA and 0.1%, 0.2%, 0.5% or 1% Triton X-100 depending on the antigen (Supplementary Data 5). The samples were incubated with primary antibodies overnight at 4 °C in blocking solution. After that, the antibody was washed with three washes with PBS and the samples were incubated overnight at 4 °C or for 8 h at room temperature with secondary antibodies diluted in blocking solution and, if required, also phalloidin and DAPI were added to the secondary antibody mix. The samples were washed three times with PBS and subsequently cleared using fructose-glycerol clearing solution (25 ml glycerol, 5.3 ml dH$_2$O and 22.5 g fructose–60% glycerol and 2.5 M fructose). The samples were stored in PBS until they were cleared for imaging as described above. The antibodies and dilutions used are listed in Supplementary Data 5.

For haematoxylin and eosin (H&E) staining, organoids were collected in cold DPBS (Gibco) and fixed with 4% PFA for 30 min and dehydrated and embedded in paraffin using standard methods. Paraffin sections (8 μm) were cut and stained for H&E using standard protocols.

## Immunostaining of thin and thick tissue sections

For thin tissue sections (8–12 μm) and staining, human liver tissues were fixed in 10% formalin overnight with rolling at 4 °C. After fixation, tissues were washed with PBS and incubated with 10% sucrose for 1–2 h, then transferred to 30% sucrose in PBS for 24 h and subsequently embedded in OCT compound (VWR, 361603E) to generate OCT cryopreserved tissue blocks. Tissue blocks were cryosectioned on a CryoStar NX70 cryostat (ThermoScientific). Sections were blocked in PBS with 10% donkey serum (DS) and 0.1% Triton X-100 for 2 h at room temperature, incubated with primary antibodies diluted in PBS with 3% donkey serum and 0.1% Triton X-100 overnight at 4 °C and subsequently washed and incubated with secondary antibodies diluted in 0.05% BSA in PBS and DAPI for 2 h at room temperature. Sections were mounted in Vectashield. The list of antibodies used is available in Supplementary Data 5.

For thick tissue sections and staining, the protocol from ref. 62 was used. Immediately after surgical resection, liver tissue samples were cut into smaller pieces and fixed in 4% PFA for 24 h on a rotator at 4 °C and washed three times with PBS, followed by quenching with 50 mM ammonium chloride solution (NH$_4$Cl) for 24 h and again washed three times with PBS. For storage, liver pieces were kept in PBS at 4 °C. For sectioning, livers were embedded in moulds with 4% low-melting agarose (Bio-Rad, 1613111) in PBS and cut into 50- or 100-μm-thick sections on a vibratome (Leica, VT1200S). For deep tissue imaging, if antigen retrieval was required, tissue sections were placed in Eppendorf tubes with prewarmed 1× citrate buffer (Sigma-Aldrich, C9999), pH 6, at 80 °C for 30 min in a shaking heating block and then washed three times with PBS. Tissue sections were permeabilized with 0.5% Triton X-100 in PBS for 1 h at room temperature. The primary antibodies were diluted in Tx buffer (0.2% gelatin, 300 mM NaCl and 0.3% Triton X-100 in PBS) and incubated for 48 h at room temperature. After washing three times for 15 min each with 0.3% Triton X-100 in PBS, the sections were incubated with secondary antibodies, DAPI (1 mg ml$^{-1}$; 1:1,000) and phalloidin for another 48 h. After washing three times for 15 min each with 0.3% Triton X-100 in PBS and three times for 1 min each with PBS, the optical clearing started by incubating the slices in 25% fructose for 4 h, continued in 50% fructose for 4 h, 70% fructose overnight, 100% fructose (100% wt/vol fructose, 0.5% 1-thioglycerol and 0.1 M phosphate buffer, pH 7.5) overnight, followed by a final overnight incubation in SeeDB solution (80.2% (wt/wt) fructose, 0.5% 1-thioglycerol and 0.1 M phosphate buffer)[63]. The samples were mounted in SeeDB. A list of antibodies and dyes used is available in Supplementary Data 5.

For immunohistochemistry of tissue sections from xenotransplanted mice, mouse liver tissue samples were cut into smaller pieces and fixed in 10% formalin overnight. Sections (4 μm) were subjected to immunohistochemical staining, which was performed using a Dako REAL EnVision detection system (Dako, K5007). Anti-human GAPDH antibody (Abcam) (Supplementary Data 5) was used as the primary antibody and nuclei were counterstained with haematoxylin. Stained tissues were viewed under a Virtual Slide System (Leica, ScanScope CS2).

The immunohistochemistry analysis for PDGFRA, DCN and ASPN in healthy human liver tissue was obtained from the publicly available image dataset from Human Protein Atlas (HPA)[64] (version 24protein-atlas.org). The corresponding URL is indicated in the figure legend.

## Imaging of organoids, assembloids and tissues

Bright-field images of organoids were obtained with a Leica DMIL LED inverted microscope and Leica DFC 450C camera or with a Leica M80 stereoscope and MC170HD camera and Leica LAS software.

H&E staining of organoids was obtained with a Leica DM4B microscope and DMC5400 camera and Leica LAS X software.

Confocal images of organoids and thick tissue sections were acquired on an inverted single-photon point scanning confocal microscope (Zeiss Cell Discoverer 7 with LSM 900 and Airyscan 2) using a Zeiss APOCHROMAT ×20/0.95-NA Autocorr air objective, with a tube lens of ×0.5 or ×1, and a voxel size of 0.4 × 0.4 × 0.5 μm or 0.5 × 0.5 × 0.5 μm for organoids and 0.3 × 0.3 × 0.3 μm for thick tissue sections. Laser lines 405, 488, 561 and 640 were used for excitation of fluorophores, and GaAsP-PMT detectors were used for detection. High-resolution Airyscan images were acquired using this system for imaging polarity in detail for the tissue sections with a voxel size of 0.0823 × 0.0823 × 0.3 μm. Image processing was done using Zen software or ImageJ/Fiji.

Imaging of assembloids and thin tissue sections was performed using an inverted multiphoton laser-scanning microscope (Zeiss LSM 780 NLO). To improve the resolution, image denoising was performed with deconvolution using HuygensPro. Raw image stacks were imported into the software, and a point spread function (PSF) was either estimated based on the imaging conditions (numerical aperture, wavelength and refractive index) or obtained from PSF calibration images. The HuygensPro classic maximum likelihood estimation (CMLE) algorithm was applied for deconvolution, with an iteration stop criterion based on optimal signal-to-noise ratio and minimal change in successive iterations.

### Image analysis

Quantification of the percentage of YAP-positive and YAP-negative nuclei was performed using Arivis 4D Pro software (version 4.2.0). The steps of the analysis pipeline included background correction, denoising, nuclear segmentation based on DAPI and quantification of the fluorescence intensity of YAP immunofluorescent staining in the nuclei. The total number of nuclei and the number of YAP-positive nuclei were quantified, and, subsequently, the number of YAP-negative nuclei was calculated by subtracting the number of YAP-positive nuclei from the total number of nuclei. Finally, the percentages of YAP-positive and YAP-negative nuclei were calculated.

Quantification of cytoplasmic to nuclear area was performed using Arivis 4D Pro software (version 4.2.0). For this, a representative 2D z slice was taken from each organoid. The analysis pipeline included preprocessing steps of background correction on the phalloidin channel (marking cell borders) and normalization and denoising on the DAPI channel (marking nuclei). To obtain the nuclear area, nuclear segmentation was done based on DAPI, followed by quantification of the total nuclear area. For the cytoplasmic area, segmentation was done based on phalloidin to obtain the outline of the area occupied by the cytoplasm. Finally, the ratio of cytoplasmic area to nuclear area was calculated.

For 3D visualization of bile canaliculi, high-resolution images were obtained as described above. Segmentation was performed on CD13 (for bile canaliculi) and F-actin (cell borders) staining with phalloidin. Analysis of bile canaliculus morphology and bile canaliculus network properties was performed using a custom-made Fiji script publicly available at https://git.mpi-cbg.de/huch_lab/assembloid-paper. A description of the script can be found in ref. 51 In brief, immunofluorescence images from several conditions were used in this analysis: EM2, DM and liver tissue, from hereon referred as 'structure'. We refer to individual bile canaliculus networks as 'network'. We determined the connectivity of the network by analysing the total number of branching points (number of triple junctions) per structure. We determined the length of the network per structure by analysing the total length of all branches in the structure. To compare structures in different conditions, we plotted these values as dot plots in which each dot was one structure. In the case of tissue, each dot was one field of view. The features extracted from Fiji were exported as .csv files and plotted using Prism.

For assembloids, to visualize the structure from different angles, immunofluorescence images were visualized in 3D using MotionTracking (http://motiontracking.mpi-cbg.de)[43]. For this, Gaussian blurring was applied to the channels of interest and then visualized in 3D.

For quantification of cholangiocytes and portal fibroblasts in assembloids, Arivis 4D software (Zeiss) was used. For the analysis, nuclei were segmented based on diameter, probability threshold and split sensitivity to align with the expected morphology in the fluorescence images. When segmentation was incomplete due to weak fluorescence signals, missing nuclei were manually added. This approach was used to determine the number of nuclei per cell and the number of cells per organoid. All segmentation results were manually reviewed and corrected as necessary.

### Isolation of mRNA and RT–qPCR analysis

RNA was extracted from organoid cultures or freshly isolated tissue using the RNeasy Mini RNA Extraction Kit (Qiagen) with DNase treatment and reverse-transcribed using Moloney murine leukaemia virus reverse transcriptase (Promega). All targets were amplified (40 cycles) using gene-specific primers (Key Resource Table) and PowerUp SYBR Green master mix (ThermoFisher) or iQ SYBR Green Supermix (Bio-Rad) and run on a qPCR instrument (Thermo Fisher QuantStudio 7 Pro or GeneAmp PCR System 9700; Applied Biosystems respectively). Data were analysed using Design & Analysis 2.7.0 software (ThermoFisher).

### Karyotyping

Mitotic metaphases for karyotyping were obtained by subculturing hepatocyte organoids in the active growth phase. The following day, cells were exposed to 0.2 μg ml$^{-1}$ colcemid (Gibco) for 60 min at 37 °C to arrest them in metaphase. Organoids were dissociated into single cells using TrypLE Express (Gibco). After centrifugation and removal of the supernatant, cells were subjected to hypotonic treatment with a solution of 0.075 M KCl for 30 min at 37 °C, followed by fixation in a 3:1 methanol to acetic acid solution. The preparation was washed three times with the fixative before slide preparation. Chromosomes were stained with Giemsa (Merck) diluted in Gurr buffer (pH 6.8; Gibco). Images were taken with a Zeiss Axio Imager.Z2 upright motorized stand with an ApoTome.2 for improved z contrast.

### Functional assays

For functional assays, h-HepOrgs were cultured in EM and DM media and assembloids in DM media as described above. As negative controls, we used h-CholOrgs grown as described above. As positive controls, we used freshly isolated PHHs cultured in standard 2D hepatocyte monolayer culture or in sandwich culture[65]. In brief, for the positive control of 2D hepatocytes, fresh isolated PHHs were plated onto collagen (1.8 mg ml$^{-1}$; RatCol collagen, Advanced Biomatrix)-coated 24-well plates at 500,000 or 250,000 cells per well in William's E medium (PAN Biotech) supplemented with 10% FBS, penicillin/streptomycin and 100 nM dexamethasone for 3 h for attachment. For the monolayer culture (1d-PHH monolayer control), the cells were cultured on William's E medium supplemented with 1% HEPES + 1% GlutaMAX + 1% penicillin/streptomycin and 100 nM dexamethasone for 18 h (or 24 h, for albumin assays) and then processed for the functional assays. For sandwich cultures, fresh isolated PHHs were plated onto collagen as above and overlayed with a second collagen layer (1.2 mg ml$^{-1}$; RatCol collagen, Advanced Biomatrix) and cultured for 7 days in William's E medium supplemented with CM4000 cell maintenance supplement (ThermoFisher Scientific).

To determine albumin secretion, supernatant from 24 h was collected and the amount of albumin was determined using a human-specific albumin ELISA kit (Assay Pro) following the manufacturer's instructions on an ELISA plate reader (Tecan Spark 20M). To measure cytochrome P450 activity, on the day of the experiment cholangiocyte and hepatocyte organoids in EM2 or DM were removed from Matrigel using Cell

Recovery solution (Corning). Organoids, 2D hepatocyte monolayers or 2D sandwich cultures were then all cultured in William's E medium supplemented with 1% HEPES + 1% GlutaMAX + 1% penicillin/streptomycin supplemented with luciferin-H substrate (100 µM) or luciferin-IPA (3 µM) for 6 h. Cytochrome activity was measured using the P450-Glo Assay Kit (Promega) according to the manufacturer's instructions on a plate reader (PerkinElmer Envision). Results were normalized to total viable cell counts per well.

## Urea synthesis assay
To determine urea secretion, cell culture supernatants were collected from 48-well plate after 12 h of culture. The concentration of secreted urea was measured by Urea Assay Kit (Abnova) according to the manufacturer's instructions.

## Measurement of gluconeogenesis
Gluconeogenesis was assessed using a Glucose-Glo Assay (Promega). Organoids/assembloids were first washed twice with PBS to remove residual glucose and then incubated for 24 h in glucose-free medium (Gibco) to deplete intracellular glucose stores. Subsequently, the organoids were stimulated for 24 h in gluconeogenesis-inducing medium (glucose-free medium supplemented with 10 mM lactate; Sigma-Aldrich, L7022) to promote hepatic glucose production.

After incubation, 25 µl of supernatant from each well was transferred to a 96-well assay plate and mixed with an equal volume of glucose detection reagent. Following incubation for 60 min at 37 °C, luminescence was measured using a luminometer.

## Cell counting
h-HepOrgs were dissociated into single cells using 10× TrypLE (Gibco) after 10 and 15 days of culture in specified media. Cell counts were determined using a Countess II FL automated cell counter (ThermoFisher Scientific).

## Quantification of xenobiotic metabolism by mass spectrometry
h-HepOrgs were cultured in DM as previously described. Assembloids were maintained under the same conditions for 6 days. Freshly isolated PHHs were cultured in a monolayer for 24 h, also as described above. Following culture, all cells were washed twice with PBS. The medium was then replaced with 100 µl of William's E medium supplemented with 1% HEPES, 1% GlutaMAX, 1% penicillin/streptomycin and verapamil (Merck) at a final concentration of 4 µM. Cells were incubated for 6 h, after which the supernatant was collected and analysed by mass spectrometry.

Organoids and assembloids were dissociated into single cells using 10× TrypLE and manually counted using a haemocytometer. The resulting cells were washed twice with PBS and stored at −20 °C.

Metabolites were separately extracted from the supernatant and from the cells by isopropanol:methanol:chloroform mixture (4:2:1, v/v/v) containing 7.5 mM ammonium formate (termed MS mix). A supernatant aliquot of 100 µl was diluted 20-fold (v/v) with MS mix, vortexed, centrifuged for 7 min at 13$g$ and the pellet was discarded. Cells suspended in 100 µl PBS were first lysed using ~25 stainless steel beads of 0.5 mm in size (Next Advance, USA, 152034) in the Qiagen Ratsch Tissue Lyser at 30 Hz for 8 min and metabolites were extracted as above. Each sample was prepared in three biological replicates and analysed by mass spectrometry immediately after extraction.

Mass spectrometry analysis was performed on a Q Exactive hybrid quadrupole Orbitrap tandem mass spectrometer (ThermoFisher Scientific) in positive ion mode by direct infusion of total extracts. Prior analyses, the internal standard verapamil-[13]C3 hydrochloride (Merck) was dissolved in methanol and spiked into samples to a final concentration of 200 nM. Aliquots of 40 µl of each sample were then placed on twin.tech PCR Plate 96 (Eppendorf, 0030128.648) and infused into the mass spectrometer via TriVersa NanoMate robotic ion source (Advion Interchim Scientific) using nanoflow chips with a nozzle diameter of 4.1 µm. The ion source was controlled using Chipsoft 8.1.0 software. Spraying voltage and gas back pressure were set to 1.25 kV and 0.95 psi, respectively. The ion transfer capillary temperature was set to 200 °C and the S-lens RF level was set to 50%. A target mass resolution ($R_{m/z}$) of 200 was set to 140,000 (full width at half maximum, FWHM) for both Fourier transform mass spectrometry (FT MS) and FT MS/MS spectra. To acquire FT MS spectra, the automated gain control (AGC) was set to $3 × 10^6$, the maximum injection time was set to 500 ms, the acquired mass range $m/z$ was 50–700, the lock masses $m/z$ 445.12003 and $m/z$ 338.34174. The acquisition cycle consisted of recording FT MS[1] spectra for 1.2 min followed by two FT MS/MS[2] spectra for 1.8 min from the precursors with $m/z$ 455.291 (for verapamil [M + H][+]) and $m/z$ 441.275 (for norverapamil [M + H][+]); precursor $m/z$ isolation width was 3 Th.

Spectra were averaged in Xcalibur Qual Browser v.3.0 (ThermoFisher Scientific) over a 30-s time range corresponding to stable spray; peaks of metabolites and standard extracted with 5 ppm mass accuracy. The absolute amount of norverapamil was calculated from its molecular ion intensity normalized to the intensity of the standard. For calibration, aliquots of William's E medium containing verapamil (Merck, V-002-1ML) with the concentration ranging from 2 µM to 8 nM were diluted 20-fold with MS mix, spiked with the internal standard and analysed as described above. The determined abundance of norverapamil in supernatant and in cellular pellets was summed up, normalized to $10^4$ cells and its production rate was expressed in pmol/h.

## Xenotransplantation in $Fah^{-/-}Rag2^{-/-}Il2rg^{-/-}$ (FRG) mice
Male and female $Fah^{-/-}Rag2^{-/-}Il2rg^{-/-}$ (FRG) mice were obtained from Jackson Laboratory. Mice were housed and maintained under specific-pathogen-free conditions in accordance with the principles of laboratory animal care and the guide set by the HYU Industry-University Cooperation Foundation. All animal experiments were conducted under protocols approved by the Institutional Animal Care and Use Committee (IACUC) of Hanyang University (2024-0148B). Experimental groups were not predetermined based on the sex of the mice, and all animals were randomly assigned to experimental procedures. Male mice accounted for approximately 25% of the total cohort. FRG mice 8–16 weeks old were used for all experiments. For their maintenance, mice were administered ad libitum NTBC (2-(2-nitro-4-trifluoromethylbenzoyl)-1,3-cyclohexanedione) in their drinking water.

Mice aged 8–16 weeks of both sexes were kept on NTBC in drinking water until 3 days before the experiment, when NTBC was withdrawn. h-HepOrgs expanded in EM2 and differentiated in DM were dissociated into single cells and prepared for injection. For transplantation experiments, commercially available frozen PHHs were used (F-PHH2; Supplementary Table 2). Organoids cultured under EM2 medium as well as isolated hepatocytes (PHHs) from the same donors were used as controls. Following dissociation, 500,000 dissociated organoid cells or 800,000 PHHs were resuspended in 100 µl AdDMEM/F-12 medium and injected into the spleen. The non-injected negative-control group received 100 µl PBS instead of cells. Mice were cycled in and out of NTBC treatment for 3 days every time their body weight dropped below 80% of the initial weight.

## IPA
We performed IPA (Qiagen) to identify potential candidate signalling pathways. For this, we first generated three DEG lists as DEGs between liver cancer organoids and liver healthy (list 1) or cancer (list 2) tissue (Supplementary Data 1_S1) and DEG list between partial hepatectomy and healthy tissue (list 3). Gene lists were generated as follows: lists 1 and 2: gene expression matrices from hepatocellular carcinoma (HCC)-derived organoids, HCC liver tissue and liver tissue from healthy donors were obtained from the Gene Expression Omnibus (GEO) under accession number GSE84073 (ref. 35). DEGs were identified using DESeq2 (ref. 2), applying a threshold of |log2 fold change| > 1 and an

adjusted *P* value of <0.1 (Supplementary Data 1_S1). For list 3, DEGs comparing partial hepatectomy and undamaged liver hepatocytes in mouse were sourced from the supplementary tables in ref. 27. Additionally, a list of genes mutated in both HCC-derived organoid lines was derived from the whole-exome sequencing (WES) results in ref. 35 (list 4). The full list of DEGs from lists 1–4 is provided in Supplementary Data 1_S1.

The three DEG lists and the mutated gene list (lists 1–4) were analysed using IPA, using the canonical pathway analysis and upstream regulator prediction functions (QIAGEN Inc., https://digitalinsights.qiagen.com/ipa). In brief, the significance of the association between the dataset and canonical pathways was determined using a right-tailed Fisher's exact test, followed by Benjamini–Hochberg correction for multiple testing. For analyses in which log fold changes were available, an activity *z* score was computed to predict the activation or inhibition likelihood of specific pathways base. Upstream regulator analysis used a computational algorithm to identify upstream regulators potentially responsible for the observed gene expression changes. From the IPA canonical pathway analysis, pathways were filtered based on an adjusted *P* value of <0.05 and the presence of the keyword 'signalling' in the pathway name (Supplementary Data 1_S2). Selected pathways of interest with a mean adjusted *P* value and frequency of pathway significance across comparisons are plotted in Extended Data Fig. 1c (Supplementary Data 1_S3). Activity *z* scores from the selected pathways were individually plotted as well as their corresponding mean values in Fig. 1b (Supplementary Data 1_S4,5). Next, results from the upstream regulator analysis were filtered for (1) an adjusted *P* value of <0.1 as upstream regulator and (2) the molecules from the two selected signalling pathways (Supplementary Data 1_S6). Key components of the signalling pathways and their adjusted *P* value in upstream regulator analysis are plotted in Extended Data Fig. 1d (Supplementary data 1_S7).

### Bulk RNA-seq library preparation
mRNA was isolated from on average 270 ng total RNA by poly(dT) enrichment using the NEBNext Poly(A) mRNA Magnetic Isolation Module (NEB) according to the manufacturer's instructions. Samples were then directly subjected to the workflow for strand-specific RNA-seq library preparation (Ultra II Directional RNA Library Prep, NEB). For ligation, NEB Next Adapter for Illumina of the NEB Next Multiplex Oligos for Illumina Kit was used. After ligation, adapters were depleted by XP bead purification (BeckmanCoulter) adding the bead solution in a ratio of 0.9:1 to the samples. Unique dual indexing was done during the following PCR enrichment (12 cycles) using amplification primers carrying the same sequence for i7 and i5 index (i5: AATGATACGGCGACCACCGAGATCTACACNNNNNNNNACATCTTTCCCTACACGACGCTCTTCCGATCT; i7: CAAGCAGAAGACGGCATACGAGATNNNNNNNNGTGACTGGAGTTCAGACGTGTGCTCTTCCGATCT). After two more XP bead purification steps (0.9:1), libraries were quantified using the Fragment Analyzer (Agilent). Libraries were sequenced on an Illumina NovaSeq 6000 in 100-bp paired-end mode to a depth of 40 million read pairs per library.

### RNA-seq data processing
Raw bulk RNA-seq data were processed using nf-core/rnaseq v3.18.0 (https://doi.org/10.5281/zenodo.1400710) of the nf-core collection of workflows[66], using reproducible software environments from the Bioconda[67] and Biocontainers[68] projects. The pipeline was executed with Nextflow (v24.10.5)[69]. The reference genome used was *Homo sapiens* GRCh38 (Ensembl release 111). The pipeline was run with custom parameters for trimming (extra_trimgalore_args: '--nextseq 20 --length 15'), alignment (extra_star_align_args: '--outFilterMismatchNmax 999 --outFilterMismatchNoverLmax 0.1 --alignMatesGapMax 200000 --chimSegmentMin 20 --twopassMode Basic --alignIntronMin 20 --alignIntronMax 200000') and quantification (extra_salmon_quant_args: '--seqBias --gcBias --posBias'). The resulting MultiQC report was inspected to ensure overall sequencing quality and pipeline performance.

Transcript-level abundance estimates were imported using the tximeta package[70] to generate a gene-level count matrix. Next, variance stabilizing transformation (VST) from DESeq2 (refs. 71,72) was used to normalize the data. Euclidean distance matrices, principal-component analysis (PCA) and heat map visualizations were computed on the VST-transformed values. On some heat maps, minimum–maximum scaling was applied. In Extended Data Fig. 2a,b, batch correction was performed on the VST-transformed values using limma's removeBatchEffect, with sample material type (tissue versus organoid) treated as the batch variable[73]. For differential expression analysis, DESeq2 was used. For comparison between MM + WntS + TRULI and primary (fresh isolated PHHs), the design formula ~ donor + condition_l3 was applied (Extended Data Fig. 2). Log-fold changes were shrunken using lfcShrink with the ashr method (type = 'ashr'), applying a fold-change threshold of 1.5 and a significance threshold of $\alpha$ = 0.05 (ref. 74). For the comparison between DM and EM2 (Fig. 2e), the design formula ~ batch + donor + condition_l1 was applied. Log-fold changes were shrunken using lfcShrink with the ashr method (type = 'ashr'), applying a fold-change threshold of 1.5 and a significance threshold of $\alpha$ = 0.05. For the comparison between h-HepOrgs and portal fibroblasts (Extended Data Fig. 6h), the design formula ~sex + cell_type was applied. Log-fold changes were shrunken using lfcShrink with the ashr method (type = 'ashr'), applying a fold-change threshold of 4 and a significance threshold of $\alpha$ = 0.05. Gene set enrichment analysis (GSEA) was conducted using the clusterProfiler package, leveraging gseKEGG, gseGO and gsePathway for pathway enrichment analysis[75].

The zonated gene list (Extended Data Fig. 3h) was obtained by manually curating genes that have been confirmed to be portally or centrally zonated from human spatial transcriptomic datasets[10,11,46,47] (a full list is provided in Supplementary Data 2_S6). We then intersected this refined zonated gene list with our list of differentially expressed genes in the DM versus EM2 comparison.

Donor-specific genes were identified separately for batches Y1/Y2 and S1/S2 using a likelihood ratio test (LRT) with the full model ~donor and the reduced model ~1. Genes with an adjusted *P* value of <0.05 were retained, and the resulting gene lists from the two batches were merged. Pairwise correlations between organoids and primary cells were computed using the donor-specific genes. For the heat map shown in Extended Data Fig. 4e, sex-specific genes were excluded.

The complete software stack for downstream analysis is available as a Docker container (rnaseq-notebook:2025-04-21) archived at https://quay.io/repository/fbnrst/rnaseq-notebook and archived on Zenodo (https://doi.org/10.5281/zenodo.17704466).

### Single-cell transcriptomics with 10x Genomics
For scRNA-seq analysis, assembloids were generated by assembling h-HepOrgs, cholangiocytes/ductal cells derived from cholangiocyte organoids (n-GFP) and portal fibroblasts (n-RFP) at a ratio of 1 h-HepOrg to 25 portal fibroblasts and 100 cholangiocytes. At 5–6 days after aggregation, assembloids were collected as follows: periportal assembloids were dissociated to single cells using 10× TrypLE for 5 min at 37 °C. The cells were resuspended in DM and 10 µg ml⁻¹ DNase in BSA-coated tubes and filtered through a 100-µm strainer. Cell suspensions (30,000–50,000 cells) were concentrated by centrifugation (50*g*, 5 min, 4 °C) and the volume was reduced to ~55 µl. Cells were carefully resuspended and visually inspected under a light microscope to determine cell concentration and quality. The concentrations of the single-cell suspensions were adjusted to 138–912 cells per microliter and carefully mixed with the reverse transcription mix before loading cells on the 10x Genomics Chromium system[76] in a Chromium Single-Cell G Chip targeting 3,000–10,000 cells per reaction. Following the guidelines of the 10x Genomics Chromium Single-Cell Kit v3.1 user manual, the droplets were directly subjected to reverse transcription, the emulsion was broken and cDNA was purified using Dynabeads MyOne Silane (10x Genomics). cDNA was first amplified with 12 cycles, and then purified with

0.6× SPRIselect beads (BeckmanCoulter) to enrich cDNA fragments (>400 bp). A quality and quantity control of cDNA on the Fragment Analyzer (using the DNF-473 NGS Fragment Kit, Agilent) was eventually performed to obtain its concentration. The 10x Genomics scRNA-seq library preparation–involving fragmentation, dA tailing, adapter ligation and 11 or 12 cycles of indexing PCR, was performed based on the manufacturer's protocol. After quantification, the libraries were sequenced on an Illumina NovaSeq 6000 in paired-end mode (R1/R2, 100 cycles; I1/I2, 10 cycles), generating 230–370 million fragment pairs.

## scRNA-seq data analysis

The raw scRNA-seq data were processed using nf-core/scrnaseq v3.0.0 (https://doi.org/10.5281/zenodo.3568187) of the nf-core collection of workflows[66], using reproducible software environments from the Bioconda[67] and Biocontainers[68] projects. The pipeline was executed with Nextflow (v24.10.5)[69]. STARSOLO was used as the aligner. The reference genome was set to *Homo sapiens* GRCh38 (Ensembl release 111) with custom additions for RFP and GFP transgenes, obtained from SnapGene (DsRed1 and EGFP, respectively). Outputs were inspected for quality control, and one sample with poor quality control was excluded from further analysis. Within nf-core/scrnaseq, technical artefacts were eliminated using CellBender[77]. The CellBender output was used for data visualization. Doublet detection for each sample was performed using scrublet[78].

Further analysis was performed using scanpy[79]. Quality control was applied with the following thresholds: minimum total counts of 5,000, minimum detected genes of 2,000, the maximum percentage of counts in the top 50 genes set at 50%, the maximum percentage of mitochondrial counts set at 15% and a maximum doublet score of 0.15. Gene filtering was performed to retain genes expressed in at least ten cells. After filtering, the data underwent normalization, log transformation and identification of the top 3,000 most highly variable genes. PCA was performed, and batch correction was implemented through Harmony integration[80]. UMAP visualization and Leiden clustering were used to identify the three expected cell types[81,82].

To compare homeostatic-like and fibrotic-like organoids, pseudobulk aggregation was performed using decoupleR for each cell type[83]. Pseudobulk data were generated by summing raw counts for each sample and cell type, with a minimum requirement of ten cells per group and 1,000 total counts. Differential expression analysis was conducted using pyDESeq2 (ref. 84). For each cell type, DESeq2 datasets were created with design factors that included 'donor' and 'condition', using the 'homeostatic-like' condition as the reference. Differentially expressed genes between the homeostatic-like and fibrotic-like conditions were ranked on the basis of the test statistic. Subsequently, gene set enrichment analysis (GSEA) was performed on the ranked lists using clusterProfiler, focusing on KEGG, Reactome and GO terms.

The complete software stack for downstream analysis is available as a Docker container (singlecell-notebook:2025-04-21) archived at https://quay.io/repository/fbnrst/singlecell-notebook and archived on Zenodo (https://doi.org/10.5281/zenodo.17704461).

## Comparison to public datasets

Data from refs. 12,13,49 were downloaded in h5ad format from https://cellxgene.cziscience.com/. Additionally, data from ref. 11 were obtained from https://data.mendeley.com/datasets/yp3txzw64c/1, and the dataset from ref. 8 was downloaded from https://datashare.ed.ac.uk/bitstream/handle/10283/3433/tissue.rdata and converted to h5ad format using the sceasy package.

These public datasets were merged with the raw count matrix of our quality control-filtered organoid data. Subsequently, the combined dataset underwent normalization, followed by log transformation and detection of the top 4,000 most highly variable genes. We performed PCA and integrated the dataset using Harmony, specifying concatenation of the paper and donor as batch variables, with a maximum of 20

iterations and a theta value of 1.5. Selected genes were visualized in a dot plot (Fig. 4h).

Pseudobulk analyses were then conducted using the decoupleR package to summarize gene expression by cell type. This involved generating a pseudobulk dataset in which raw counts were summed by sample and cell type, ensuring a minimum of 30 cells per group. Following pseudobulk aggregation, the data were normalized and log transformed, with the top most highly variable genes identified on the basis of mean expression and dispersion. Additionally, the 'paper' variable was regressed out to mitigate batch effects. Next, PCA was performed on the pseudobulk data, using 50 principal components for subsequent analyses. Hierarchical clustering was executed using the Pearson correlation metric, and Pearson correlation matrices were plotted, as shown in Figs. 4g and 5f.

Marker genes for the three major cell types were computed separately for our organoid data and the merged public data using scanpy's rank_genes_groups function. For each dataset, the top 300 marker genes for each cell type were selected. Subsequently, GSEA was performed using the gseapy package, leveraging the Enrichr method[85]. The analysis focused on the KEGG 2021 Human and Reactome 2022 gene sets, with a *P*-value cut-off of 0.05. Shared enriched pathways between the organoid and tissue datasets were identified, and the combined enrichment scores for selected terms were plotted (Extended Data Fig. 8c,d).

## Data statistical analysis

The specific statistical test is specified in the legend. $P < 0.05$ was considered statistically significant. In all cases, data from at least three independent experiments were used. Calculations were performed using the Prism 9 software package. All *P* values are given in the corresponding figure legends or in the corresponding figure or in the corresponding source data file. Dispersion and precision measures (for example, mean, median, s.d., s.e.m.) are specified in the figure legends. No statistical methods were used to predetermine sample size. All scRNA-seq statistics are described above in the corresponding section.

## Reporting summary

Further information on research design is available in the Nature Portfolio Reporting Summary linked to this article.

## Data availability

Ethical approval for the generation of organoid lines was granted under the condition of restricted access. Organoid lines may be obtained upon request to the corresponding author, signing of a project-specific material transfer agreement and receiving ethical approval covering the planned work. Approval will be granted for work covered by the consent granted by patients. Depending on the requested organoid line, the ethical approval will be handled by the institutional review boards of the University Hospital Carl Gustav Carus Dresden or the University of Leipzig, according to where the patient was consented. The raw scRNA-seq (EGAD50000001453) and bulk RNA-seq (EGAD50000001454) data (FASTQ files) generated in this study have been deposited in the European Genome-phenome Archive (EGA) under accession number EGAS50000000994. Following EGA regulations, access to these data is controlled to protect the privacy and identity of study participants. Access requests can be submitted via the EGA website. Requests will be evaluated by the Data Access Committee (DAC) EGAC50000000112 to ensure that the proposed data use is consistent with the consent provided by participants. Approved users will be required to sign a data access agreement (DAA) that specifies the permitted uses of the data. A template of the DAA is available at https://edmond.mpg.de/api/access/datafile/250610. Applications are normally reviewed within 10 working days.

Count matrices and fully processed data, together with the source code for the sequencing data analysis, are publicly available via Zenodo (https://doi.org/10.5281/zenodo.17251198)[86].

Comprehensive lists of DEGs, GSEA terms, and marker genes are provided in Supplementary Data 2–4.

All other images as well as qPCR and measurement data are presented in the manuscript, and data used to plot the graphs are provided as supplementary information with this paper.

The URLs used to generate the UMAP plots from the data in ref. 8 for Extended Data Fig. 5 and the histology images from the Human Protein Atlas database for Extended Data Fig. 5 are all provided in the Extended Data Fig. 5 legend.

The raw data from mass spectrometry analyses are publicly available at the following repository: https://doi.org/10.17617/3.Z9GMJE. Source data are provided with this paper.

## Code availability

The source code for bulk RNA-seq and scRNA-seq data analysis is available at https://git.mpi-cbg.de/huch_lab/yuan_dawka_kim_liebert_et_al_2025_sequencing (https://doi.org/10.5281/zenodo.17251198)[86].

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

**Acknowledgements** M.H. is supported by the Max Planck Gesellschaft and is the recipient of an Allen Distinguished Investigator Award, a Paul G. Allen Frontiers Group advised grant of the Paul G. Allen Family Foundation, which supported A.L. and R.R.d.C. This project was partially supported by the European Research Council under the European Union's Horizon Europe research and innovation programme (grant agreement no. 101088869) awarded to M.H. Views and opinions expressed are however those of the authors only and do not necessarily reflect those of the European Union or the European Research Council. Neither the European Union nor the granting authority can be held responsible for them. Part of this work was also funded by the LiSyM grant from the Bundesministerium für Bildung und Forschung (BMBF Federal Ministry of Education and Research) awarded to M.H. (031L0258C and 031L0315B) and to G.D. and D.S. (031L0258E and 031L0315D). This project was also partially supported by the Deutsche Forschungsgemeinschaft (DFG, German Research Foundation, 514150034 and 513752256). D.L.H.T. was supported by a Croucher Foundation fellowship. D.E.S. was supported by a German Cancer Aid Max Eder Grant (70113745). We thank F. Ahmed for help with chromosome counting. We thank J. Jarrells and J. Hernandez (MPI-CBG) for assistance with fluorescence-activated cell sorting (FACS), the light microscopy facility for imaging troubleshooting and training (J. Peychl and R. Maraspini), the Technology Development Studio facility for the high-throughput imaging and image analysis (R. Barsacchi and M. Stöter) and the Dresden Concept Genome Center for the RNA-seq and scRNA-seq library (S. Reinhardt and J. Bläsche at the DcGC Dresden-concept Genome Center, a core facility of the CMCB and Technology Platform of the TUD (Technische Universität Dresden). We thank J. Pöche for assistance with hepatocyte isolation and the surgical research laboratory and especially S. Hübner as well as the operative team of the Department of Visceral, Thoracic and Vascular Surgery, University Hospital Dresden, for assistance in liver tissue processing. We thank the whole team of the Department of Hepatobiliary Surgery and Visceral Transplantation, Leipzig University Medical Center, for their support in patient acquisition and liver tissue logistics. We thank G. Schicht for her assistance in the hepatocyte isolations at Leipzig University Medical Center. We thank M. Marass for insightful comments and discussions on the manuscript.

**Author contributions** M.H. designed the study. L.Y., S.D., Y.K., A.L. and R.A.-B. performed most of the experiments and, together with M.H., interpreted the results. F.R. performed the scRNA-seq analysis. F.R. and D.L.H.T. performed the bulk RNA-seq analysis. R.R.d.C. performed the chromosome analysis. A. Schumann, Anna Shevchenko and Andrej Shevchenko applied direct-infusion mass spectrometry to characterize drug metabolism capacity. F.B., D.E.S. and C.G. assisted with tissue processing and hepatocyte isolation. D.S., G.D. and D.E.S. obtained patient consent for the tissue samples used in the study. S.D. performed image analysis and, with A. Sljukic, the bile canaliculus reconstructions. A.M.D. and A. Sljukic contributed to the first phases of the project. Y.K. and D.C., with assistance from S.K., conducted the xenotransplantation experiments. L.Y., S.D., Y.K., A.L., F.R. and M.H. wrote the manuscript. All authors read and commented on the manuscript.

**Funding** Open access funding provided by Max Planck Society.

**Competing interests** The Max Planck Institute of Molecular Cell Biology and Genetics is an applicant for a patent on the hepatocyte organoids and assembloids described in this work (EP23207475 and EP25169082.2). M.H. is an inventor on several patents on organoid technology. Y.K., S.D., R.A.-B. and M.H. are inventors on a patent on human hepatocyte organoids (EP23207475). M.H., L.Y., S.D., A.M.D. and A. Sljukic are co-inventors on a patent on human assembloids (EP25169082.2). The remaining authors declare no competing interests.

**Additional information**
**Correspondence and requests for materials** should be addressed to Meritxell Huch.

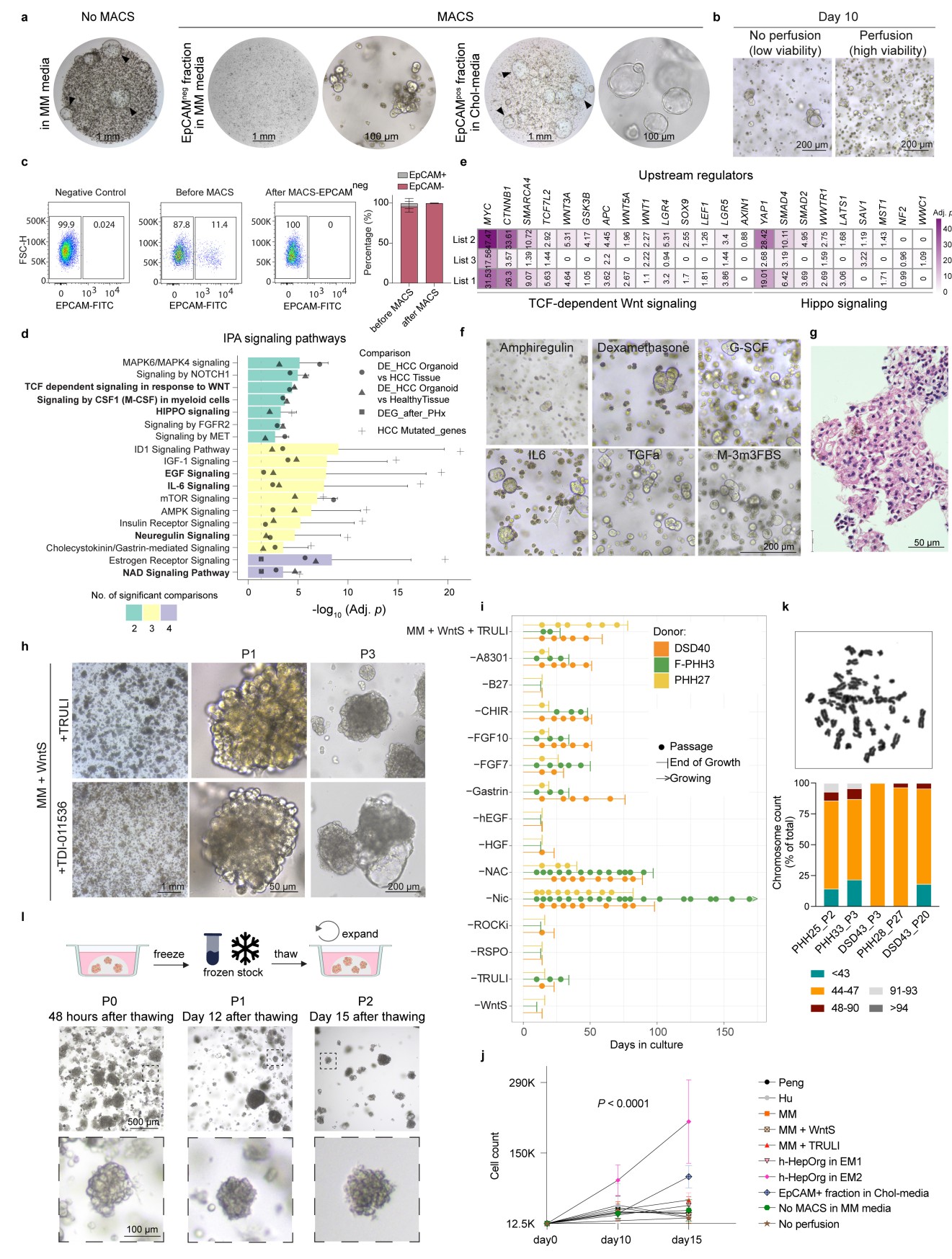

**Extended Data Fig. 1** | See next page for caption.

**Extended Data Fig. 1 | Establishment of the culture conditions to expand human hepatocyte organoids long-term while retaining their genetic stability over time. a-j**, Refinement of the isolation procedure and media conditions for expanding patient-derived human hepatocyte organoids. To remove cholangiocytes from the digested prep, the perfused digested tissue was processed for EpCAM-negative selection by MACS (**a-c**). To identify culture conditions, Ingenuity Pathway analysis was performed (**d-e**) and several signalling pathways found differentially expressed were screened for hepatocyte organoid growth (**f-j**). **a**, Representative brightfield images from n > 10 different experiments showing primary human hepatocytes (PHHs) seeded directly after isolation in MM medium (no-MACS, left) or after exclusion of EpCAM-positive cholangiocytes by MACS (right). The EpCAM-negative fraction, enriched for PHHs was seeded in MM (EpCAM$^{neg}$ fraction), while the EpCAM-positive fraction, enriched in human cholangiocytes, was seeded in Chol-media (EpCAM$^{pos}$ fraction). See Methods and Supplementary Table 1 for details. Scale bar, 1 mm (left), 100 μm (right). **b**, Representative brightfield images of seeded PHHs with either high or low viability after isolation following the indicated procedure. Scale bar, 200 μm. **c**, Efficiency of cholangiocyte removal following EpCAM$^+$ MACS-based separation. Flow cytometry analysis of EpCAM$^+$ cells on the liver cell preparation before MACS and on the EpCAM neg fraction after MACS. The absence of EpCAM$^+$ cells in the 'after-MACS EpCAM-neg fraction' confirms the efficient depletion of cholangiocytes from the liver cell preparation. Graph presents the quantification of EpCAM$^-$ cell percentages before and after MACS. Data are represented as mean ± SD from n = 3 donors, from n = 3 independent experiments. **d**, Ingenuity Pathway Analysis (IPA) was performed on n = 4 comparisons using datasets from Broutier et al.[35] and Hu et al.[27] (Supplementary Dataset 1_S1). Shown are pathways enriched in ≥2 comparisons (adj. p < 0.05, grey line). Bars represent mean −log10 (adj. p), error bars the SD; bar colour indicates the number of significant comparisons (Supplementary Datasets 1_S2–S3). **e**, IPA upstream regulator analysis for the different datasets used (Lists 1–3, see Methods). The heatmap plots selected members from the 2 indicated signalling pathways. Gradient colour bar, -log (adjusted p-value) are plotted for prediction of the upstream regulators from different DEG comparison list (Supplementary Dataset 1_S7). **f**, Screen for activation of a subset of signalling pathways identified in 'd' using the indicated components. Representative brightfield images are shown. Scale bar, 200 μm. **g**, H&E staining of h-HepOrgs grown in h-HepOrgs-EM2 medium shows that the h-HepOrgs grow as solid structures in vitro. Representative images from n = 3 independent experiments are shown. **h**, h-HepOrgs cultured in the presence of Wnt-ligand (WntS) and either TRULI or TDI-011536 show lumen formation after several passages under TDI-011536 treatment. Representative images of n = 3 independent experiments are shown. Scale bar, 1 mm (left), 50 μm (middle), 200 μm (right). P, passage. **i**, Graph showing the expansion potential of h-HepOrgs from multiple donors in EM1 (complete) and in EM1 with the removal of individual components as indicated. Note that removal of Nic increased the longevity of the cultures for all the donors tested. This medium is subsequently called h-HepOrgs-EM2. Dot, passage. **j**, Comparison of expansion potential between h-HepOrgs grown in h-HepOrgs-EM2 medium and other media conditions. Following hepatocyte isolation, 12,500 cells were seeded in Matrigel and cultured in our h-HepOrgs-EM2 or EM1 medium or in the medium published by Peng et al.[28], or Hu et al.[27], or in MM medium supplemented or not with Wnt or TRULI. Liver cell preparations obtained from the same perfused tissue but not-MACS as well as from the same tissue but not perfused (no perfusion) were also analysed. EpCAM$^+$ cholangiocyte fraction cultured in our cholangiocyte medium was used as control. Total cell count was measured at the indicated time points. Graph presents the mean ± SEM of n = 3 different donors with n = 3 technical replicates from n = 3 independent experiments. Statistical significance was determined by two-way ANOVA with Dunnett's test for multiple comparisons, comparing each condition to the EM2 group at the last time point (day15); p < 0.0001 for all comparisons. The colour coding indicates the comparison of our h-HepOrgsEM2 medium to the respective medium in the legend. **k**, Chromosome analysis of h-HepOrgs expanded for long or short time in culture. Representative image of a chromosome spread is shown (top). Graph shows the number of chromosomes from h-HepOrgs-EM2 at different passages from multiple donors indicating the maintenance of genetic stability over time. P, passage. **l**, Cryopreserved h-HepOrgs grown in EM2 can be recovered from cryopreservation without exhibiting any signs of loss of expansion potential. Representative brightfield images of h-HepOrgs after freezing and thawing are shown. Scale bar, 500 μm (top), 100 μm (bottom). P, passage.

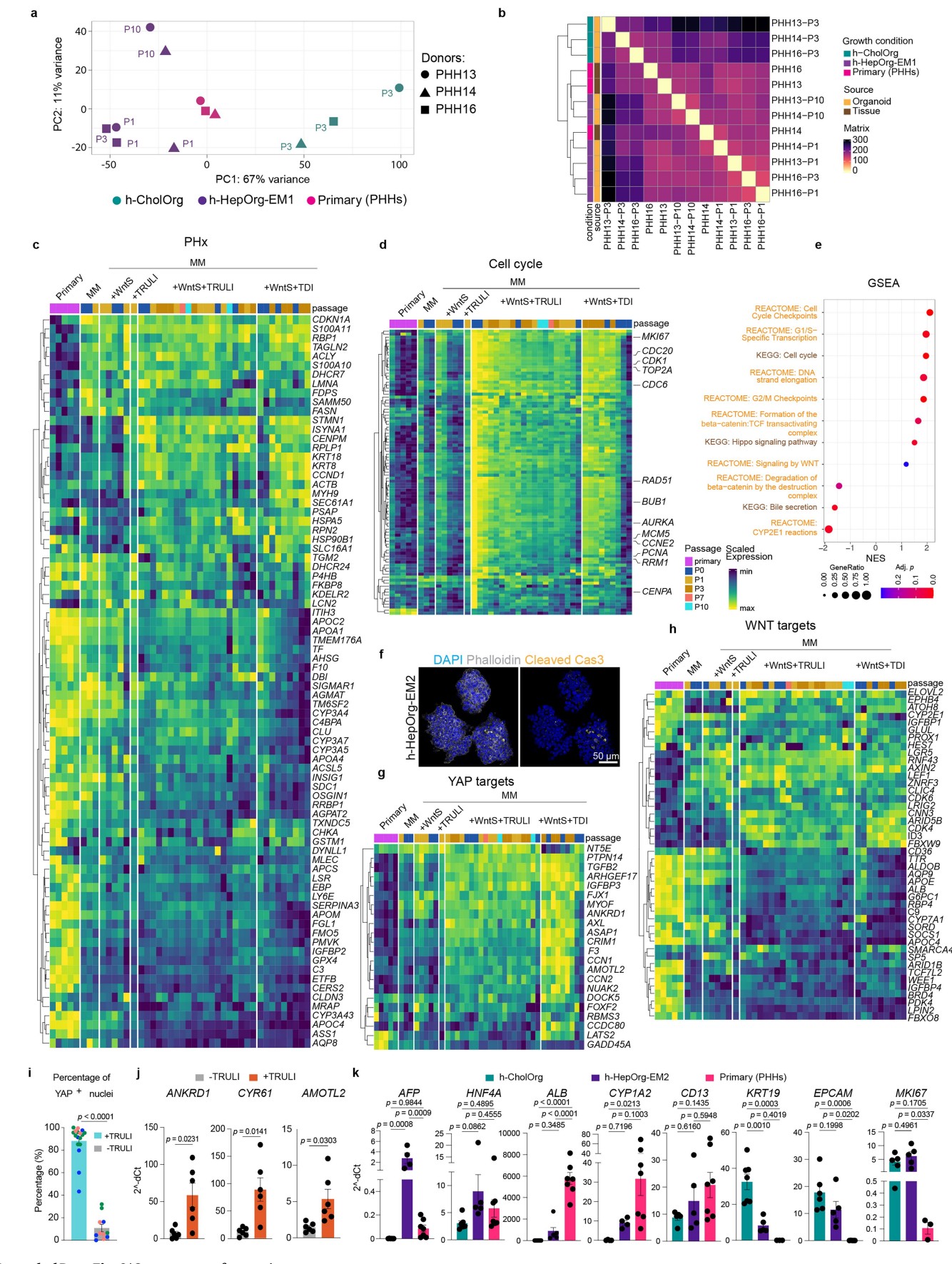

**Extended Data Fig. 2** | See next page for caption.

**Extended Data Fig. 2 | Serially expanded human hepatocyte organoids exhibit gene expression programs of regenerating livers. a-h,** h-HepOrgs organoids were cultured for several passages (from P1 to P10) in the presence of Wnt-ligand (WntS) and LATS1/2 inhibitor (TRULI or TDI-011536). At the indicated time point, serially expanded cultures were collected and processed for either RNA (**a-e, g-h, j-k**) or immunofluorescence (**f**) analysis. **a-b,** PCA (**a**) and correlation analysis (**b**) for fresh isolated human primary hepatocytes (primary, pink) and cholangiocyte organoids (h-CholOrg, green) and h-HepOrgs (purple) at early (P1) and late (P10) passage. Both, PCA and correlation analysis were batch corrected for tissue and culture. **c,** Heatmap showing the expression of the most DEG in regenerating mouse livers after partial hepatectomy (PHx) in h-HepOrgs under specific culture conditions, as well as in fresh isolated hepatocytes (primary). The top 100 DEG from Hu et al.[27] were used from which 89 direct orthologs as annotated by ENSMBL were identified and are displayed. Column, donor. Passage number is indicated with colour according to the legend in **d. d,** Heatmap displaying the expression of the top 100 cell cycle genes described in *Tirosh* et al.[87], in h-HepOrgs under specific culture conditions, as well as in fresh isolated hepatocytes (primary). Column, donor. Passage number is indicated with colour. **e,** Selected list of gene sets significantly enriched in h-HepOrgs under MM+WntS+TRULI compared to fresh isolated hepatocytes. The full list is presented in Supplementary Dataset 2. The results are presented as a dot plot, where dot colour represents the adjusted *p*-value (permutation test implemented in clusterProfiler, adjusted using the Benjamini–Hochberg method), and dot size corresponds to the GeneRatio (number of core enrichment genes divided by the total number of genes in the pathway). NES, normalized enrichment score. Brown, gene sets enriched in KEGG database. Orange, gene sets enriched in Reactome database.

**f,** Immunofluorescent staining for cleaved caspase 3 (yellow), F-actin (Phalloidin, white) and DAPI (nuclei, blue) in h-HepOrgs serially expanded under h-HepOrgs-EM2 medium. n = 3 independent experiments. Scale bar, 50 µm. **g,** Heatmap displaying the expression of YAP targets from Wang et al.[88], in h-HepOrgs under specific culture conditions, as well as in fresh isolated hepatocytes (primary). Column, donor. Passage number is indicated with colour. **h,** Heatmap displaying the expression of liver specific WNT targets according to the TCF4 ChIP seq dataset described in Boj et al.[89], in h-HepOrgs under specific culture conditions, as well as in fresh isolated hepatocytes (primary). Column, donor. Passage number is indicated with colour according to the legend in **g. i,** Graph showing mean ± SEM of the percentage of YAP-positive and YAP-negative nuclei in h-HepOrgs cultured in presence of TRULI (+TRULI, EM2 condition) or after TRULI removal for 1 week (-TRULI). Each dot represents one organoid, colours indicate different donors (n = 3 donors). Ordinary two-way ANOVA (factors: condition and donor) was used for statistical analysis, p < 0.0001. **j,** Quantitative RT-PCR analysis of the YAP downstream target genes *ANKRD1*, *CYR61*, and *AMOTL2* in cultures treated for 12 h with TRULI (+TRULI, orange) or without TRULI (−TRULI, grey). Gene expression levels are normalized to *HPRT* and presented as 2^−ΔCt. Data are shown as mean ± SEM (n = 6 donors). Significance was assessed using an unpaired two-tailed Student's t-test. *p*-values are indicated. **k,** qPCR analysis of the indicated genes in human cholangiocyte organoids (h-CholOrg, green) and h-HepOrgs cultured in EM2 medium (purple) and fresh isolated hepatocytes (Primary PHHs, pink). Gene expression levels are normalized to *HPRT* and presented as 2^−ΔCt. Graph represents the mean ± SEM of n ≥ 3 independent donors, Dot, donor. Significance was assessed using one-way ANOVA followed by Tukey's multiple comparison test, *p*-values are indicated on the graph.

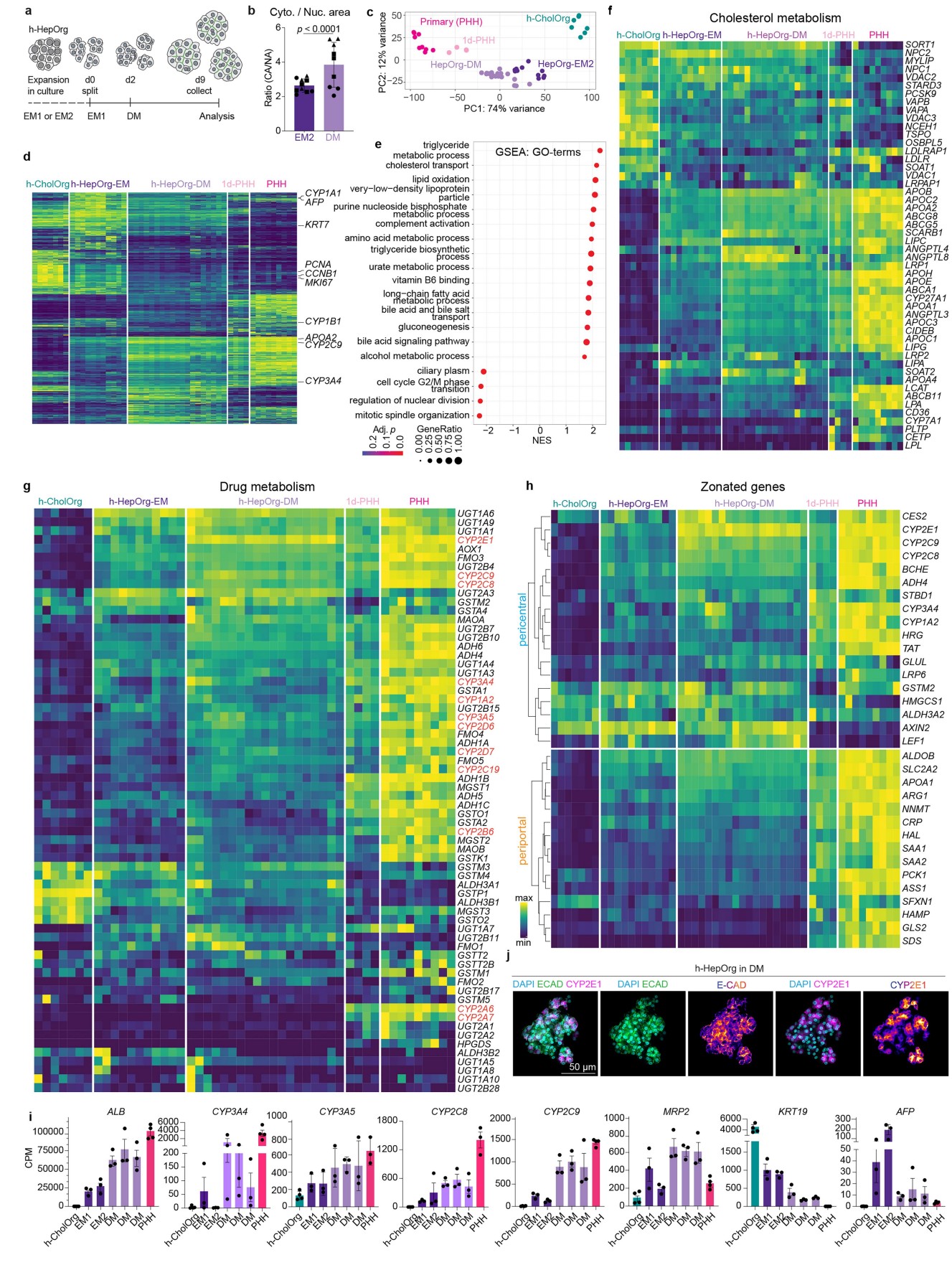

**Extended Data Fig. 3** | See next page for caption.

**Extended Data Fig. 3 | Differentiated human hepatocyte organoids retain gene expression of mature in vivo hepatocytes. a**, Experimental design. **b**, Graph showing the mean ± SD of cytoplasmic to nuclear ratio measured as area (CA/NA) in h-HepOrgs in EM2 and in DM. Dot, individual organoid. Shape, independent donor (n = 3 donors). Ordinary two-way ANOVA (factors: medium and donor) was used for statistical analysis, p < 0.0001. **c-i**, h-HepOrgs organoids were cultured for several passages in h-HepOrgs-EM2 medium and then differentiated in DM as indicated in methods. At day 7 of the differentiation protocol the organoids were harvested and processed for RNAseq analysis. **c**, PCA analysis showing the PC1 and PC2 components from the RNAseq analysis from fresh isolated primary hepatocytes (PHHs, pink), hepatocytes cultured for 1 day in standard monolayer culture (1d-PHH monolayer, light pink), cholangiocyte organoids (h-CholOrg, green) and h-HepOrgs in EM2 (dark purple) or DM (light purple). **d**, Heatmap showing all DEG between h-HepOrgs grown in DM compared to EM2 medium, and their expression in fresh isolated hepatocytes (Primary, PHHs), hepatocytes cultured for 1 day in standard monolayer culture (1d-PHH monolayer) and in cholangiocyte organoids (h-CholOrg). Note the similarity between DM, 1d-PHHs and primary hepatocytes. Column, donor. **e**, Selected list of gene sets from the GO-terms database that are significantly enriched in h-HepOrgs under DM compared to EM2. The full list is presented in Supplementary Dataset 2. Results are presented as dot plot, where dot colour represents the adjusted *p*-value (permutation test implemented in clusterProfiler, adjusted using the Benjamini–Hochberg method), and dot size corresponds to the GeneRatio (number of core enrichment genes divided by the total number of genes in the pathway). NES, normalized enrichment score. Note that many of positively enriched gene sets are related to classical hepatocyte functions such as lipid and drug metabolism and bile secretion and transport. Conversely, negatively enriched gene sets are related to cell cycle. **f-g**, Heatmap showing the expression of the full list of genes from the KEGG cholesterol dataset (**e**) and drug metabolism (**f**) datasets from the GSEA analysis between h-HepOrgs grown in DM compared to EM2 medium, and their expression in fresh isolated hepatocytes (Primary, PHHs), hepatocytes cultured for 1 day in standard monolayer culture (1d-PHH monolayer, light pink) and in cholangiocyte organoids (h-CholOrg). Note the similarity between DM, 1d-PHHs, and primary hepatocytes. Column, donor. **h**, Heatmap showing the expression of DEG between DM and EM2 that intersect with the gene list of pericentrally or periportally zonated genes 1. Details are found in Supplementary Table 2. **i**, CPM values from the RNAseq for the hepatocyte marker Albumin (*ALB*), the mature hepatocyte marker and metabolizing enzymes *CYP3A4, CYP3A5, CYP2C8, CYP2C9*, the bile salt transporter *MRP2*, the cholangiocyte marker *KRT19* and the hepatoblast marker *AFP*. Note that upon DM the expression of mature hepatocyte markers increases while the expression of the cholangiocyte and foetal marker decreases. Bars represent mean ± SEM for n = 4 (h-CholOrg) or n = 3 (all others) biologically independent samples (donors). **j**, h-HepOrgs in DM stained for the pericentral marker CYP2E1 (magenta) and periportally enriched marker ECAD (green) (n = 3 donors). CYP2E1 and ECAD shown in Fire LUT for enhanced visualisation. Nuclei were stained with DAPI (cyan). Scale bars, 50 μm. Panel **a** adapted from ref. 51, Springer Nature Limited.

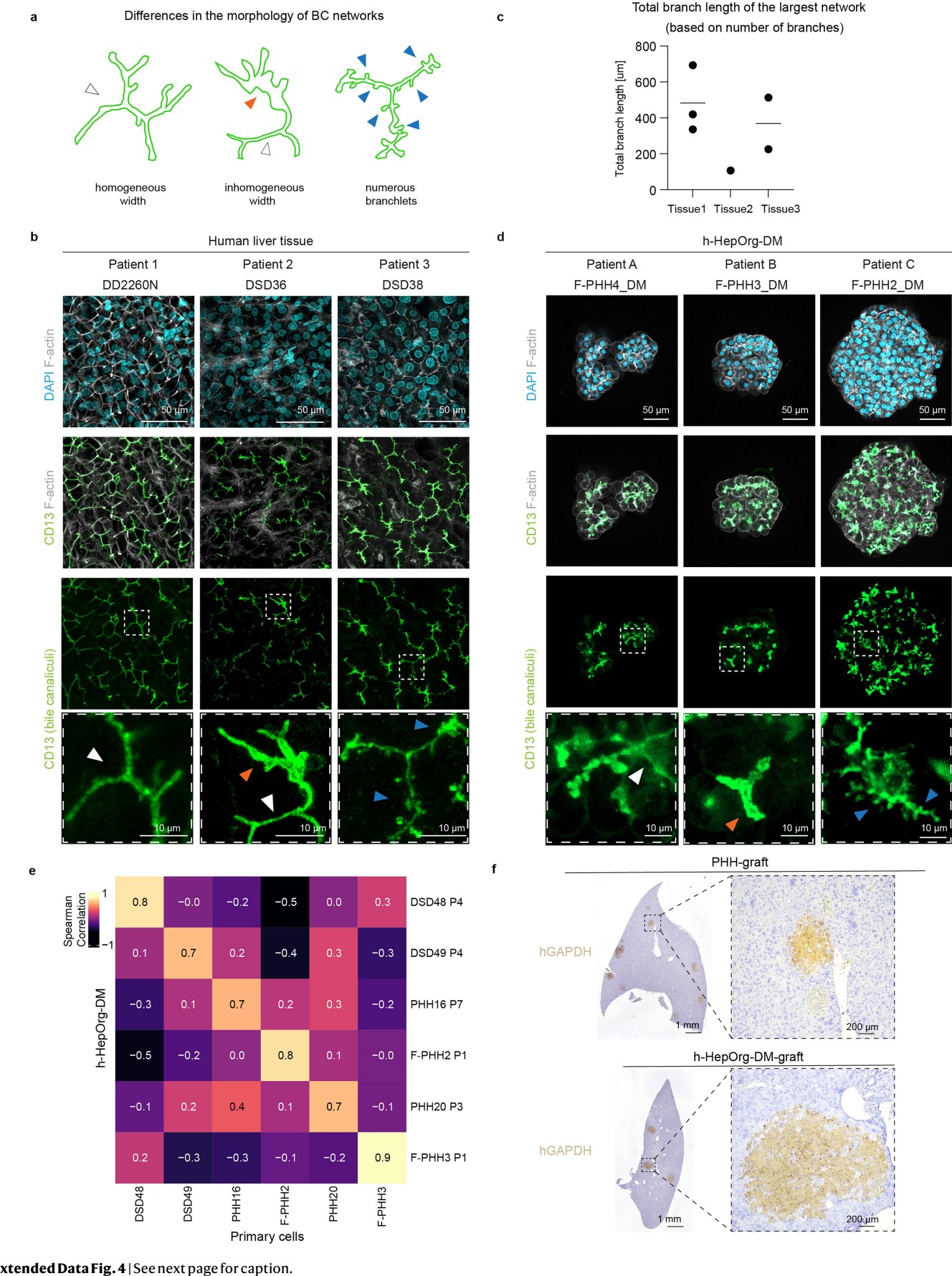

**Extended Data Fig. 4 |** See next page for caption.

**Extended Data Fig. 4 | h-HepOrgs present bile canaliculi architectural heterogeneity similar to patient cohorts and capture patient-specific gene expression programs. a**, Illustration showing the heterogeneity in the BC morphologies observed. Compare the illustration to the bottom magnified panels in figures in '**b**' and '**d**'. Note that both, tissues and organoids from different patients present fine detailed differences in BC morphology, with some being thin and homogenous (white arrowhead), others wider and inhomogeneous (orange arrowheads) and some full of branchlets (blue arrowheads). **b**, Immunofluorescent images for bile canaliculi (CD13, green), cell borders (F-actin, grey) and nuclei (DAPI, cyan) (top) showing variation in bile canaliculi (BC) networks between tissue samples from different donors (n = 3). White arrowheads, thin BC; orange arrowheads, wide BC; blue arrowheads, BC with branchlets. Scale bar, 50 μm (top three panels); 10 μm (bottom). **c**, Graph showing the differences in the total bile canaliculi branch length for the largest network (network with the maximum number of branches) between the different tissue samples from different donors (n = 3 donors). Each dot represents one field of view. Refers to data from Fig. 3c. **d**, Immunofluorescent images for bile canaliculi (CD13, green), cell borders (F-actin, grey) and nuclei (DAPI, cyan) (top) showing variation in bile canaliculi (BC) networks between h-HepOrgs from different donors cultured in DM (n = 3 donors). White arrowheads, thin BC; orange arrowheads, wide BC; blue arrowheads, BC with branchlets. Scale bar, 50 μm (top three panels); 10 μm (bottom). **e**, Spearman correlation heatmap showing pairwise similarities between primary hepatocyte samples and differentiated hepatocyte organoids. Donor-specific genes were first identified from fresh primary hepatocytes, and these genes were then used to calculate pairwise correlations with gene expression profiles from organoids. Warmer colours indicate higher correlation coefficients, representing stronger transcriptomic similarity. P, passage. **f**, Human GAPDH immunohistochemical results of a representative liver section of h-HepOrgs-DM (bottom) and fresh isolated primary human hepatocytes (PHH, top) transplanted in $Fah^{-/-}Rag2^{-/-}Il2rg^{-/-}$ (FRG) mice at day 90 and 92 after transplantation, respectively. Representative images from X number of independent images from n = 2 mouse (HepOrg-DM) and n = 1 mouse (PHH). Scale bar, 1 mm (left); 200 μm (right).

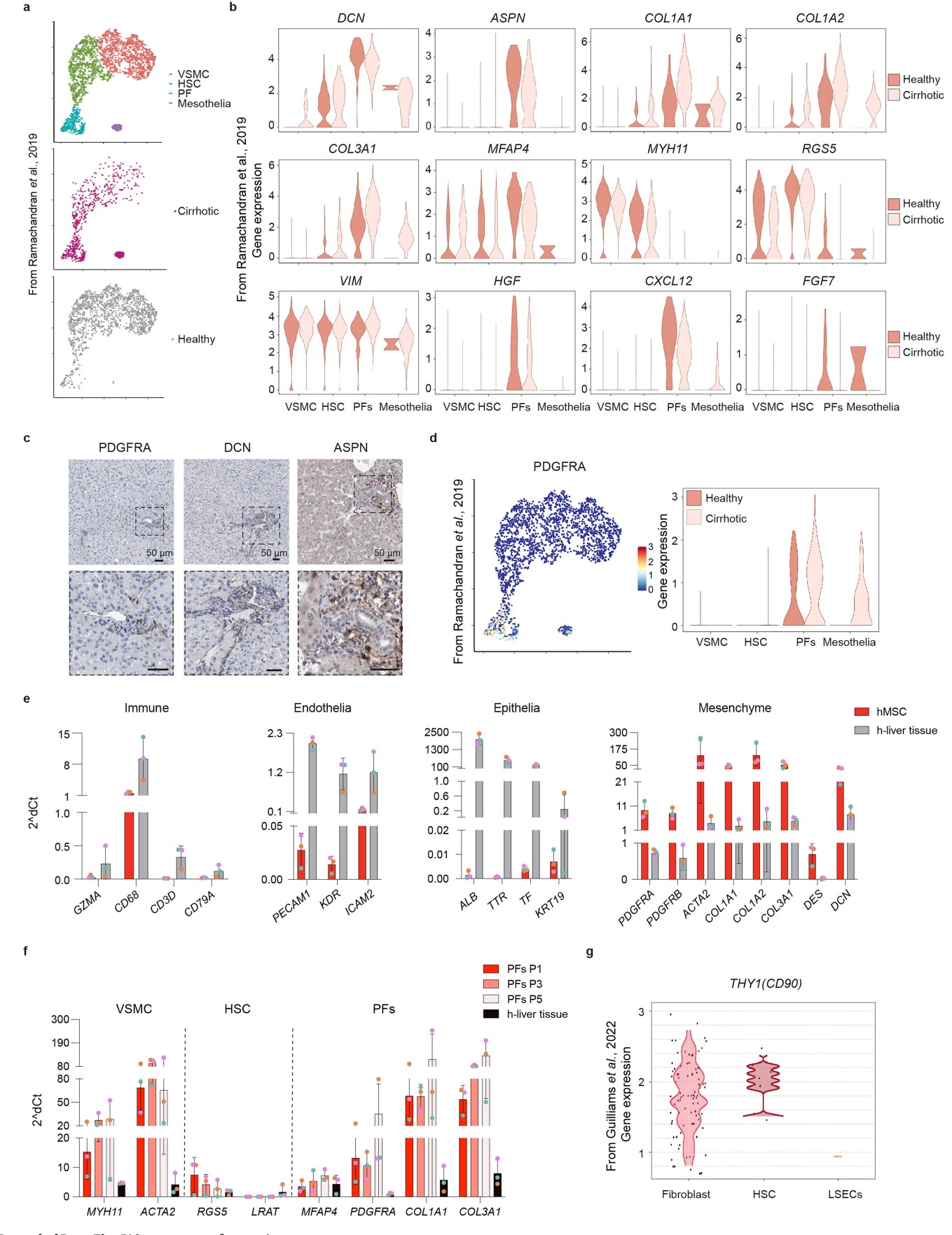

**Extended Data Fig. 5** | See next page for caption.

**Extended Data Fig. 5 | Expression profile of human primary mesenchymal (MSC) populations. a-b**, **d** and **g**, scRNA-seq analysis of human hepatic MSC populations reported in Ramachandran et al.[8] (**a-b, d**) and Guilliams et al.[49] (**g**). The datasets were explored for the expression of specific mesenchymal markers in human healthy liver using the interactive sites https://shiny.igc.ed. ac.uk/livercellatlas/[8] and https://www.livercellatlas.org/umap-ststmouseFibro. php[49] provided by the authors. **a**, tSNE plots show the clustering analysis of the distinct human liver mesenchymal sub-populations in healthy (right) and cirrhotic (middle) human livers reported in Ramachandran et al.[8]. **b**, Violin plot indicates the data point distribution of gene expression for *THY1* (CD90) in the indicated mesenchymal subpopulations of healthy human liver, reported in Guilliams et al.[49]. **c**, Immunohistochemistry for PDGFRA, DCN and ASPN in healthy liver tissue from cohorts of patients from the human protein atlas database (HPA). Scale bars, 50 μm. **d**, Gene tSNE [t-distributed stochastic neighbour embedding] (left) and violin plots (right) show the mRNA expression levels and distribution for *PDGFRa* across different mesenchymal subpopulations. **e**, RT-qPCR analysis of the gene expression of immune, endothelia, epithelia and mesenchyme markers in isolated human mesenchyme (hMSC) and human liver (h-liver). Graph presents mean ± SD from n = 3 independent human donors. Gene expression was normalized to the house-keeping gene *HPRT*. **f**, Gene expression levels for VSMC, HSC and PF markers in serially expanded human portal fibroblasts (hPFs, magenta) and human liver (h-liver, black). Graph presents mean ± SD (n = 3). **g**, Violin plots show the data point distribution of gene expression in the distinct mesenchymal subpopulations for the indicated genes in healthy and cirrhotic human livers reported in Ramachandran et al.[8]. PF: portal fibroblasts /(myo)fibroblasts, HSC: hepatic stellate cells. VSMC vascular smooth muscle cell. Panels **a** and **d** reproduced from ref. 8, Springer Nature Limited. Images in panel **c** reproduced from the Human Protein Atlas (HPA), available under a CC BY-SA 4.0 license (https://creativecommons.org/ licenses/by-sa/4.0/deed.en). PDGFRA, https://www.proteinatlas.org/ ENSG00000134853-PDGFRA/tissue/liver#img; DCN, https://www.proteinatlas. org/ENSG00000011465-DCN/tissue/liver#img; ASPN, https://www.proteinatlas. org/ENSG00000106819-ASPN/tissue/liver#img.

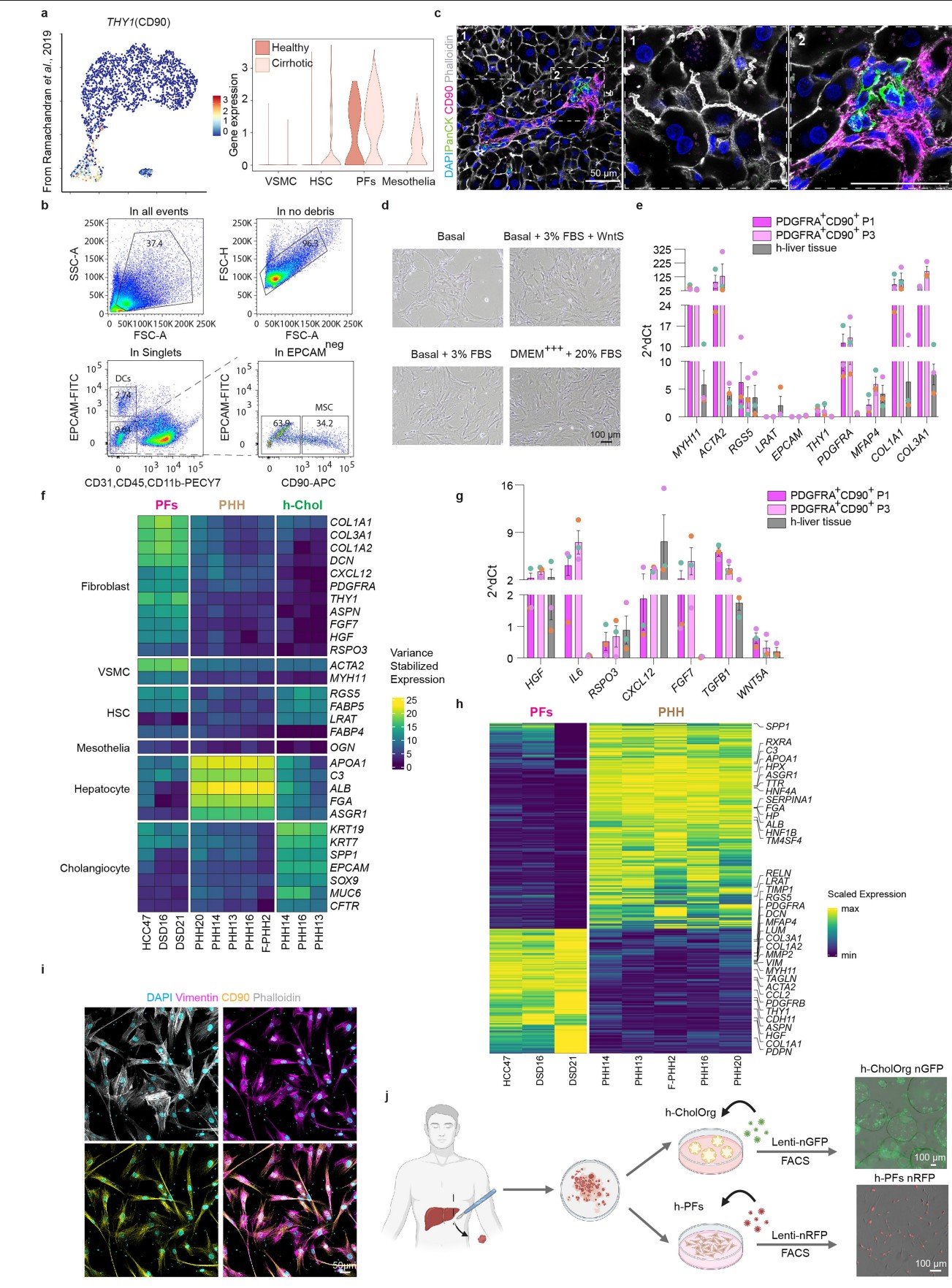

**Extended Data Fig. 6** | See next page for caption.

**Extended Data Fig. 6 | Human portal fibroblasts from patient donors expand in vitro while retaining the expression profile of in vivo portal fibroblasts. a**, Mesenchymal cell populations in human healthy and cirrhotic livers from the publicly available scRNAseq data from Ramachandran et al.[8]. The dataset was explored for the expression of specific portal fibroblast markers in human healthy liver using the interactive site https://shiny.igc.ed. ac.uk/livercellatlas/ provided by the authors. tSNE map (left) and corresponding violin plots (right) depict THY1 (CD90) transcript distribution and expression levels across mesenchymal subpopulations. **b**, FACS gating strategy for isolating PFs from human liver tissue. Immune and endothelial cells are excluded through negative gating for CD31, CD45, and CD11b markers. hPFs are isolated based on EpCAM$^{neg}$/THY1$^{pos}$ sorting. **c**, Immunofluorescence staining of human liver tissue reveals the distribution of different liver cell types (CD90, PF marker, magenta; pan cytokeratin, PanCK, bile duct marker, green). Membranes are marked with phalloidin (white) nuclei with DAPI (blue). Dotted boxes, magnifications showing midzonal tissue parenchyma (1) and liver portal tract (2) region. Note that CD90 is exclusively expressed in the portal region. n = 3 independent experiments. Scale bar, 50 µm. **d**, Representative images of PFs monoculture, cultured in Basal medium, Basal medium with 3% FBS and Wnts, Basal medium with 3% FBS and DMEM with 20% FBS. n = 2 independent experiments. Scale bar, 100 µm. **e**, RT-qPCR gene expression analysis for portal fibroblast (*THY1, PDGFRA, MFAP4, COL1A1, COL3A1*), HSC (*RGS5, LRAT*), cholangiocytes (*EPCAM*) and VSMC (*ACTA2, MYH11*) markers in cultured PDGFRA$^+$CD90$^+$ cells at passage 1 and 3 (P1-P3) and human liver tissue (h-liver). Graph presents mean ± SD (n = 3). Values are presented relative to the house-keeping gene *HPRT*. **f**, Heatmap showing the expression of hepatocyte, cholangiocyte and mesenchymal markers from the indicated subpopulations (PFs, HSC, mesothelia, VSMC) in cultured portal fibroblasts (PF, magenta) and fresh isolated hepatocytes (primary, brown) and cholangiocytes (h-Chol, green) from different donors. PF, portal fibroblast; HSC, hepatic stellate cell; VSMC, vascular smooth muscle cell. Column, donor. **g**, Gene expression levels for the indicated secreted molecules in cultured PDGFRA$^+$CD90$^+$ cells and h-liver analyzed by RT-qPCR. Data are expressed as mean ± SD (n = 3). **h**, Heatmap of all differentially expressed genes in cultured hPFs and freshly isolated hepatocytes (PHH). Note that genes showing expression of mesenchymal markers are highly expressed in the PF population. **i**, Immunofluorescence staining for the portal fibroblast marker, CD90 (yellow) and the pan-mesenchymal marker, Vimentin (magenta) in cultured portal fibroblasts, n = 3 independent experiments in n = 1 donor. Note that all the cells are positive for the PF marker, CD90. Scale bar, 50 µm. **j**, To identify the PFs and cholangiocyte cells after assembly in periportal assembloids, PFs and human cholangiocytes were isolated from human donors (matching whenever possible) and cultured as described in Methods. After several passages, the cells were infected with lentiviral particles containing nuclear-GFP or nuclear-RFP to generate reporter fibroblast lines. Left, schematic diagram of viral infection for h-CholOrg (top) and hPFs (bottom). Representative images of h-CholOrg (top) and PFs (bottom) following transduction with specific nuclear-GFP or nuclear-RFP viral vectors. n ≥ 3 independent experiments. Scale bars, 100 µm. Panel **j** created in BioRender. Yuan, L. (2025) https://BioRender.com/g8uci15.

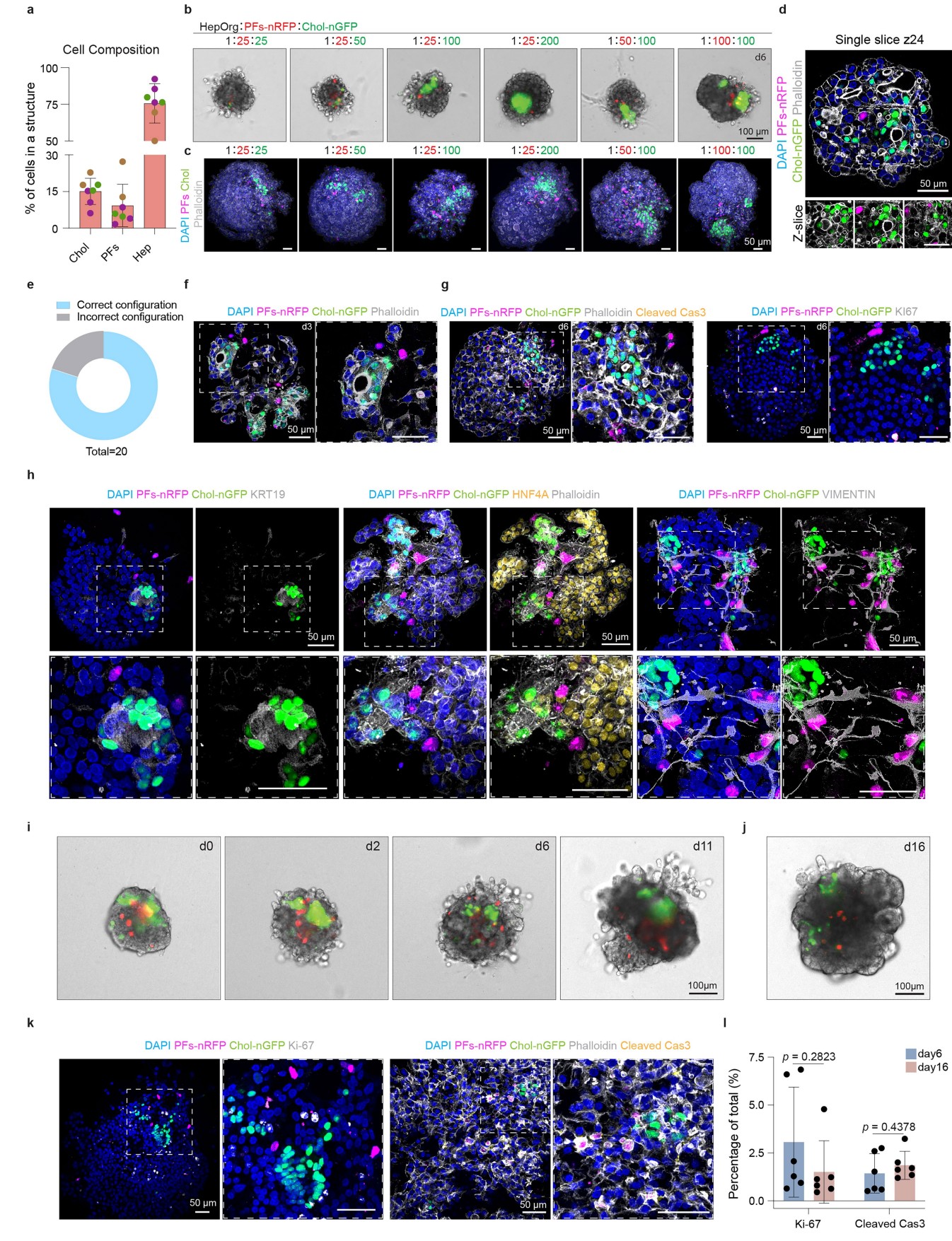

**Extended Data Fig. 7 |** See next page for caption.

**Extended Data Fig. 7 | Optimization and characterization of periportal assembloids. a**, To determine the ratio of hepatocytes, ductal cells/cholangiocytes to mesenchyme to test for the generation of assembloids, human liver tissue was stained for the cholangiocyte marker pan-cytokeratin (PanCK) and mesenchymal marker (CD90). A representative image is presented in Fig. 4d. The numbers of pan-cytokeratin positive ductal cells (PanCK$^+$) and CD90 positive portal mesenchyme were determined and used to calculate the ratio of both populations in human liver tissue. The results are presented as violin plots showing the ratio of PanCK$^+$ to CD90$^+$ cells from n = 3 donors. Dot, one independent portal region. Dot colour, independent donor. **b-d, f-j**, Periportal assembloids were generated by mixing a defined number of human hepatocyte organoids (h-HepOrgs) with n-RFP tagged portal fibroblasts (PFs, magenta) and n-GFP tagged cholangiocytes from cholangiocyte organoids (Chol, green). **b-c**, Representative bright field and epifluorescence images (**b**) and confocal immunofluorescence images (**c**) of day 6 assembloids showing different ratios of hPFs (Red) and Chol (Green) in periportal assembloids. In (c) nuclei were stained with DAPI (blue) and membranes with F-actin (Phalloidin, white). Scale bars, 100 μm (**b**) and 50 μm (**c**). **d**, Representative immunofluorescence images of assembloids after 24 h in AggreWell™ demonstrating that h-PFs (Red) and h-Chol (Green) have already assembled with h-HepOrgs (not labelled) to generate composite structures. Nuclei were stained with DAPI (blue) and membranes with F-actin (Phalloidin, white). Scale bars, 50 μm. **e**, Donut chart illustrating the proportion of correctly (blue) and incorrectly (grey) organized

periportal-like structures amongst assembloids (n = 20). **f**, Representative confocal images of periportal assembloids cultured for 3 days. Nuclei are stained with DAPI (blue); cell membranes with Phalloidin (white). Scale bars, 50 μm. **g**, Representative confocal images showing the proliferative (right, Ki-67, white) and apoptotic (left, Cleaved Caspase 3, yellow) states on day-6 assembloids. Nuclei are stained with DAPI (blue) and in the left panel membranes are stained with F-actin (Phalloidin, white). Scale bars, 50 μm. **h**, Left to right, representative confocal images of periportal assembloids cultured for 3 days and stained for cholangiocyte (KRT19, white, left), hepatocyte (HNF4A, yellow, middle) and hPF (vimentin, white, right) markers. Nuclei are stained with DAPI (blue) and in the middle panel, membranes are stained with F-actin (Phalloidin, white). Scale bars, 50 μm. **i-j**, Representative bright field images of human liver assembloids cultured for 2 weeks indicating that the three populations persist in the composite structures. Scale bars, 100 μm. **k**, Representative confocal images showing the proliferative (left, Ki-67, white) and apoptotic (right, Cleaved Caspase 3, yellow) states on long-term cultured assembloids (16 days). n = 3 independent experiments. Scale bars, 50 μm. **l**, Quantification of Ki-67$^+$ and Cleaved Caspase 3$^+$ cells in assembloids cultured for 6 (the 'short term' condition, as described in Extended Data Fig. 7f, for comparison) or 16 days. Results are presented as mean ± SD of n = 3 donors, biological replicates. $p$-values were calculated using two-sided unpaired Student's t-test.

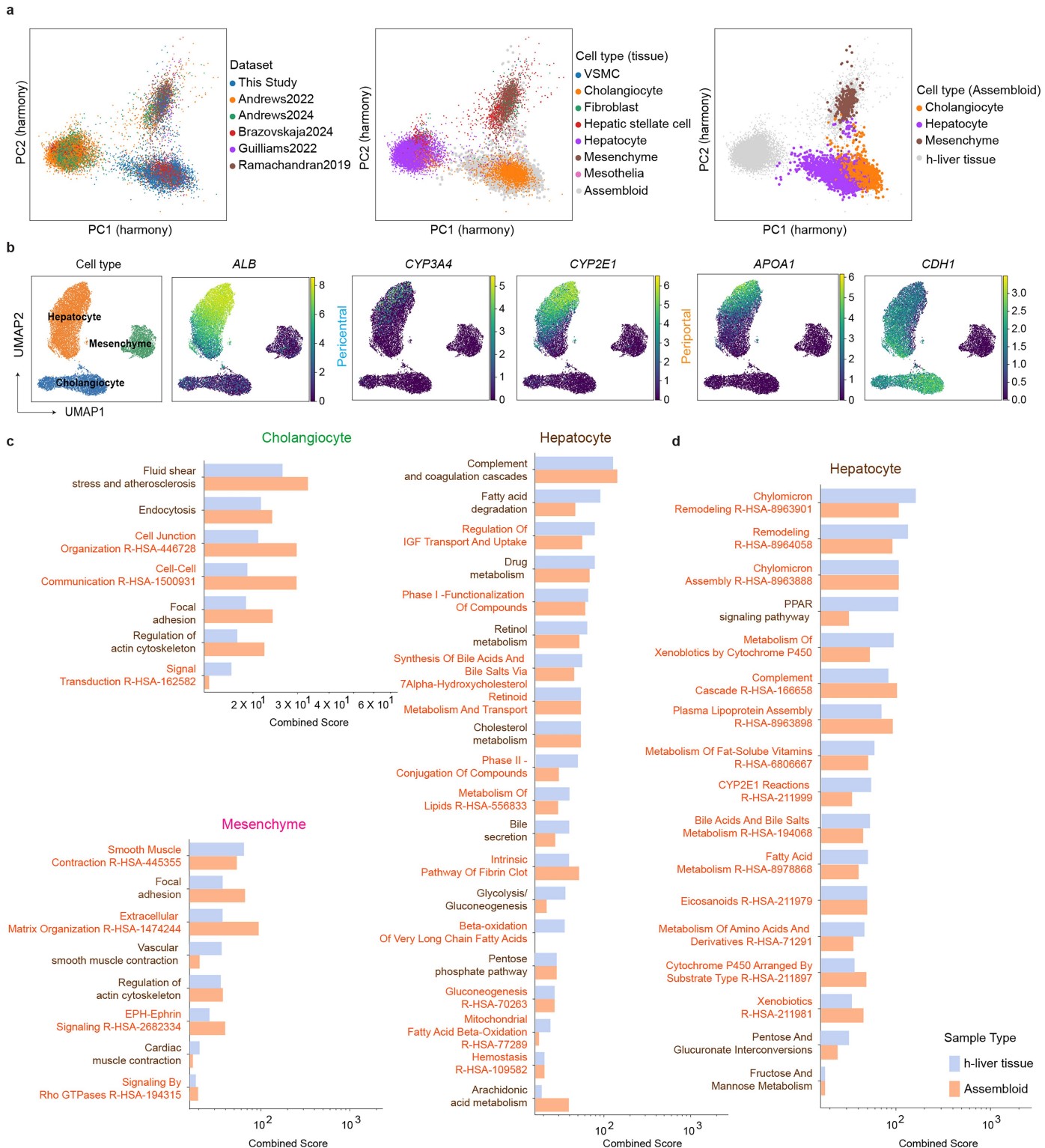

**Extended Data Fig. 8 | scRNAseq analysis of human periportal assembloids.**
**a-c**, Assembloids were generated by assembling h-HepOrgs, cholangiocytes/
ductal cells derived from cholangiocytes organoids (nuclear-GFP) and portal
fibroblasts (PFs, nuclear-RFP) at a ratio 1 h-HepOrgs: 25 PFs:100 Cholangiocytes
and 5-6 days later the cultures were collected and submitted for scRNAseq
analysis (see Methods for details). **a**, PCA after batch correction with Harmony
showing the integration of scRNAseq data from assembloids (This study) with
human liver cell atlas datasets. **b**, UMAPs showing the 3 different clusters of
cells in assembloids and the expression of the hepatocyte marker (*ALB*, albumin),
the pericentral markers (*CYP3A4* and *CYP2E1*) and the periportal markers

(*APOA1* and *CDH1*). The full list of markers for each population can be found in
Supplementary Dataset 3. **c**, GSEA for Reactome (orange) and KEGG (brown)
pathways in cholangiocytes (top) and Mesenchyme (bottom) from periportal
assembloids (orange bar) and the corresponding enrichment in in vivo human
liver tissue (blue bar). Data is presented as combined score. The full list can be
found in Supplementary Dataset 3. **d**, GSEA for Reactome (orange) and KEGG
(brown) pathways in hepatocytes from periportal assembloids (orange bar)
and the corresponding enrichment in in vivo human liver tissue (blue bar).
Data is presented as combined score. The full list can be found in Supplementary
Dataset 3.

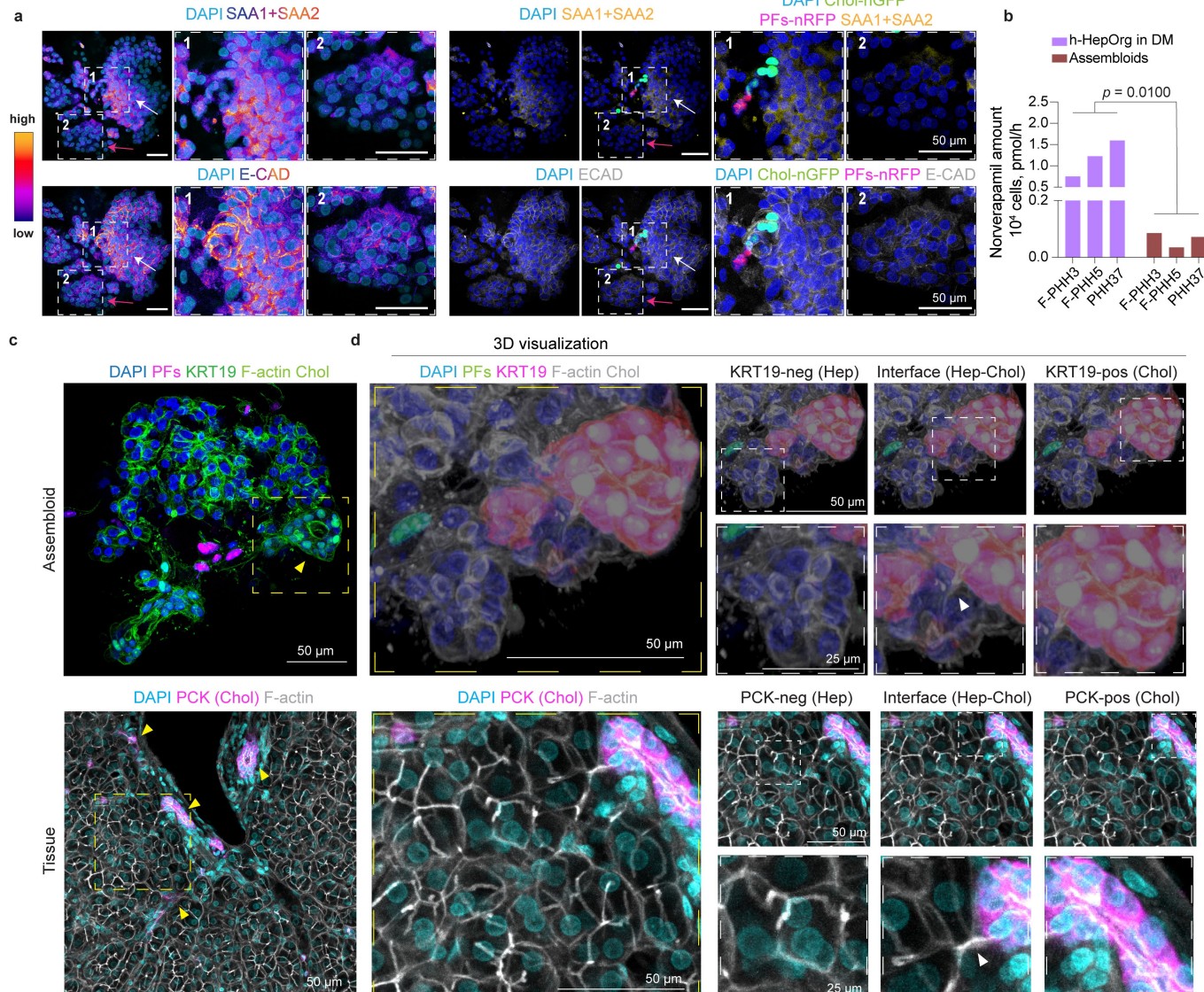

**Extended Data Fig. 9 | Hepatocytes, cholangiocytes and mesenchyme in assembloids express markers and establish cell-cell interactions similar to the in vivo portal tissue. a**, Immunofluorescence staining of day 4 periportal assembloids using the periportal markers SAA1 and SAA2 (yellow) and the epithelial and periportal enriched marker ECAD (white). Hepatocytes with positive (white arrow, region 1) and negative (magenta arrow, region 2) SAA1/SAA2 staining are shown. Left, robust expression of the portal markers SAA1 and SAA2 (top) and ECAD (bottom) in hepatocytes from assembloids. FIRE-LUT illustrates the different intensity of the corresponding markers. Right, co-localization of SAA1 and SAA2 staining with cholangiocytes (Chol-nGFP, green) and portal Mesenchyme (PFs-nRFP, magenta) highlights that the hepatocytes with strongest expression of portal markers SAA1 and SAA2 localize nearby cholangiocytes surrounded by portal fibroblasts (PFs-nRFP, magenta), confirming the periportal-like nature of assembloids. Portal identity is further contextualized by co-localization with ECAD (E-cadherin; epithelial marker) (bottom panels). Representative images from n = 2 samples from n = 1 donor. Scale bars, 50 µm. **b**, Mass spectrometry analysis was used to detect Norverapamil, the primary metabolite of Verapamil, in assembloids (6-day culture) and differentiated hepatocyte organoids (h-HepOrgs-DM). Unpaired two-tailed Student's t-test with Welch's correction, comparing biological replicates (n = 3). Values for h-HepOrgs correspond to the data in Fig. 3 and are shown here for comparison against assembloids. **c**, Immunofluorescent images showing the similarity between human assembloid (top) and human liver tissue (bottom). In assembloids, mesenchymal cells are labelled in magenta, Cholangiocytes labelled in green, F-actin and KRT19 are stained in green as well, and nuclei with DAPI (blue), n = 3 independent experiments. The assembloid image corresponds to the one presented in Fig. 5f where KRT19 in white is shown. In human liver tissue, staining was done for cholangiocytes (PCK, magenta), cell borders (F-actin, grey) and nuclei (DAPI, cyan), n = 3 independent experiments from n = 2 independent donors. Yellow arrowheads indicate the presence of ductal structures amidst the hepatocytes in both assembloids and tissue. Scale bar, 50 µm. **d**, Top, images showing magnified areas and maximum intensity projection of the assembloids (from panel c, top) with hepatocytes (DAPI$^{pos}$, KRT19$^{neg}$), cholangiocytes (KRT19$^{pos}$, nuclei white), and the interface between the two cell-types - indicated by the white arrowhead, n = 3 independent experiments. The images are immunofluorescent images visualized in 3D to enable viewing from another angle. Bottom, immunofluorescent images showing magnified areas of human liver tissue (from panel c, bottom) with hepatocytes (PCK$^{neg}$), cholangiocytes (PCK$^{pos}$), and the interface between the two cell-types - indicated by the white arrowhead, n = 3 independent experiments from n = 2 biologically independent donors. Scale bar, 50 µm, 25 µm.

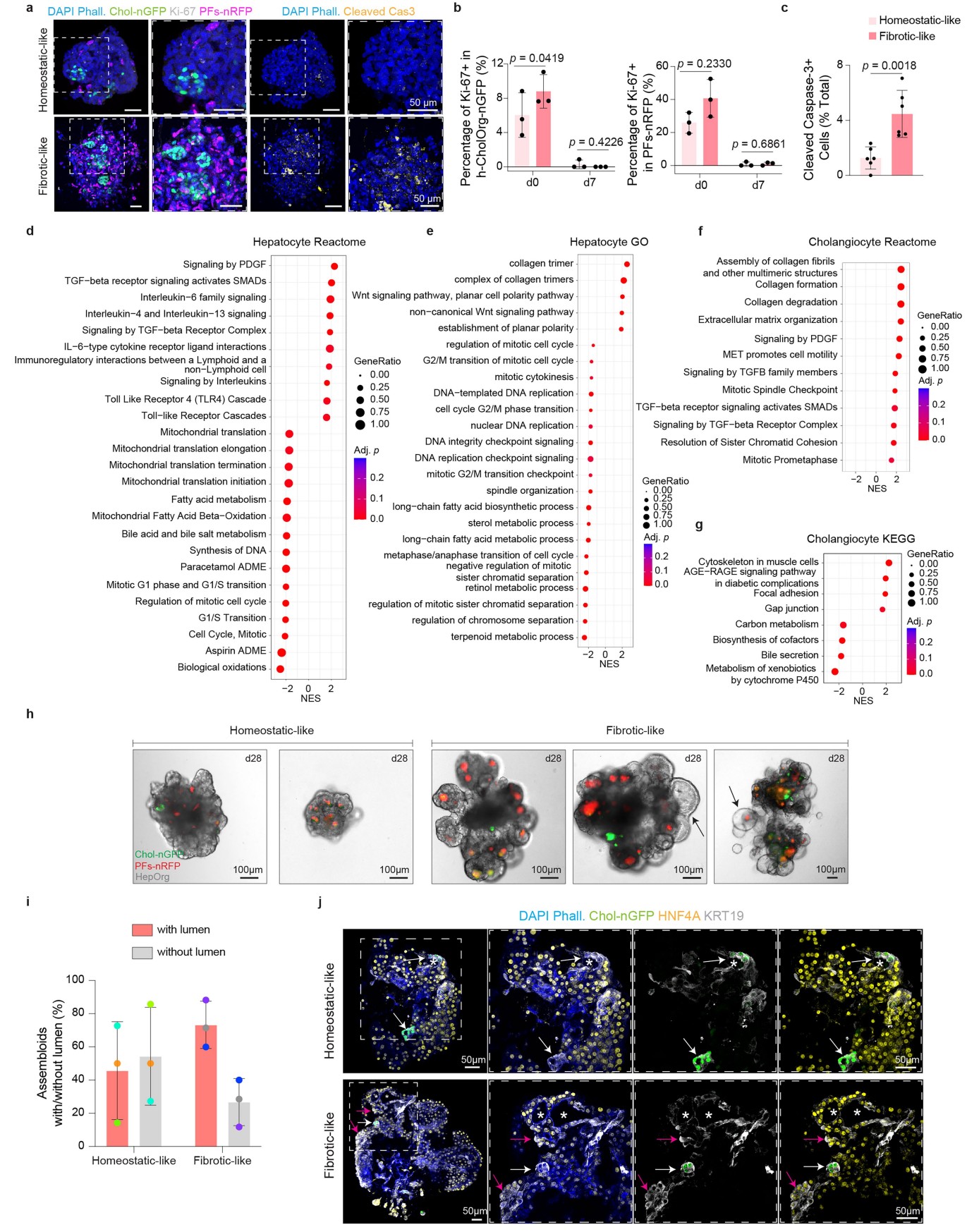

**Extended Data Fig. 10 | Modelling hepatocyte transdifferentiation and epithelial responses to excess portal mesenchyme in human periportal assembloids. a**, Representative immunofluorescence images of periportal assembloids with homeostatic (top) or fibrotic (bottom). Ki-67 (white) and Cleaved Caspase-3 (yellow) mark proliferating and apoptotic cells, respectively. Phalloidin (blue) marks membranes; DAPI (blue), nuclei. Scale bars, 50 μm. **b**, Quantification of Ki-67$^+$ in cholangiocytes and PFs in 24 h (day 0) and day 7-homeostatic (light pink) and fibrotic (dark pink) assembloids. Note the significantly higher proliferation of cholangiocytes under fibrotic-like conditions compared to homeostatic-like at day 0. Mesenchymal cells showed a trend toward higher proliferation although was not significant. At day 7 proliferation rates decreased. Results are presented as mean ± SD of n = 3 independent donors. *p*-values were calculated using two-sided paired (by donor) Student's t-test. **c**, Quantification of total number of cleaved Caspase-3$^+$ cells per assembloids at day 7 in homeostatic-like and fibrosis-like assembloids. Data is expressed as a percentage of + cells per total cell numbers. Dot, organoid per donor (n = 3 donors). Bars represent mean ± SD. *p*-value was calculated using a two-sided paired (by donor) Student's t-test. **d-g**, Additional gene sets enriched in each of the three cell types in fibrotic-like *vs* homeostatic-like assembloids. Note that fibrotic-like hepatocytes and cholangiocytes are positively enriched for inflammatory signatures. Additionally, fibrotic-like hepatocytes are negatively enriched for functional pathways and cell cycle functions. Results are presented as dot plot, where dot colour represents the adjusted *p*-value (permutation test implemented in clusterProfiler, adjusted using the Benjamini–Hochberg method), and dot size corresponds to the GeneRatio (number of core enrichment genes divided by the total number of genes in the pathway). NES, normalized enrichment score. The full list can be found in Supplementary Dataset 4. **h-j**, Assembloids with homeostatic numbers (homeostatic-like) or excess amount of mesenchyme (fibrotic-like) were generated as described in Methods and cultured for 28 days in assembloids medium. **h**, Representative bright field images of homeostatic-like (left) and fibrotic-like (right) assembloids cultured for 28 days containing cholangiocytes/ductal cells (green), mesenchyme (magenta) and hepatocytes (unlabelled). Note that only fibrotic assembloids opened big lumina (arrow) and acquired a ductal/cholangiocyte cystic-like organoid shape similar to the organoids presented in Extended Data Fig. 1a and published in Huch et al.[4] Scale bars, 100 μm. **i**, Quantification of the number of organoids with cystic phenotypes (with lumen) *vs* solid organoids (without lumen). Graph presents the mean ± SD of n = 3 independent experiments with n = 2 independent donors. Dot colour, independent experiment. **j**, Representative confocal images of homeostatic (top) and fibrotic-like (bottom) periportal assembloids stained for hepatocyte (HNF4A, yellow) and cholangiocyte (KRT19, white) markers. Nuclei and membranes were also stained with DAPI (blue) and phalloidin (blue, left and middle), respectively. Note that in homeostatic assembloids the lumen (asterisk) is formed by nGFP$^+$KRT19$^+$ ductal cells (white arrow), while in fibrosis-like assembloids the lumen (asterisk) is also formed by cells that are double positive for HNF4A$^+$ KRT19$^+$ (magenta arrow).

# Reporting Summary

## Statistics

For all statistical analyses, confirm that the following items are present in the figure legend, table legend, main text, or Methods section.

| n/a | Confirmed | |
|---|---|---|
| ☐ | ☒ | The exact sample size (*n*) for each experimental group/condition, given as a discrete number and unit of measurement |
| ☐ | ☒ | A statement on whether measurements were taken from distinct samples or whether the same sample was measured repeatedly |
| ☐ | ☒ | The statistical test(s) used AND whether they are one- or two-sided<br>*Only common tests should be described solely by name; describe more complex techniques in the Methods section.* |
| ☐ | ☒ | A description of all covariates tested |
| ☐ | ☒ | A description of any assumptions or corrections, such as tests of normality and adjustment for multiple comparisons |
| ☐ | ☒ | A full description of the statistical parameters including central tendency (e.g. means) or other basic estimates (e.g. regression coefficient) AND variation (e.g. standard deviation) or associated estimates of uncertainty (e.g. confidence intervals) |
| ☐ | ☒ | For null hypothesis testing, the test statistic (e.g. *F*, *t*, *r*) with confidence intervals, effect sizes, degrees of freedom and *P* value noted<br>*Give P values as exact values whenever suitable.* |
| ☒ | ☐ | For Bayesian analysis, information on the choice of priors and Markov chain Monte Carlo settings |
| ☒ | ☐ | For hierarchical and complex designs, identification of the appropriate level for tests and full reporting of outcomes |
| ☐ | ☒ | Estimates of effect sizes (e.g. Cohen's *d*, Pearson's *r*), indicating how they were calculated |

*Our web collection on statistics for biologists contains articles on many of the points above.*

## Software and code

Policy information about availability of computer code

| Data collection | The following software was used for data collection:<br>Confocal Imaging: Zeiss ZEN blue edition (version 3.6.095.09000)<br><br>Brightfield imaging - Leica Application Suite v.4.6.0 (Leica DMIL LED microscope) or v.4.13.0 (Leica M80 stero-microscope) ;<br><br>qPCR - QuantStudio™ Design & Analysis 2.7.0 software (ThermoFisher);<br><br>Gluconeogenesis, Urea synthesis, Albumin ELISA and bile acid assay - Perkin Elmer Envision 2104 EnVision Manager v.1.13.3009.1401.<br>Cell Counting- Countess® II FL Automated Cell Counter v.1.0.247<br>Quantification of xenobiotic metabolism by mass spectrometry - Q Exactive hybrid quadrupole Orbitrap mass spectrometer (ThermoFischerScientific, USA) and TriVersa NanoMate robotic ion source (Advion Interchim Scientific, USA) Chipsoft 8.1.0 software |
|---|---|

| Data analysis | Confocal imaging data was analysed with FIJI v.2.14.0/1.54f, and segmented using Arivis Vision 4D (Version: 4.1.0. Build: 16702. 20200324) or Motion Tracking v. 8.100.6 (http://motiontracking.mpi-cbg.de). Data analysis of bile canaliculi was performed using a custom script in FIJI, presented in  https://github.com/JulienDelpierre/BileCanaliculiSegmentation.<br>Data presentation and statistical analysis performed with GraphPad Prism v10 (10.3.0) for non-sequencing data.<br>Source code used for data analysis and visualisation of bulkRNAseq and scRNAseq dataset are available at https://git.mpi-cbg.de/huch_lab/yuan_dawka_kim_liebert_et_al_2025_sequencing and together with processed data on Zenodo (https://doi.org/10.5281/zenodo.17251198).<br>Spectra were averaged in Xcalubur Qual Browser v.3.0 (ThermoFischerScientific, USA) |

For manuscripts utilizing custom algorithms or software that are central to the research but not yet described in published literature, software must be made available to editors and reviewers. We strongly encourage code deposition in a community repository (e.g. GitHub). See the Nature Portfolio guidelines for submitting code & software for further information.

# Data

Policy information about availability of data

All manuscripts must include a data availability statement. This statement should provide the following information, where applicable:
- Accession codes, unique identifiers, or web links for publicly available datasets
- A description of any restrictions on data availability
- For clinical datasets or third party data, please ensure that the statement adheres to our policy

The raw single-cell RNA sequencing (dataset id EGAD50000001453) and bulk RNA sequencing (dataset id EGAD50000001454) data (FASTQ files) generated in this study have been deposited in the European Genome-phenome Archive (EGA) under accession number EGAS50000000994. Access to these data is controlled to protect the privacy and identity of study participants. Access requests can be submitted via the EGA website. Requests will be evaluated by the Data Access Committee (DAC) EGAC50000000112 to ensure that proposed data use is consistent with the consent provided by participants. Approved users will be required to sign a Data Access Agreement (DAA) that specifies the permitted uses of the data. A template of the DAA is available at https://edmond.mpg.de/api/access/datafile/250610. Applications are normally reviewed within 10 working days. Count matrices and fully processed data, together with the source code for the sequencing data analysis, are available via Zenodo (https://doi.org/10.5281/zenodo.17251198). Comprehensive lists of differentially expressed genes, gene set enrichment analysis (GSEA) terms, and marker genes are provided in Supplementary Datasets 2–4.

Sources for annotated scRNAseq/snRNAseq count matrices used as reference datasets in the scRNAseq analysis: Data from Andrews et al., 2022, Andrews et al., 2024, and Guiliams et al., 2022 were downloaded from CZ CELLxGENE Discover (census version 2024-07-01)(https://cellxgene.cziscience.com/) using the Python package cellxgene-census (v1.16.2). For these three publications, author annotations of cell types were retrieved by downloading the h5ad files for the respective publications from the CZ CELLxGENE Discover web interface. The data from Brazovskaja et al., 2024 was downloaded from Mendeley Data (https://doi.org/10.17632/yp3txzw64c.1, "allcells_css_annot.rds"). The data from Ramachandran et al., 2019 was downloaded from Edinburgh DataShare (https://datashare.ed.ac.uk/handle/10283/3433, "tissue.rdata"). We provide Jupyter notebooks that perform the data download in our git repository (see code availability statement).

The raw data from mass spectrometry analyses are available at the following repository: https://doi.org/10.17617/3.Z9GMJE.

All other images, qPCR and measurement data are presented within the manuscript.

# Research involving human participants, their data, or biological material

Policy information about studies with human participants or human data. See also policy information about sex, gender (identity/presentation), and sexual orientation and race, ethnicity and racism.

| Reporting on sex and gender | All patient information regarding sex is provided in Supplementary Table 1 and 2 |
| Reporting on race, ethnicity, or other socially relevant groupings | N/A |
| Population characteristics | N/A |
| Recruitment | N/A |
| Ethics oversight | N/A |

Note that full information on the approval of the study protocol must also be provided in the manuscript.

# Field-specific reporting

Please select the one below that is the best fit for your research. If you are not sure, read the appropriate sections before making your selection.

☒ Life sciences　　　☐ Behavioural & social sciences　　　☐ Ecological, evolutionary & environmental sciences

For a reference copy of the document with all sections, see nature.com/documents/nr-reporting-summary-flat.pdf

# Life sciences study design

All studies must disclose on these points even when the disclosure is negative.

| | |
|---|---|
| Sample size | No statistical methods were used to estimate sample sizes before the study. We based our sample numbers on a combination of previous experience with in vivo and in vitro systems, and standards in the field. |
| Data exclusions | scRNAseq thresholding involved excluding cells with abnormally low transcripts, and based on high mitochondrial gene content as is standard practice for downstream analysis of this sequencing data. We also excluded doublets to improve stringency, as described in the methods. |
| Replication | With the exception of scRNAseq due to the nature of the analysis, all experiments were replicated as indicated in the figure legend. For the scRNAseq we did n=4 different donors per condition |
| Randomization | Experimental procedures always involved processing control and experimental samples in a random order, rather than by condition. |
| Blinding | Investigators were blinded to the culture conditions for the experiments in Fig1 -3 and associated Extended Data Fig 1-3 and 8-10. For the remainder of the experiments the researchers were not blinded. |

# Reporting for specific materials, systems and methods

We require information from authors about some types of materials, experimental systems and methods used in many studies. Here, indicate whether each material, system or method listed is relevant to your study. If you are not sure if a list item applies to your research, read the appropriate section before selecting a response.

## Materials & experimental systems

| n/a | Involved in the study |
|---|---|
| ☐ | ☒ Antibodies |
| ☐ | ☒ Eukaryotic cell lines |
| ☒ | ☐ Palaeontology and archaeology |
| ☐ | ☒ Animals and other organisms |
| ☒ | ☐ Clinical data |
| ☒ | ☐ Dual use research of concern |
| ☒ | ☐ Plants |

## Methods

| n/a | Involved in the study |
|---|---|
| ☒ | ☐ ChIP-seq |
| ☐ | ☒ Flow cytometry |
| ☒ | ☐ MRI-based neuroimaging |

## Antibodies

| | |
|---|---|
| Antibodies used | List of antibodies used in this study are provided in Supplementary Dataset 5_reagents_S1. |
| Validation | All antibodies are commercially available, and were validated for specificity and application by manufacturers listed above. Antibodies were used at concentrations suggested in previous published methodologies or titrated in-house. |

## Eukaryotic cell lines

Policy information about cell lines and Sex and Gender in Research

| | |
|---|---|
| Cell line source(s) | The mesenchymal cells and organoid lines are all primary material derived from freshly isolated human livers by investigators in this study. Ethical approval is provided in methods. Five lines (F-PHH1, F-PHH2, F-PHH3, F-PHH4 and F-PHH5) were obtained from Lonza as described in Supplementary Table 2. |
| Authentication | N/A |
| Mycoplasma contamination | Mycoplasma contamination was regularly tested on all laboratory lines throughout this study, using MycoAlert® Mycoplasma Detection Kit (Lonza #LT07-118). |
| Commonly misidentified lines (See ICLAC register) | No ICLAC lines were used in this study. |

# Animals and other research organisms

Policy information about studies involving animals; ARRIVE guidelines recommended for reporting animal research, and Sex and Gender in Research

| | |
|---|---|
| Laboratory animals | Male and female Fah-/-/Rag2-/-/Il2rg-/- (FRG) mice were obtained from Jackson Laboratory. Mice were kept under standard husbandry in a pathogen-free environment with a 12 h day/night cycle. Sterile food and water were given ad libitum. |
| Wild animals | This study did not involve wild animals. |
| Reporting on sex | Both male and female mice were used in this study. |
| Field-collected samples | This study did not involve field-collected samples. |
| Ethics oversight | Mice were maintained in accordance with the Principles of Laboratory Animal Care and the Guide set by the HYU Industry-University Cooperation Foundation. |

Note that full information on the approval of the study protocol must also be provided in the manuscript.

# Flow Cytometry

## Plots

Confirm that:

☒ The axis labels state the marker and fluorochrome used (e.g. CD4-FITC).

☒ The axis scales are clearly visible. Include numbers along axes only for bottom left plot of group (a 'group' is an analysis of identical markers).

☒ All plots are contour plots with outliers or pseudocolor plots.

☐ A numerical value for number of cells or percentage (with statistics) is provided.

## Methodology

| | |
|---|---|
| Sample preparation | Primary human portal fibroblasts culture in DMEM supplemented with 1% HEPES, 1% GlutaMax, 1% Penicillin/Streptomycin, and 20% FBS at 37°C and 5% $CO_2$ were trypsinized using TRYPLE 1x and stained with 1 μg/test Anti-human CD90 (THY1)-APC, 20 μL/test Anti-human CD140a (PDGFRa)-PE, Anti-CD11b/CD31/CD45-PECy7, and EpCAM-Alexa 488 for 30 minutes on ice and washed twice and sorted on a FACS aria, collected in DMEM supplemented with 1% HEPES, 1% GlutaMax, 1% Penicillin/ Streptomycin, and 20% FBS and cultured as described in methods at 37°C and 5% $CO_2$. |
| Instrument | BD FACSAria Fusion |
| Software | BD-FACS-Diva |
| Cell population abundance | *Describe the abundance of the relevant cell populations within post-sort fractions, providing details on the purity of the samples and how it was determined.* |
| Gating strategy | Cells were first gated according to their SSC-A/FSC-A. Then, single cells were identified using FSC-H/FSC-A. The single cell population was then gated according to their EpCAM-FITC+ and CD31,45,11b -PE-Cy7 expression. The negative gate was used to further separate THY1+ vs THY- cells according to the THY-APC positive gating. |

☒ Tick this box to confirm that a figure exemplifying the gating strategy is provided in the Supplementary Information.

