## [Peer Review File · Nature]

Human assembloids recapitulate periportal liver tissue in vitro

Corresponding Author: Professor Meritxell Huch

Version 0:

Reviewer comments:

Referee #1

(Remarks to the Author)

In this interesting manuscript entitled "Human periportal liver assembloids recapitulate periportal liver tissue in vitro" Meritxell Huch and her team describe the development of a human hepatocyte organoid model and its evolution into a periportal assembloid model, by adding additional cell types. The authors generated a biobank from different donors, carefully characterized their model and compared key physiological and pathophysiological features with patient tissue and primary counterparts of the cell types used. The manuscript is well written, the illustrations are beautiful, and conclusions are supported by solid data. However, in my opinion the authors should better highlight the advance of the organoid model over previously published models, assess to what extent their models retain disease characteristics of the donor/source tissue and add more use case data relevant for drug discovery before publication.

Major points:

- 1) The introduction is missing relevant publications describing proliferating primary hepatocytes (ProlHH cells from the Hui lab) and previously described hepatocyte organoids (from the Clevers and Nusse labs). The authors should more clearly highlight how their organoid model represents an advance over other culture systems. I clearly see the advance of the assembloid system over mono-cellular systems. However, the hepatocyte organoid system itself requires more differentiation. Most other hepatocyte organoid systems lose metabolic function and suffer from biliary metaplasia as well as EMT. The present hepatocyte organoid system should be compared to other similar systems in these aspects.
- 2) Do patient-derived organoids retain functional characteristics of the diseased source liver or do they get rectified over time in culture? Were the primary hepatocytes used in the profiling experiments source material for h-HepOrgs? Moreover, what if mesenchyme from different patients is used, does this have disease-related implications to the epithelial assembloid part?
- 3) Figure 1b and SuppFig 1: The terminology related to the YAP/Hippo pathway is confusing to me. Usually people refer to the Hippo pathway and YAP signaling. Likewise, LATS1 mRNA downregulation does not inform on YAP signaling status. The authors should check YAP target genes to assess that status of the pathway. Also, the authors should better talk about activating YAP signaling rather than blocking HIPPO signaling, to not confuse the reader.
- 4) Since TRULI is a rather unspecific kinase inhibitor, hitting many other kinases in addition to LATS1/2, one cannot refer its effects solely to YAP/HIPPO signaling. There are more specific LATS kinase inhibitors out there (NIBR-LTSi) but noting limitations of TRULI may be sufficient, rather than repeating experiments. In addition, this statement should be refined, as the data shows modulation of the respective pathways but does not indicate specificity of the drugs: "Gene set enrichment analysis showed that h-HepOrg exhibited high expression of Wnt and YAP targets, indicating that the Wnt agonist and the LATS1/2 inhibitor used act specifically through these respective pathways».
- 5) SOX9 and KRT7 should be added to Figure 2D since they are better markers to show biliary metaplasia in hepatocytes

compared to the more mature biliary markers (KRT19 and EPCAM) used in the heatmap. Moreover, EMT is key issue for primary hepatocytes in culture since under normal 2D culture conditions they become fibroblast like and lose metabolic function. It would therefore be great if the authors could look into this in their model and compare to other systems in addition to biliary metaplasia.

6) An important use case for physiological hepatocyte-like system is testing of drug metabolism. It would be great if the authors could compare metabolism for few drugs in their h-HepOrgs versus assembloids versus primary hepatocytes.

Minor points:

7) Fig5e – it would be great if the authors could provide information of the % composition in a patient liver for comparison.

8) Are patients 1-3 and organoids A-B in Extended data Figure 4 related? The data is not clear in whether the organoids retain the patient characteristics or whether there are just different types of BC in general in both conditions.

Referee #2

(Remarks to the Author)

This manuscript by Yuan, Dawka, Kim and Liebert et al. describes the development of human hepatocyte organoids from adult human donor livers, followed by the development of multi-lineage human organoids containing hepatocytes, portal mesenchyme and cholangiocytes. The authors claim that these liver assembloids retain the histological arrangement, gene expression and cell-cell interactions observed in vivo in human periportal liver tissue - however I do not think the data presented supports these claims, ie that these liver assembloids closely recapitulate and mimic the topography, architecture and function of human portal tracts.

I have a number of concerns regarding this manuscript which I have listed below:

1. The first few sections of this manuscript describe how the authors optimised their protocols to allow long-term expansion of human hepatocyte organoids. Although this is of course important background work, a large amount of results text was used to describe the optimisation process. I think it would be helpful to the reader to shorten the amount of results text currently used to describe what is essentially method optimisation, and be more concise in the description of how the human hepatocyte organoid expansion protocols were iterated.

2. Fig 1c and Ext Fig 1a+b: It would be helpful to add cell viability graphs to accompany these data.

3. Ext Fig 1e, line 156: the authors should show quantitation and statistics regarding these data as they state significance in the results text ('resulted in significant hepatocyte organoid growth').

4. Ext Fig 1h, passage information would be useful to include in this graph, similar to that shown in Fig 1f.

5. Fig 2f. Regarding the h-HepOrg panels - are these different patients? I.F. markers are not consistent between tissue and hHepOrgs eg Alb and ECAD. It would help the reader if the markers / stains were consistent across all 3 panels.

6. Lines 195-198: 'Gene set enrichment analysis showed that h-HepOrg exhibited high expression of Wnt and YAP targets, indicating that the Wnt agonist and the LATS1/2 inhibitor used act specifically through these respective pathways (Extended Data Fig.2e, g-h)'.

The gene set enrichment analysis is insufficient to support the claim that 'the Wnt agonist and the LATS1/2 inhibitor used act specifically through these respective pathways' - more experimental data should be presented to substantiate the claim that the inhibitors used act specifically through these pathways.

7. Fig2b: The percentage of nuclear YAP+ and nuclear YAP- cells should be quantitated and statistics shown.

8. Fig 2c: Cytoplasmic to nuclei ratio and bile canalicular architecture should be quantitated and statistics shown.

9. Fig 2f: The albumin staining in the upper panel in human liver tissue is not convincing. Further, the ECAD and GS staining in the middle and lower panels should be quantitated and measured across multiple donors to allow the reader to assess the variation / reproducibility across donors.

10. Fig 3c - Connectivity within the BC network - statistical analysis should be performed to compare the groups.

11. Extended data fig 4b-d: Are patients 1,2 and 3 the same or different from patients A,B and C in these panels? It would be useful to quantitate these readouts in tissue vs organoids from the same donor livers to allow assessment of how well each individual set of organoids recapitulate in vivo liver tissue from the same donor.

12. Fig 3d: The authors state 'Differentiated h-HepOrg presented mature hepatic functions similar to freshly isolated hepatocytes, including albumin secretion and, to a lesser extent, cytochrome p450 activity (Fig.3d-e)'.

However the CYP2C9 and CYP3A4 activity presented (Fig3d) is statistically significantly different between PHHs and h-HepOrg-DM - please clarify.

13. The authors state that the assembloids mimic human portal architecture, however much of the data shows immunofluorescence staining of central hepatocyte markers, and many central hepatocyte genes are present in the RNA sequencing data. The authors need to further characterise the assembloids in terms of portal vs central hepatocyte ratios / marker expression to investigate whether the assembloids are enriched for portal hepatocytes, in line with trying to mimic portal tissue architecture and function.

For example, in Fig 6c, it is interesting to note that most of the assembloid cells express central hepatocyte genes (Glul/Cyp2e1) despite the biliary context put forward by the authors. As mentioned above, this requires further investigation - if the authors seek to mimic the portal tract it would be important to characterise in-depth whether there are many peri-portal hepatocytes in the assembloids (markers such as HAL could be used to assess this with I.F. staining) and do these cells exhibit portal hepatocyte functions (assessed using assays beyond gene-level data).

14. Fig 3f: Xenotransplantation of organoids and PHHs into the Tyrosinemia type I liver disease mouse model (Fah^{-/-}Rag2^{-/-}Il2rg^{-/-} mouse):

Only the survival curve data is presented here. The authors should show the following data:

a) the number of mice in each group - I could not find this information in the Results, Methods or Fig legend text.

b) What was the efficiency of engraftment of the organoids and PHHs? An intrasplenic injection was performed, but there is no data presented demonstrating that the hepatic organoids engrafted efficiently into the liver.

And if the organoids did engraft in the liver, what was the topographical location of the organoids? eg which zone in the liver? This could be assessed with immunofluorescence staining of the livers across the various groups of mice.

15. Line 334, Fig 5c and Ext Fig 7b: authors state low adhesion U-plates readily generated assembloids within 24h, however Ext Fig 7b states D6. Fig 5f has no mention of culture time. Please clarify in results text, figures and legend.

16. Extended data 5d: The Thy-1 staining presented is not convincing.

17. Fig 5d: The assembloid images in this figure and multiple subsequent figures (eg Extended Fig 7f; Extended Fig8 b,d) are not convincing in terms of recapitulating the structure of in vivo portal tracts, ie cholangiocytes clearly forming central lumens and mesenchymal cells surrounding the bile duct structures. Many of the mesenchymal cells are distal to cholangiocytes in the assembloids.

Furthermore, in Ext data 7e the phalloidin staining is picking out many lumens which have no associated cholangiocytes, and in Ext data 7g PFs (portal fibroblasts) rarely co-locate with cholangiocytes.

18. Fig 5f: there are no scale bars on the lower images. Further, in panel 4 the phalloidin stain looks nuclear and overlaps with HNF4a. Why is this different to the rest of the manuscript and the expected expression of phalloidin?

19. Line 344: the authors state periportal assembloids retain hepatocyte functions, however only Alb secretion is assessed. Further types of hepatocyte function need to be assessed.

The authors also state that secretion of Alb is comparable to hepatocyte organoids alone, however is that with or without DM? As well as performing statistical analyses, the authors need to increase the 'n' for the assembloids.

Line 348 short term culture in Ext Fig 7f: length is not stated here – what is meant by short term?

20. Ext Fig 8a: bars show assembloids at day 1 and 6, however the figure legend states day 1 and 7.

21. Ext Fig 8 is largely descriptive. Please quantify cell death and proliferation. And to allow comparison with the results presented in Ext Fig 7f, the authors need to provide quantitation comparing the 'short term' and 'longer term' cultures.

What do the authors mean by 'architectural disorganisation'? Showing brightfield images does not make this clear. Perhaps F-actin or phalloidin staining over the time course would clarify this?

Furthermore, the authors should quantify assembloid function over time eg Alb secretion and other markers of healthy organoids.

22. Extended Fig 8a: The 'n' is very low for the assembloids (day 1 and day 6) in this figure (n=2) - more donor assembloids should be assessed in this dataset, ie the 'n' should be increased in these groups.

23. Extended Data Fig 8f (upper panel): the number of bile canaliculi per assembloid, across donors should be quantitated.

24. Fig 7h, Ext Data fig 10c - Ki67 immunofluorescence staining of mesenchymal cells and cholangiocytes should be performed and quantitated.

Referee #3

(Remarks to the Author)

The manuscript by Lei Yuan et al. presents a novel method for developing human periportal liver assembloids. The systematic exploration of pathway activators and inhibitors, supported by pathway analysis (IPA), is unbiased. The authors combined Wnt and Hippo signaling modulation in addition to ingredients from past methods to generate human hepatocyte organoid culture successfully. They established a living biobank and provided a detailed characterization of these organoids. Additionally, the authors developed methods to culture portal mesenchyme and integrated it with human liver organoids and cholangiocyte organoids. These efforts culminated in a platform capable of modeling certain aspects of biliary fibrosis. The statistics are used appropriately, and the conclusions are robust. The manuscript is written very well and easy to follow.

My concerns are below:

1- Long-term expansion of adult human hepatocytes as organoids (h-HepOrg) holds significant potential for modeling liver tissue and advancing regenerative medicine. However, it remains unclear how the current study addresses persistent challenges in this field and substantially advances our understanding of the underlying factors. Metrics such as high albumin levels or Apo proteins, while exciting, have been almost achieved in previous studies. The full maturation spectrum of hepatocytes still has not been reached in this work which has been a main target for many past studies. These are metrics such as P450 family enzymes including master drug metabolism enzyme CYP3A4. The level that is on par with primary human hepatocytes still not achieved as it is shown in the study.

2- Other studies have also attempted to tackle these challenges by using primary hepatocytes to expand culture methods, and they have also shown limited success. An experimental comparison with prior methods, such as those by Peng WC et al. (Cell, 2018) and Hu H et al. (Cell, 2018), would have provided a helpful reference point to know how much improvement in this study could be achieved. This is at the level of hepatocyte organoids as well as organoids with other Non-parenchymal cells (PFs). To my assessment the current system demonstrates some of the same limitations, including a degree of immaturity in drug metabolism / maturation markers like CYP3A4. However, it shows improvements in factors such as albumin production and cholestatic mechanisms.

*please also make sure that Peng et al is cited.

3- What is the starting hepatocyte population, and would the initial zonation of hepatocyte impact the outcome?

4- The addition of portal mesenchyme is a novel aspect. However, it is not clear if the presented outcomes could have been achieved with any other cells with myofibroblasts or fibroblasts origin. For example, they could use stellate cells with hepatocyte organoids as control and compare what would be special about this combination and how it can shed light on specific periportal-related phenotypes that are beyond a simple wound healing response. Hence, unfortunately while promising it still is unclear whether the model demonstrates specific events beyond a generic desmoplastic inflammatory reaction that could be achieved with other cells.

5- A parallel study is under review, almost the same title in mice from the same team and reported in the references #39.

While this study focuses on the human model, the differences between the mouse model and the human model reported here could be minimal and can take away from innovation of this study.

6- The MACS sorting of EpCAM-positive cells for the depletion of these precursors of biliary organoids should be validated through FACS analysis to ensure accuracy and reproducibility and remove any possibility of residual cells used for making organoids.

7- The efficiency of the method and inter-individual variations should be reported to provide a clearer understanding of the robustness and applicability of this approach.

Version 1:

Reviewer comments:

Referee #1

(Remarks to the Author)

The authors have addressed all my points. Well done!

Referee #2

(Remarks to the Author)

The authors have comprehensively addressed my comments, I only have one further minor comment (listed below). My congratulations to the authors on an outstanding body of work.

Minor comment:

Fig 2f: In the h-HepOrg-DM images, the upper and lower panels appear to be from different samples - it would be helpful to the reader to show combined (dual) immunofluorescence staining of GS and HAL in the h-HepOrg-DM, thereby highlighting the heterogeneity and spatial distribution of hepatocyte function within the same h-HepOrg-DM.

Referee #3

(Remarks to the Author)

While the manuscript has improved by addressing the comments, I have some reservations about the novelty and groundbreaking aspects of the work and some parts are not well developed yet.

1) there are similar studies in the literature, including the examples below that used a version to achieve similar goal and would be hard to substantiate the differences or superiority. Unfortunately this leaves me with dampened enthusiasm on innovation and significance.

Concerning Hep. expansion:

1. Hui et al., Cell Stem Cell (2018): "In Vitro Expansion of Primary Human Hepatocytes with Efficient Liver Repopulation" — Developed a defined medium to expand primary human hepatocytes to large quantities and proliferating hepatocytes exhibited both hepatocyte and progenitor features, could revert to mature phenotypes in organoid culture, and notably repopulated mouse livers with high efficiency.....

2. <https://www.biorxiv.org/content/10.1101/2024.06.10.598262v1.full.pdf>

"Mass Generation and Long-term Expansion of Hepatobiliary Organoids from Adult Primary Human Hepatocytes" by Spee, Bart et al.

3. www.biorxiv.org/content/10.1101/2024.12.28.630269v1.full.pdf

"Expandable, Functional Hepatocytes Derived from Primary Cells Enable Liver Therapeutics"

Concerning assembloids :

1. Nature (2025): "Mouse liver assembloids model periportal architecture and biliary fibrosis" (DOI: 10.1038/s41586-025-09183-9) — The authors of the current manuscript highlight some differences in Table 3, but in my view, these are limited, and the fundamental innovation of the model remains modest. While species-specific differences are important and justify developing human models, the claimed advances here appear incremental as the principle used to generate the self-organizing structure is similar via mixing the cell types in defined ratios.

2) The term "periportal triad" refers to a group of three key structures located at the periphery of hepatic lobules in the liver: the portal vein, hepatic artery, and bile duct with oxygenated blood. In the current work, the development of self-organizing periportal-like structures notably lacks endothelial vasculature and limited control on oxygen. These limitations need to be acknowledged in the Discussion.

3) A more critical issue is that the frequency and spatial organization of the (chol-PF-Hep) structures are not well characterized. The authors mention that "mesenchymal cells are closely associated with cholangiocytes," but are they really? From the images provided, the positioning of mesenchymal and Chol cells appears limited, and in some cases they happen to fall next to each other. Can the authors quantify how many bona fide periportal structures are formed, and what proportion of them show the correct configuration? The current data shows a weak association.

Perhaps given that single-cell data are available, the authors could also use their cell-cell communication to determine whether there is any biological basis or signaling logic suggesting that mesenchymal cells are recruited toward biliary lumens to improve the model rationale.

Furthermore, what percentage of the Chol cells actually form biliary lumens and what percent are left in parenchymal scattered? Addressing these points is very useful to evaluate whether the model partly includes the hallmark periportal-like and adds to the rigor of the model for people to depend and use.

4) Authors mention "Staining for the periportal hepatocyte markers SAA1 and SAA2 confirmed the spatially confined expression of these portal markers in the hepatocytes located within cholangiocyte rich, ECAD⁺ regions, with portal Mesenchyme (PFs-nRFP) in close proximity, further supporting the establishment of a periportal-like domain in the assembloids (Extended Data Fig. 9a)".

>>> However in Fig 9a control staining of hepatocyte organoids are missing without which it is hard to recognize the true signal. Also the tissue patterning as they state is weak.

5) It appears that when authors use 1D-Hep as the control, the medium lacks ITS (Insulin, transferrin, selenium widely used) and Dexamethasone. This importantly can compromise the quality of the control group and can affect the reported differences. A high-quality control is essential to demonstrate the CYP3A4 (or other cyps) differences convincingly. Additionally, there seems to be a discrepancy between the drug testing data and the CYP luc assay as one suggests the system performs on par with the control, while the other does not. I could not find an explanation for this inconsistency. Improving and clarifying this part of the data would make the findings more reliable.

--

1) Minor comment:

The authors state : "For our goal to reconstruct the physiological cell-cell interactions of the periportal region of the liver lobule it is a prerequisite that the hepatocytes are mature enough to interact and establish connections with the neighbouring ductal epithelium".

>> this is an assumption that is not backed up by data or literature.

Version 2:

Reviewer comments:

Referee #3

(Remarks to the Author)

I congratulate the authors as they have been able to provide new data and discussion and satisfactory addressed my earlier concerns about the remaining parts of the manuscript.

We sincerely thank the editor and Reviewers for their thorough evaluation of our manuscript and for the constructive and insightful comments provided. We greatly appreciate the time and effort invested. The suggestions have been extremely helpful in guiding us to improve the clarity, rigor, and overall quality of the manuscript.

In response, we have carefully addressed all of the Reviewer's points by conducting additional analyses and experiments, and have revised the manuscript accordingly to incorporate the Reviewers' feedback.

Before providing the detailed point-by-point response, please let us summarize the major revisions:

1) We have now **compared to published hepatocyte cultures**, including the hepatocyte organoid models from Peng et al., 2018 and Hu et al., 2018 and also primary human hepatocyte cultures as requested by *Reviewer#1_point 1, Reviewer#2_point 2, and Reviewer #3_point2*. Briefly, through side-by-side comparisons using tissues from the same donors, we show that our culture system is the only one that allows robust long-term expansion of human hepatocytes compared to methods from Peng et al., 2018; Hu et al., 2018; and other 2D-cited models. Our optimized h-HepOrg medium has enabled us to generate a biobank of up to 28 donors (New Extended Data Fig. 1j, and Supplementary Table 2). Additionally, we have compared the gene expression of our h-HepOrg to matching primary human hepatocytes and 1-day cultured PHHs and now show transcriptionally proximity to primary human hepatocytes (PHHs) (revised Fig. 2d).

2) As answer to *Reviewer#1_point 2, Reviewer2_point 11 and Reviewer #3_point2*, we now **prove patient-specific fidelity** over time: We show that even at late passages, hepatocytes maintain donor-specific gene expression profiles, complementing our observation from the first version of the manuscript regarding the variability in bile canaliculi (BC) architecture in organoids and patient cohorts (new Fig. 3d, new Extended Data Fig. 4e, and new Supplementary Table 4).

3) We demonstrate that **our model outperforms PHHs on p450-mediated drug metabolizing activity**. As response to the *Reviewer#1_point 1, Reviewer2_point 12 and Reviewer #3_point1*, we have now evaluated the drug metabolizing capability of our h-HepOrg using mass spectrometry analyses. We show that our differentiated hepatocyte organoids outperform conventional hepatocyte cultures in drug metabolism, retaining patient-specific activity (revised Fig. 3e, f and new Fig. 3g, revised Fig. 2d, and Extended Data Fig. 3g and 3i).

4) We now provide evidence that our **human periportal assembloids recapitulate portal-specific identity and function** (*Reviewer #2_point 13*): We show that periportal assembloids acquire portal-like gene expression and function, outperforming isolated hepatocyte organoids and primary hepatocytes from matching donors in portal functions (urea, glucose production), while showing reduced central

vein gene expression and central-vein functions (e.g. drug metabolizing activity) (new Figure 5i-k and new Extended Data Fig. 9a-b). Notably, we demonstrate that the structural fidelity of our organoid systems, albeit not perfect and lacking portal endothelium, mirrors the complex *in vivo* scenarios, without requiring xenotransplantation in mice.

Point by point response letter to the Reviewers:

We sincerely thank the editor and Reviewers for their thorough evaluation of our manuscript and for the constructive and insightful comments provided. The suggestions have been extremely helpful in improving the clarity, rigor, and overall quality of the manuscript.

Below, we provide a detailed point-by-point response, with the Reviewer comments in black regular font and our replies in blue and *italics*. In the revised manuscript, we also indicate the location of changes using track changes and highlighted in yellow.

Referee #1:

In this interesting manuscript entitled “Human periportal liver assembloids recapitulate periportal liver tissue in vitro” Meritxell Huch and her team describe the development of a human hepatocyte organoid model and its evolution into a periportal assembloid model, by adding additional cell types. The authors generated a biobank from different donors, carefully characterized their model and compared key physiological and pathophysiological features with patient tissue and primary counterparts of the cell types used. The manuscript is well written, the illustrations are beautiful, and conclusions are supported by solid data.

A: We sincerely thank Reviewer for the positive comments regarding our manuscript. We are grateful for the recognition of the novelty of our periportal liver assembloid model, the careful characterization, and the clarity of the presentation.

However, in my opinion the authors should better highlight the advance of the organoid model over previously published models, assess to what extent their models retain disease characteristics of the donor/source tissue and add more use case data relevant for drug discovery before publication.

A: We appreciate the Reviewer’s valuable suggestions and comments. Because these refer to the points below, namely point_1 (comparison of our model to existing ones), point_2 (disease characteristics of the donor) and point_6 (drug metabolism), we have answered them below in the corresponding points.

Major points:

1) The introduction is missing relevant publications describing proliferating primary hepatocytes (ProlIH cells from the Hui lab) and previously described hepatocyte organoids (from the Clevers and Nusse labs).

A: We apologize for this oversight and thank the Reviewer for pointing out the omission of the Nusse (Peng et al., 2018)¹ and Hui’s lab (Zhang et al., 2018)² publications from our introduction. We have now included these references in the revised manuscript (now references 34 and 18, respectively). The Clevers’ reference (Hu et al., 2018)³ was cited as reference #33 in the previous version and remains as reference #33 in the revised manuscript.

The authors should more clearly highlight how their organoid model represents an advance over other culture systems. I clearly see the advance of the assembloid system over mono-cellular systems. However, the hepatocyte organoid system itself requires more differentiation.

A: We thank the Reviewer for this valuable suggestion, which has helped to further strengthen the scope of our manuscript. To better highlight the advancements of our system compared to existing culture methods, we have now conducted comparative analyses of expansion potential, hepatocyte function, and gene expression between our HepOrg model and previously published organoid systems and hepatocyte cultures.

*First, we compared the expansion potential of our optimized culture conditions (h-HepOrg-EM1 and h-HepOrg-EM2) with those described by Peng et al., 2018¹ and Hu et al., 2018³, as recommended. Freshly isolated human hepatocytes were cultured in either our optimized media or in the published conditions. As shown in the **new Extended Data Fig. 1j**, robust (~20-fold) expansion of human hepatocytes was observed only in our optimized conditions (h-HepOrg-EM2). This is consistent with Hu et al., 2018³, who noted that adult hepatocytes could not be maintained beyond a few days in culture under their conditions. Consequently, only our h-HepOrg-EM2 allowed the long-term culture of these cells, as we had shown already in the previous version of the manuscript (Fig. 1f). These results, presented as a panel in **Extended Data Fig.1j**, are pasted below for the Reviewer’s convenience, **Figure R1.1**)*

Figure R1.1: Cell expansion comparison between h-HepOrg grown in our h-HepOrg-EM2 medium and previously reported hepatocyte cultures. Following hepatocyte isolation, a total of 12,500 cells were seeded in Matrigel and cultured in our h-HepOrg-EM2 or EM1 medium or in the medium published by Peng et al., or Hu et al., or in our MM medium supplemented or not with Wnt, TRULI. We also used as controls the conditions where the preparation was not MACS or the tissue was not perfused, as well as the EpCAM+ cholangiocyte fraction cultured in our cholangiocyte medium. The number of cells was counted on an automatic cell counter at the indicated time points (day 0, 10, and 15). We only observed robust cell expansion in either h-HepOrg grown in our EM2 medium or in the h-CholOrg grown in cholangiocyte medium from Huch et al 2015. Graph represents the mean +/- SD of n=3 independent donors with n=3 technical replicates from n=3 independent experiments. Statistical significance was determined by two-way ANOVA with Dunnett’s test for multiple comparisons was used comparing each condition to the h-HepOrg-EM2 group at the last time point (day15). The colour coding indicates

the comparison of our h-HepOrgEM2 medium to the respective medium in the legend. Error bars represent standard deviation ($n = 3$ donors, biological replicates).

In parallel, we compared the gene expression profiles of our expanded (h-HepOrg-EM2) and differentiated (h-HepOrg-DM) HepOrg with publicly available datasets. To facilitate this comparison, we leveraged datasets already compiled in Ardismita et al., 2022⁴ that include the human fetal HepOrg models from the Clevers laboratory, as cited by the Reviewer, as well as other hepatocyte culture systems and fresh isolated PHHs. As shown in Figure R1.2, we found by PCA that differentiated h-HepOrg samples (h-HepOrg-DM, red circles) and h-HepOrg in expansion (EM1/EM2, green circles) cluster closely with PHHs from other studies (turquoise triangles) as well as with fetal (purple triangles) and PSC-derived hepatic cells (pink triangles) from Hu et al., 2018³ and Wang et al., 2020⁵ respectively.

However, because these comparisons were not performed on samples from the same donors, we cannot fully exclude that donor variation may influence these results. Additionally, due to too strong batch effects, we could not use all the samples from our manuscript, mostly only the samples from the first sequencing batch, hence why there is only 2 samples in DM condition. Therefore, due to these 2 caveats, we have opted to present these results below for the Reviewer's consideration only (Figure R1.2, for Reviewer only). Should the Reviewer or Editor feel that inclusion of these findings would further strengthen the manuscript, we would be pleased to incorporate them accordingly.

Figure R1.2, for the Reviewer only: Comparison between our h-HepOrg models and published human culture systems. PCA was performed on gene expression profiles from our study (circles), including h-HepOrg in expansion medium (h-HepOrg-EM, green) or DM (h-HepOrg-DM, red) or h-CholOrg cultured in our cholangiocyte

medium (grey), and in the publicly available datasets compiled in Ardisasmita et al., 2022 (triangles), which include studies containing primary human hepatocytes (PHHs, turquoise), Common Bile Duct tissue (CBD, light yellow), Liver tissue (liver, dark yellow), Fetal hepatocytes (Fetal_Hep, light green) and pluripotent stem cells (PSC, dark purple) and corresponding hepatic cultures of these (HLC, hepatic liver cells) including Cholangiocyte organoids (Chol_HLC, blue), Fetal hepatocyte organoids (FHep_HLC, light purple), hepatocytes in monolayer (Hep_HLC, grey), HepG2 cells (orange) and PSC-derived hepatic liver cells (PSC_HLC, pink). Each point represents an individual sample in each study. Colours indicate different conditions and can include more than one study. Note that many samples from our study, both in EM and DM, cluster close to the PHHs (turquoise) from many studies, as well within the Fetal hepatocyte organoids (FHep-HLC, light purple) and the PSC-derived HLC from Wang et al., 2020⁵. Due to too strong batch effects, we could only use the samples from our first sequencing batch, hence why there is only 2 samples in DM condition. P, passage.

Most other hepatocyte organoid systems lose metabolic function and suffer from biliary metaplasia as well as EMT. The present hepatocyte organoid system should be compared to other similar systems in these aspects.

A: We thank the Reviewer for this very valuable suggestion, which has significantly contributed to improving the quality of our manuscript. To address the question regarding metabolic functionality, we have now directly compared hepatocyte function (Albumin secretion, CYP activity, and drug metabolism) between h-HepOrg and matched primary human hepatocytes (PHHs) cultured in standard 2D conditions for 1 or 7 days.

As mentioned above, a direct comparison with the published mouse (Nusse¹) and human foetal (Clevers³) HepOrg models was not possible, as we were unable to expand these cultures under the reported conditions.

Our analyses revealed that h-HepOrg differentiated in DM secreted comparable amounts of Albumin to matched PHHs cultured for either 1 or 7 days. Similarly, differentiated h-HepOrg showed CYP2C9 activity levels equivalent to 7-day PHHs, and slightly lower CYP3A4 activity, although 1-day fresh PHH cultures outperformed both for these activities (revised Fig. 3e, f).

Importantly (as we further detail below in point #6 from this Reviewer), mass spectrometry analysis demonstrated that our differentiated h-HepOrg significantly outperformed PHHs from the same donors in their drug metabolizing capacity for Verapamil (new Fig. 3g). These functional results align with our transcriptomic analyses, which indicated that differentiated h-HepOrg and 1-day cultured PHHs express similar levels of key metabolizing enzymes, including CYP2E1, CYP2C9, CYP1A2, CYP2C8, CYP3A4, and CYP3A5—the latter three being responsible for the N-demethylation of Verapamil into Norverapamil, the major in vivo metabolite (revised Fig. 2d, and Extended Data Fig. 3g and 3i). These new results are now presented in the revised manuscript (revised Fig. 3e, f and new Fig. 3g, revised Fig.

2d, and Extended Data Fig. 3g and 3i), and we have combined them below in (Figure R1.3) for the Reviewer's convenience.

Figure R1.3: Functional and transcriptional characterization of human hepatocyte organoids cultured in differentiation medium. **a-b.** Human hepatocyte organoids expanded in EM2 (passage 1-7) and differentiated in DM were tested for functional cytochrome activity (a) and albumin secretion (b) and compared to human liver cholangiocyte organoids (h-CholOrg) and fresh isolated primary human hepatocytes (PHH) cultured as standard 2D monolayers for 1 or 7 days. Graphs represent mean +/- SEM for n=4-7 donors from n= 3 independent experiments. Results are expressed as RLU (a) or ng/ml (b) normalized by the total cell count. Two-way ANOVA with Tukey test for multiple comparisons was used. **c.** The production rate of norverapamil reflects the xenobiotic metabolism capacity of Hepatocytes in HepOrgs. The rate (in pmol/h per 10³ cells) was determined by mass spectrometry for h-HepOrg and primary human hepatocytes (PHHs) obtained from 3 donors. For h-CholOrg (used

as a negative control), the production rate was below the detection limit. Bars represent mean values with SD ($n = 3$). Statistical significance between PHHs and h-HepOrg was assessed using an unpaired two-tailed Student's *t*-test with Welch's correction, comparing biological replicates ($n = 3$ donors per group). **d.** Heatmap representing the scaled expression of the indicated genes for known hepatocyte and cholangiocyte (Chol.) markers. Their region-specific expression (pericentral or portal) or function is indicated in different colours. **e.** Heatmap showing the expression of the full list of genes from the drug metabolism datasets from the GSEA analysis between h-HepOrg grown in DM compared to EM2 medium, and their expression in fresh isolated hepatocytes (Primary, PHHs) and in cholangiocyte organoids (h-CholOrg). Note the similarity between DM and primary hepatocytes. Column, donor. **f.** CPM values from the RNAseq for the hepatocyte marker Albumin (ALB), the mature hepatocyte marker and metabolizing enzyme CYP3A4, CYP3A5, CYP2C8 and CYP2C9, the bile salt transporter MRP2, the cholangiocyte marker KRT19 and the hepatoblast marker AFP. Note that upon DM the expression of mature hepatocyte markers increases while the expression of the cholangiocyte and foetal marker decreases. Graphs present the CPM values mean \pm SEM from $n=3$ different donors. Dot, independent donor.

Most other hepatocyte organoid systems lose metabolic function and suffer from biliary metaplasia as well as EMT. The present hepatocyte organoid system should be compared to other similar systems in these aspects.

*A: This point is related to point #5 below. We thank the Reviewer for his/her suggestion to check markers of biliary metaplasia. In response, we have **revised Fig. 2d** to assess the expression of SOX9 and KRT7, which are more specific and established markers of biliary metaplasia, as indicated by the Reviewer in point #5. Our analysis reveals that KRT7 levels are lower in h-HepOrg cultured in DM compared to EM, while SOX9 expression shows patient-dependent variability, although in all cases their expression was higher than PHHs. These updated data are now included in the **revised Fig. 2d** and are further supported by **Figures R1.3d and R1.8**.*

*Nonetheless, and despite this relative expression of biliary markers, we would like to highlight the findings presented in **Figure R1.2, for Reviewer only**, which indicate that our h-HepOrg, both in EM and DM, consistently cluster closely with primary human hepatocytes (PHHs) from other studies (turquoise triangles), as well as with foetal (purple triangles) and PSC-derived hepatic cells (pink triangles) from Hu et al., 2018³ and Wang et al., 2020⁵, even across various passages (P3, P7, and P10). Therefore, these results suggest that our system maintains hepatic identity, even during prolonged in vitro culture.*

Regarding the question of epithelial-to-mesenchymal transition (EMT), we have now compared the expression of validated EMT markers in our HepOrg cultures at early (P1) and late (P7–P10) passages, alongside primary hepatocytes (PHHs) cultured in 2D. Portal fibroblasts were used as positive controls. Due to time constraints during this revision, we were unable to perform these analyses on matched PHHs and therefore utilized publicly available datasets from 2D-cultured PHHs. Overall, we observed

consistent expression of the classical epithelial marker CDH1 across all samples, irrespective of passage number, confirming that the cells maintain their epithelial identity even after extended expansion in vitro. Expression of the majority of EMT markers was negligible, with the exception of Snail2. For Snail2, we noted patient-specific variability, with some donor lines (PHH-14 and PHH-13) showing slight upregulation at late passages, while others (PHH-16) did not. Importantly, even in cases of upregulation, Snail2 expression levels remained significantly lower than those observed in bona fide mesenchymal cells. We have opted to present these results below for the Reviewer's consideration only (Figure. R1.4, for Reviewer only). Should the Reviewer or Editor feel that inclusion of these findings would further strengthen the manuscript, we would be pleased to incorporate them accordingly.

Figure R1.4, for the Reviewer only: Analysis of EMT marker expression in HepOrg cultures from early (P1/P2) and late (P10) passage and in Primary human hepatocytes cultured for 1 day in monolayer.

Taken together, our results indicate that our model represents a significant advance compared to published human hepatocyte culture models, including the organoid models from the Nusse¹ and Clevers³ lab as:

(i) it allows the long-term expansion of primary human hepatocytes from fresh adult patient tissue, allowing us to generate millions of cells over short period of time from 28 patients while retaining gene expression similar to PHHs without showing clear signs of biliary metaplasia or EMT over time.

(ii) the expanded h-HepOrg secreted Albumin similar to existing PHH cultures and retained cytochrome activity over time. Remarkably, h-HepOrg metabolizing capabilities outperformed current hepatocyte culture models used for toxicity assays, at least for the drug tested.

To comprehensively highlight the advance of our model compared to published organoid models and ProlHH model we provide a detailed side-by-side comparison of our model with relevant previously published systems, in the new Supplementary Table 3 of the revised version of the manuscript, which we hope emphasises the unique features and advantages of our HepOrg and assembloid model.

We hope these additional analyses satisfactorily address the Reviewer's concerns.

2) Do patient-derived organoids retain functional characteristics of the diseased source liver or do they get rectified over time in culture?

A: To address whether patient-derived h-HepOrgs retain functional and disease-related characteristics of the source liver tissue, we have extended our previous transcriptomic profiling (RNA-seq) to include additional samples from patients recruited during this round of revision. We compared organoids to their source hepatocytes and observed that organoids derived from different donors maintained distinct molecular signatures that reflect inter-patient variability.

Notably, many patient-specific genes expressed both in the organoids and their corresponding source cells have been previously associated with liver-related pathologies, including susceptibility to hepatitis virus infection (e.g., IL1RL1, ERAP2), liver cancer (e.g., GPC3), and cholestasis during pregnancy (e.g., GABRP). Additionally, several donor-specific genes were involved in key metabolic pathways, such as glutathione metabolism (GSTM3, GSTM1), lactate metabolism (LDHC), and lipid metabolism (APOA4, FAR2, ACSM1), among others (New Fig. 3d and New Supplementary Table 4).

These results suggest that *h-HepOrgs* retain stable, donor-specific molecular characteristics rather than undergoing normalization (rectified) over time in culture, at least up to passage 7. These findings are presented in new Fig. 3d, Extended Data Fig. 4e, and Supplementary Table 4, and are discussed in the Results section (Line 289-303). For the Reviewer's convenience, we have also pasted the relevant figures and data below.

Figure R1.5: Hepatocyte organoids retain donor-specific transcriptional signatures that recapitulate inter-donor variability observed in primary hepatocytes. **a.** Heatmap showing scaled expression of donor-specific genes computed across primary human hepatocyte (PHH) samples and hepatocyte organoids cultured in differentiation medium (*h-HepOrg-DM*). Columns represent individual donors, and hierarchical clustering was performed on both samples and genes. Each row corresponds to a donor-specific gene identified by differential expression analysis. The expression profiles of *h-HepOrg-DM* (yellow/brown) samples exhibit high concordance with the inter-donor variability observed in PHH samples (purple), indicating that hepatocyte organoids retain donor-specific transcriptional signatures. **b.** Spearman correlation heatmap showing pairwise similarities between primary hepatocyte samples and differentiated hepatocyte organoids. Donor-specific genes were first identified from fresh primary hepatocytes, and these genes were then used to calculate pairwise correlations with gene expression profiles from organoids. Warmer colours indicate higher correlation coefficients, representing stronger transcriptomic similarity.

Were the primary hepatocytes used in the profiling experiments source material for *h-HepOrgs*?

A: We apologize for not having made this point clear in the previous version of the manuscript. We confirm that, for at least four different donors (PHH20, PHH16, PHH2, and PHH3), the primary hepatocytes profiled were indeed the same cells used as the starting material for generating the corresponding *h-HepOrgs*. As mentioned above, we have now expanded this analysis with additional patient samples recruited during this revision, increasing the number of donors to six for improved

statistical robustness ($n=6$). We have now clarified that in the text in the Results section (Line289-293 and figure legend 3d).

Moreover, what if mesenchyme from different patients is used, does this have disease-related implications to the epithelial assembloid part?

A: To assess the impact of mesenchymal origin on the epithelial component of the assembloids, we generated assembloids in which the epithelial and mesenchymal cells were derived from independent donors. We observed no significant differences between assembloids formed with cells from the same donor and those formed with epithelial and mesenchymal cells from different donors. These findings suggest that, at least when using healthy mesenchyme, the mesenchymal source does not substantially influence the epithelial compartment of the assembloids. However, we cannot exclude the possibility that mesenchyme derived from diseased tissue could have a different effect on the epithelial behaviour. These results are described in the Results section (Line 387-389) and are presented in the revised Fig. 5f, middle panel and new Extended Data. 7e and also pasted combined below for the Reviewer's assessment.

Figure R1.6: Assembloid formation is independent of the donor source of healthy portal fibroblasts. Representative confocal images of periportal assembloids cultured for 3 days, formed using epithelial and mesenchymal cells from either matching donor cells (the same-donor, left) or not matched cells (cross-donor, right). Nuclei are stained with DAPI (blue); cell membranes are labelled with Phalloidin (white). No major morphological differences were observed between conditions. Scale bars, 50 μ m.

We hope these three answers combined clarify the robustness and fidelity of our model in capturing patient-specific features.

3) Figure 1b and SuppFig 1: The terminology related to the YAP/Hippo pathway is confusing to me. Usually people refer to the Hippo pathway and YAP signaling. Likewise, LATS1 mRNA downregulation does not inform on YAP signaling status. The authors should check YAP target genes to assess that

status of the pathway. Also, the authors should better talk about activating YAP signaling rather than blocking HIPPO signaling, to not confuse the reader.

A: We are deeply grateful to the Reviewer for this valuable suggestion, which has significantly improved the rigor and clarity of our study. We have implemented comprehensive revisions throughout the manuscript to address this concern.

As recommended, we have systematically revised our terminology to describe "YAP signaling activation" rather than "Hippo pathway inhibition" in all relevant sections, ensuring consistency with established field conventions. We also fully acknowledge the Reviewer's important point that LATS1 mRNA downregulation alone is insufficient to confirm YAP activation. In response, we conducted additional qPCR analyses of multiple canonical YAP target genes (including ANKRD1, CYR61, and AMOTL2) following 12 hours of treatment with the LATS1/2 inhibitor.

*These new results, presented in the **new Extended Data Fig. 2j** (also pasted below for the Reviewer's convenience, **Figure R1.7**), provide direct evidence of YAP pathway activation following LATS1/2 inhibition by TRULI in our model. We have carefully updated the main text and corresponding figure legends to incorporate these findings and ensure accurate interpretation.*

We sincerely thank the Reviewer for helping us improve the precision and completeness of our study.

Figure R1.7: Quantitative RT-PCR analysis of YAP downstream target genes ANKRD1, CYR61, and AMOTL2 in the presence (+TRULI, orange) or absence (-TRULI, grey) of TRULI. Gene expression levels are normalized to HPRT and presented as $2^{-\Delta Ct}$. Data are shown as mean \pm SEM ($n = 6$), and significance was assessed using an unpaired two-tailed Student's *t*-test. *P*-values are indicated.

4) Since TRULI is a rather unspecific kinase inhibitor, hitting many other kinases in addition to LATS1/2, one cannot refer its effects solely to YAP/HIPPO signaling. There are more specific LATS kinase inhibitors out there (NIBR-LTSi) but noting limitations of TRULI may be sufficient, rather than repeating experiments. In addition, this statement should be refined, as the data shows modulation of the respective pathways but does not indicate specificity of the drugs: "Gene set enrichment analysis

showed that h-HepOrg exhibited high expression of Wnt and YAP targets, indicating that the Wnt agonist and the LATS1/2 inhibitor used act specifically through these respective pathways.”

A: We sincerely appreciate the Reviewer’s excellent observation regarding the pharmacological specificity of TRULI. We fully agree that TRULI may inhibit kinases beyond LATS1/2, and thus the observed effects cannot be solely attributed to modulation of the YAP/Hippo pathway. In line with the Reviewer’s suggestion, we have revised the manuscript to acknowledge this limitation and have replaced the original statement with the following:

(New line 209-213): “While we confirmed YAP activation by qPCR following 12 hours of TRULI treatment (Extended Data Fig.2j), it is known that TRULI inhibits kinases beyond LATS1/2. Therefore, we cannot exclude the possibility that off-target effects may also contribute to the observed cell growth effects”.

5) SOX9 and KRT7 should be added to Figure 2D since they are better markers to show biliary metaplasia in hepatocytes compared to the more mature biliary markers (KRT19 and EPCAM) used in the heatmap. Moreover, EMT is key issue for primary hepatocytes in culture since under normal 2D culture conditions they become fibroblast like and lose metabolic function. It would therefore be great if the authors could look into this in their model and compare to other systems in addition to biliary metaplasia.

A: As indicated by the Reviewer, both SOX9 and KRT7 are better markers of biliary metaplasia in hepatocytes. Accordingly, we have now revised Fig 2d to include SOX9 and KRT7 to investigate whether our h-HepOrg are undergoing biliary metaplasia. While we observe lower levels of KRT7 in h-HepOrg in DM compared to EM, we found that SOX9 expression levels are patient-dependent. These results are presented in the revised Fig. 2d and also pasted below for the Reviewer (Figure R1.8).

Regarding the EMT question, as addressed above in Point 1 of this Reviewer’s comments, we compared validated EMT marker expression in early (P1) and late (P7–P10) passage HepOrg cultures, alongside 2D-cultured PHHs and portal fibroblasts as controls. CDH1 expression remained consistent across passages, indicating preservation of epithelial identity. While most EMT markers were negligible, we observed slight, patient-specific upregulation of Snail2 at late passages, although expression levels remained markedly lower than in bona fide mesenchymal cells. These findings have been presented above for the Reviewer’s consideration (Figure. R1.4), and, as noted, we would be pleased to incorporate them into the manuscript should the Reviewer or Editor deem it appropriate.

Figure R1.8: Gene Expression of selected genes in h-HepOrg, h-CholOrg, 1-day PHH cultures and fresh isolated PHHs. Heatmap representing the scaled expression of the indicated genes for known hepatocyte and cholangiocyte (Chol.) markers. Their region-specific expression (pericentral or portal) or function is indicated in different colours. Name indicated donor. Note that the donors PHH2, PHH3, DSD48, DSD49 have matching samples in all conditions.

6) An important use case for physiological hepatocyte-like system is testing of drug metabolism. It would be great if the authors could compare metabolism for few drugs in their h-HepOrgs versus assembloids versus primary hepatocytes.

A: We thank the Reviewer for this valuable suggestion, which has helped to broaden the scope and strengthen the impact of our manuscript. In response, we have now expanded the p450-activity assays to directly compare p450-mediated drug metabolism between h-HepOrg and matched primary human hepatocytes (PHHs). We selected the drug Verapamil, a L-type calcium channel blocker with antiarrhythmic, antianginal, and antihypertensive activity, because it is extensively metabolized by the liver (up to 80%) via the cytochrome P450 enzyme system. Norverapamil, the major in vivo metabolite, is the result of verapamil's N-demethylation via CYP2C8, CYP3A4, and CYP3A5, and carries approximately 20% of the cardiovascular activity of its parent drug. CYP2D6 and CYP2E1 have also been implicated in the metabolic pathway of verapamil, albeit to a minor extent⁶. Remarkably, mass spectrometry analysis demonstrated that our differentiated h-HepOrg significantly outperformed PHHs from the same donors in their p450-mediated drug metabolizing capacity for Verapamil (new Fig. 3g). These functional results align with our transcriptomic analyses, which indicated that differentiated h-HepOrg and 1-day cultured PHHs express similar levels of key metabolizing enzymes, including CYP2E1, CYP2C9, CYP1A2, CYP2C8, CYP3A4, and CYP3A5—the latter three being responsible for the N-demethylation of Verapamil into Norverapamil (revised Fig. 2d, and Extended Data Fig. 3g and 3i).

In contrast, assembloids exhibited lower overall drug-metabolizing activity, which we hypothesize reflects their more periportal-like hepatic identity (New Extended Data Fig. 9b). This interpretation is supported by our new transcriptomic analysis (Fig. 5i), which shows that differentiated Hepatocyte Organoids (h-HepOrg-DM) are enriched in pericentral-associated gene expression profiles, whereas assembloids, that are also cultured under DM conditions, display enhanced periportal gene expression and periportal functional features, including gluconeogenesis and urea cycle activity (New Fig. 5j-k). We believe these findings highlight the value of the periportal assembloid system, where the addition of portal region cells (ductal cells and portal mesenchyme) induces a portal-like identity in hepatocytes that would otherwise exhibit a more central vein-like phenotype.

These new results have been incorporated into the revised manuscript in the section for hepatocyte organoids (Fig. 3g) and in the section for assembloids (Fig. 5i-5k, Extended Data Fig. 9b). Below, we have combined the results to facilitate the Reviewer's assessment and the comparison between HepOrg, assembloids and primary hepatocytes (Figure R1.9). We believe this additional analysis not only addresses the Reviewer's request but also provides novel evidence that our platform captures both metabolic competence and physiologically relevant zonation.

Figure R1.9: Compared to Primary Human Hepatocytes (PHHs), h-HepOrg present superior drug metabolizing capabilities while periportal assembloids exhibit enhanced periportal function. **a.** Heatmap showing expression of liver zonation genes in h-HepOrg (h-HepOrg-DM) and in hepatocytes from assembloids. Note that both, h-HepOrg and Assembloids are cultured in DM medium and while h-HepOrg present enhanced expression of central vein markers, including several p450-family drug metabolizing enzymes, hepatocytes from assembloids show clear enrichment of periportal markers. **b-c,** Urea and glucose production in assembloids (brown), hepatocyte organoids (purple), PHHs (pink) and h-CholOrg (green). **b.** Assembloids cultured for 5 days (brown) were used to analyse urea synthesis compared to hepatocyte organoids cultured in differentiation medium (DM, purple), freshly isolated primary human hepatocytes cultured in monolayer for 24 h (PHHs, pink), and CholOrg (green) as a negative control. Data represent mean \pm SD, from $n=3$ independent donors. **c.** Assembloids were cultured for 6 days prior to gluconeogenesis analysis and compared to hepatocyte organoids cultured in differentiation medium (DM), and CholOrg as a negative control. Data represent mean \pm SD, from $n=3$ independent donors. **d.** Mass spectrometry analysis was used to detect Norverapamil, the primary metabolite of Verapamil, in hepatocyte organoids compared to assembloids (6-day culture). For h-CholOrg (used as a negative control), the production rate was below the detection limit. Left, comparison between hepatocyte organoids and assembloids. Right, comparison between hepatocyte organoids and primary hepatocytes. Bars represent mean values with SD ($n = 3$). Statistical significance between PHHs and h-HepOrg was assessed using an unpaired two-tailed Student's *t*-test with Welch's correction, comparing biological replicates ($n = 3$ donors per group).

Importantly, we also observed inter-donor variability in Verapamil metabolism, suggesting the presence of patient-specific metabolic phenotypes and highlighting the potential of our h-HepOrg platform for personalized drug metabolism studies. However, we fully acknowledge that

substantiating such claims would require testing a broader range of drugs across a larger cohort of donor tissues. We therefore refrain from making definitive patient-specific claims regarding drug toxicity in the current manuscript and suggest that this important question be explored in future studies.

Minor points:

7) Fig5e – it would be great if the authors could provide information of the % composition in a patient liver for comparison.

A: We sincerely appreciate the Reviewer's constructive suggestion. To address this point, we quantified the cellular composition (% of total cells) in patient-derived liver tissue samples and compared it with our in vitro model. As shown in the newly added Extended Data Fig. 7a, patient liver tissue comprises approximately 15% ductal cells (DCs), 8% portal fibroblasts (PFs), and 75% hepatocytes (Heps), whereas our in vitro system (Fig. 5e) exhibits 8% DCs, 5% PFs, and 90% Heps. Although our in vitro model displays a slightly higher proportion of hepatocytes, the relative ratios of DCs and PFs are broadly consistent with those observed in tissues. This comparison further supports the physiological relevance of our system in capturing key cellular interactions. We have clarified this point in the revised manuscript (Lines 377–378) and have also pasted the relevant figures below for the Reviewer's convenience (Figure R1.10).

Figure R1.10: Cell composition in human liver tissue. To determine the % composition of hepatocytes (Hep), ductal cells/cholangiocytes (DCs) and portal fibroblasts (PFs) to test for the generation of assembloids, human liver tissue was stained for the cholangiocyte marker Pan-Cytokeratin (PanCK) and PF marker CD90. The results are presented as bar plots showing the % of Ductal cells (DCs), PFs and Hepatocytes (Hep) from n=3 donors. Dot, one independent portal region. Dot colour, independent donor.

8) Are patients 1-3 and organoids A-B in Extended data Figure 4 related? The data is not clear in whether the organoids retain the patient characteristics or whether there are just different types of BC in general in both conditions.

A: We thank the Reviewer for this important question. In the referenced figure, the tissue and organoid samples are not directly matched. As described in the Methods section of the manuscript ([Section Image analysis]), the bile canaliculi (BC) staining protocol requires immediate fixation of the tissue in the operating room (OR), followed by rapid quenching of the fixative. This step is essential to preserve native BC structures and prevent fixation artifacts, as previously described⁷. Unfortunately, this procedure is incompatible with the requirements for hepatocyte organoid generation, which necessitates fresh tissue subjected to cold perfusion in the OR prior to collagenase digestion.

Our established SOPs for sample collection do not allow to obtain both samples simultaneously at present and amending them was not possible during the timeframe of this revision. Therefore, it was not feasible to collect both types of samples from the same donor at once during this revision. Consequently, we have limited our assessment of patient-to-patient variability to the RNA-seq analysis detailed in our response to Point 2 of this same Reviewer. To avoid any confusion, we have revised the text and discussion (Lines 286–287 and 529-532) to clarify that our model captures different types of BC structures observed across patients, rather than directly matching tissue and organoid from the same individual.

(Line 286-287): “suggesting that our model could capture different types of BC structures that are observed in patient tissue cohorts”.

(Line 529-532): “We also observed variability in bile canaliculi (BC) morphology among organoids derived from different donors. However, whether this reflects true patient-to-patient differences remains to be determined and will require further investigation”.

Referee #2 (Remarks to the Author):

This manuscript by Yuan, Dawka, Kim and Liebert et al. describes the development of human hepatocyte organoids from adult human donor livers, followed by the development of multi-lineage human organoids containing hepatocytes, portal mesenchyme and cholangiocytes. The authors claim that these liver assembloids retain the histological arrangement, gene expression and cell-cell interactions observed in vivo in human periportal liver tissue - however I do not think the data presented supports these claims, ie that these liver assembloids closely recapitulate and mimic the topography, architecture and function of human portal tracts.

A: We would like to sincerely thank the Reviewer for their careful and constructive evaluation of our manuscript. We greatly appreciate the time and effort dedicated to providing valuable feedback that has helped us to critically reassess and strengthen our study. We understand and fully respect the Reviewer's concerns regarding the claims about the recapitulation of human periportal liver architecture and function in our assembloid model. These are important points, and we agree that it is essential for the characterization of our system to be presented with greater clarity and more robust substantiation.

In response, we have carefully revisited the data and further refined our analyses to more accurately describe the features of our human organoid and assembloid models, and how they align with in vivo human periportal liver tissue. As detailed below in our point-by-point response, we have expanded the structural comparisons to tissue, incorporated additional functional assays reflecting portal region-specific functions (including gluconeogenesis and ureagenesis), and provided direct comparisons with existing models to highlight the strengths and advances of our system. We have also explicitly discussed the limitations of our model and clarified the extent to which the assembloids mimic the periportal liver architecture.

We believe these revisions, and in particular the ones in response to point 5, point 13, point 17, and point 19, substantially strengthen the manuscript and more thoroughly address the Reviewer's concerns. Below, we provide a detailed point-by-point response to each of the Reviewer's comments.

I have a number of concerns regarding this manuscript which I have listed below:

1. The first few sections of this manuscript describe how the authors optimised their protocols to allow long-term expansion of human hepatocyte organoids. Although this is of course important

background work, a large amount of results text was used to describe the optimisation process. I think it would be helpful to the reader to shorten the amount of results text currently used to describe what is essentially method optimisation, and be more concise in the description of how the human hepatocyte organoid expansion protocols were iterated.

A: We have revised the manuscript to streamline the description of the human hepatocyte organoid expansion protocol. Specifically, we have condensed the methodological iterations into a more focused narrative, retaining only the key steps that directly contributed to the final protocol. Specifically, the detailed optimization of the different components tested have been moved to the Methods section ([Section Hepatocyte organoid culture]), while the iterations on cell viability and tissue isolation are detailed in the Methods section (Isolation of primary human hepatocytes and cholangiocytes) and in the revised Supplementary Table 1 and 2, for readers interested in technical specifics.

We believe these changes enhance readability while maintaining scientific rigor, and we thank the Reviewer for this valuable suggestion.

2. Fig 1c and Ext Fig 1a+b: It would be helpful to add cell viability graphs to accompany these data.

A: We thank the Reviewer for the suggestion. In response, we have conducted additional analyses to assess cell viability under the culture conditions presented in Figure 1c and Extended Data Figures 1a-b. As part of the revision process, we expanded our analysis to include additional culture conditions and time points. Specifically, we quantified both cell numbers and viability, as recommended.

Cell counts were performed at multiple time points to evaluate growth dynamics, and a viability assay (CellTiter-Glo® 3D Cell Viability Assay) was employed to assess cell survival under each condition. The results were consistent with the morphological observations and proliferation trends shown in the original version of the manuscript (Fig. 1c and Extended Data Figs. 1a-b). Notably, amongst all the media and conditions tested, our HepOrg-EM2 medium led to a 17-fold increase in hepatocyte expansion within the first 15 days of culture. These data confirm that the culture condition we selected supports both robust expansion and high viability of the hepatocyte organoids.

We have now included the cell count quantification in the revised Extended Data Fig. 1j (pasted below in Figure R2.1b for the Reviewer's assessment) and referenced it appropriately in the text (line 182). Additionally, for the Reviewer's more comprehensive view of the findings, we have also included below the quantitative cell viability data (Figure R2.1a for the Reviewer only).

We hope that this additional dataset addresses the Reviewer's concerns.

Figure R2.1: Cell expansion comparison between h-HepOrg grown in our h-HepOrg-EM2 medium and previously reported hepatocyte cultures. **a-b.** Following hepatocyte isolation, a total of 12,500 cells were seeded in Matrigel and cultured in our h-HepOrg-EM2 or EM1 medium or in the medium published by Peng et al., or Hu et al., or in our MM medium supplemented or not with Wnt, TRULI. We also used as controls the conditions where the preparation was not MACS or the tissue was not perfused, as well as the EpCAM+ cholangiocyte fraction cultured in our cholangiocyte medium. **a.** Cell viability was assessed using the CellTiter-Glo® 3D Cell Viability Assay at days 5, 10, and 15 of culture, starting with 12,500 cells per condition. Distinct media conditions (as described in the Methods) were applied to primary h-HepOrgs at passage 0 (P0). Viability is represented as ATP concentration (μM). **b.** The number of cells was counted on an automatic cell counter at the indicated time points (day 0, 10, and 15). We only observed robust cell expansion in either h-HepOrg grown in our EM2 medium or in the h-CholOrg grown in cholangiocyte medium from Huch et al 2015. Graph represents the mean \pm SD of $n=3$ independent donors with $n=3$ technical replicates from $n=3$ independent experiments. Statistical significance was determined by two-way ANOVA with Dunnett's test for multiple comparisons was used. comparing each condition to the h-HepOrg-EM2 group at the last time point (day15). The colour coding indicates the comparison of our h-HepOrgEM2 medium to the respective medium in the legend.

3. Ext Fig 1e, line 156: the authors should show quantitation and statistics regarding these data as they state significance in the results text ('resulted in significant hepatocyte organoid growth').

A: Upon review, we realize that the original phrasing in the manuscript could have been clearer. The original sentence stated: "None of the other tested components resulted in significant hepatocyte organoid growth". We realize that this sentence could potentially cause confusion, as it implies statistical significance without providing the necessary quantification. To address this, we have revised the sentence in the manuscript for greater clarity as follows:

(Line 163-164): "None of the other tested pathway regulators, resulted in consistent or quantifiable increase in hepatocyte organoid growth (Extended Data Fig. 1f)".

We believe this modification resolves the potential ambiguity while maintaining the integrity of our results.

4. Ext Fig 1h, passage information would be useful to include in this graph, similar to that shown in Fig 1f.

A: As suggested, we have now incorporated the passage details into the figure panel, following the same format as in Fig. 1f. This addition has been clearly labeled in the revised figure and legend (now moved to Extended Data Fig. 1i), and is also pasted below for the Reviewer assessment:

Figure R2.2: Graph showing the expansion potential of h-HepOrg from multiple donors in EM1 (complete, MM+Wnt+Truli) and in EM1 with the removal of individual components as indicated. Note that removal of Nic increased the longevity of the cultures, for all the donors tested. This medium is subsequently called h-HepOrg-EM2. Dot, passage.

5. Fig 2f. Regarding the h-HepOrg panels - are these different patients? I.F. markers are not consistent between tissue and hHepOrgs eg Alb and ECAD. It would help the reader if the markers / stains were consistent across all 3 panels.

A: We appreciate the feedback and agree the inconsistency in the markers can lead to confusion and does not allow a quick and easy comprehension. To improve consistency, we have used a single marker for the periportal zone, Histidine ammonia-lyase (HAL), for both h-HepOrg and human liver. We also keep a single marker for the pericentral zone, Glutamine synthetase (GS), for both h-HepOrg and human liver tissue. Therefore, now to increase simplicity, we have reorganized the panel in the revised Fig. 2f, also copied below as Figure R2.3 for the Reviewer's convenience.

Figure R2.3: Human liver tissue and h-HepOrg in DM stained for pericentrally and periportally zoned liver markers. Left, representative images of human liver tissue stained for the pericentral marker Glutamine synthetase (GS) (magenta, top) and periportal marker, Histidine ammonia-lyase (HAL) (yellow, bottom). Right, h-HepOrg in DM stained for the pericentral marker GS (magenta, top) and periportal marker HAL (yellow, bottom). Nuclei were stained with DAPI (cyan). Left panels, fluorescence intensities are indicated in Fire LUT to better distinguish regional differences. Central vein (CV) and portal vein (PV) are indicated with a cyan dashed-line in human liver tissue (left, middle). Scale bars, 100 μm (tissue) and 50 μm (organoid). Representative images from $n=3$ independent experiments are shown.

6. Lines 195-198: 'Gene set enrichment analysis showed that h-HepOrg exhibited high expression of Wnt and YAP targets, indicating that the Wnt agonist and the LATS1/2 inhibitor used act specifically through these respective pathways (Extended Data Fig.2e, g-h)'. The gene set enrichment analysis is insufficient to support the claim that 'the Wnt agonist and the LATS1/2 inhibitor used act specifically through these respective pathways' - more experimental data should be presented to substantiate the claim that the inhibitors used act specifically through these pathways.

A: The Reviewer has a very valid point on that gene set enrichment alone does not sufficiently demonstrate pathway activity. Therefore, we have removed the original statement regarding drug specificity in the Results section and replaced it with a more precise description:

(new line 203-206): "Gene Set Enrichment Analysis (GSEA) revealed that h-HepOrg exhibited elevated expression of WNT and YAP target genes, consistent with pathway activation following treatment with the WNT agonist and LATS1/2 inhibitor".

However, to provide an answer to the Reviewer's question, now we have conducted additional qPCR analyses examining multiple canonical YAP target genes (including ANKRD1, CYR61 and AMOTL2) following 12h of treatment of the cells with the LATS1/2 inhibitor to thoroughly evaluate YAP signaling

status. These new experimental results, now presented in new Extended Data Fig. 2j (and also pasted below for the Reviewer, Figure R2.4), provide evidence of YAP pathway activation following LATS1/2 inhibition in our model system. We have carefully updated both the main text and corresponding figure legends to incorporate these findings and ensure accurate data interpretation.

We are deeply grateful to the Reviewer for this valuable suggestion, which has significantly strengthened the rigor and clarity of our study.

Figure R2.4: Quantitative RT-PCR analysis of the YAP downstream target genes ANKRD1, CYR61, and AMOTL2 in the presence (+TRULI, orange) or absence (-TRULI, grey) of the LATS1/2 inhibitor TRULI. Gene expression levels are normalized to HPRT and presented as $2^{-\Delta Ct}$. Data are shown as mean \pm SEM ($n = 6$), and significance was assessed using an unpaired two-tailed Student's t-test. P-values are indicated.

7. Fig2b: The percentage of nuclear YAP+ and nuclear YAP- cells should be quantitated and statistics shown.

A: We have performed a quantitative analysis of the percentage of nuclear YAP+ and YAP- cells, and the corresponding statistical data have now been included in the revised manuscript. The updated figure legend has been revised accordingly, and we have provided the relevant statistical information in the figure panel (new Extended Data Fig. 2i), We hope this additional data strengthens the clarity of our findings.

Figure R2.5: Percentage of YAP-positive nuclei in h-HepOrg in conditions with TRULI (+TRULI) and without TRULI (-TRULI). Graph represents mean with standard deviation. One-tailed Mann-Whitney test was used for statistical analysis. Each dot represents one organoid ($n=3$, +TRULI and $n=1$, -TRULI).

8. Fig 2c: Cytoplasmic to nuclei ratio and bile canalicular architecture should be quantitated and statistics shown.

A: We thank the Reviewer for this valuable suggestion that has helped strengthening the manuscript. In response, we have performed a quantitative analysis of the cytoplasmic to nuclear ratio as well as the bile canalicular architecture. The relevant data have now been incorporated into the revised manuscript, and the statistical analyses are now presented in revised Fig. 3c (bile canaliculi) and new Extended Data Fig. 3b (nuclear to cytoplasmic ratio) (also pasted combined below for the Reviewer's convenience in Figure R2.6). These results further support the characterization of our model and highlight its physiological relevance.

We hope this additional information addresses the Reviewer's concern.

Figure R2.6: *h-HepOrg in DM exhibit enhanced bile canaliculi network connectivity and increased cytoplasmic to nuclear ratio compared to h-HepOrg in EM2. a.* Graph showing the total number of triple junctions as a proxy for connectivity within the bile canaliculi (BC) network. For tissue, each dot represents one field of view while each colour represents a different donor. For organoids, each dot represents one structure (organoid) from the indicated donors, grown in either h-HepOrg-EM2 (EM2) or h-HepOrg-DM (DM). One-tailed Mann-Whitney test was used for statistical analysis. *b.* Graph showing the ratio of cytoplasmic area to nuclear area (CA/NA) in EM2 and in DM. Each dot represents one organoid, different shapes represent different donors (n=3 donors). One-tailed Mann-Whitney test was used for statistical analysis.

9. Fig 2f: The albumin staining in the upper panel in human liver tissue is not convincing. Further, the ECAD and GS staining in the middle and lower panels should be quantitated and measured across multiple donors to allow the reader to assess the variation / reproducibility across donors.

A: This question relates to the earlier point 5 from this same Reviewer. As mentioned above in the response to point 5 we have now replaced albumin with Histidine ammonia-lyase (HAL) as a more robust and specific periportal marker. To avoid confusion from using multiple periportal markers, we have removed the ECAD and albumin staining panels and now show zonation using HAL for both h-

HepOrg and human liver tissue, alongside GS as a central marker (revised Fig. 2f, copied in this letter as Figure R2.3 above). The results are consistent with the previous findings using ECAD and Alb and indicate that the HepOrg show heterogenous expression of zoned markers.

Regarding the Reviewer's suggestion to quantitatively assess ECAD and GS staining across multiple donors: while we fully agree that donor-to-donor variability is an important consideration, which we address in the new analysis of the RNAseq data (Fig. 3d), quantification of zoned markers remains inherently challenging, as their expression patterns are not binary but exist along a gradient from periportal to pericentral regions. This makes it difficult to define objective thresholds for signal intensity across different zones and across donors, particularly in the context of immunofluorescence, where staining intensity can also be influenced by technical factors.

That said, to address concerns regarding reproducibility and donor variability, we have ensured that the revised HAL and GS stainings are representative and derived from multiple independent donors for both the h-HepOrgs and liver tissue (n=3). This allows for qualitative assessment of zonation patterns that are consistent across biological replicates. We hope that these improvements, together with the clearer use of HAL as a periportal marker, sufficiently address the Reviewer's concerns.

10. Fig 3c - Connectivity within the BC network - statistical analysis should be performed to compare the groups.

A: This question relates to the earlier point 8 from this same Reviewer. As mentioned above in the response to point 8 we thank the Reviewer for bringing up this important point regarding the statistical analysis of connectivity within the BC network in Fig. 3c. As explained above (point 8 of this Reviewer), we have conducted the appropriate statistical analysis, which is now included in the revised version of the figure (Fig. 3c, also Figure R2.6a above). The results of this analysis further support our conclusions and provide a more robust comparison between the groups. We have updated the figure legend to include details of the statistical tests performed and the corresponding significance values.

We hope this additional analysis addresses the Reviewer's concern and improves the clarity of our findings.

11. Extended data fig 4b-d: Are patients 1,2 and 3 the same or different from patients A,B and C in these panels? It would be useful to quantitate these readouts in tissue vs organoids from the same

donor livers to allow assessment of how well each individual set of organoids recapitulate in vivo liver tissue from the same donor.

A: We thank the Reviewer for this important question. In the referenced figure, the tissue and organoid samples are not directly matched. As described in the Methods section of the manuscript ([Section Image analysis]), the bile canaliculi (BC) staining protocol requires immediate fixation of the tissue in the operating room (OR), followed by rapid quenching of the fixative. This step is essential to preserve native BC structures and prevent fixation artifacts, as previously described ⁷. Unfortunately, this procedure is incompatible with the requirements for hepatocyte organoid generation, which necessitates fresh tissue subjected to cold perfusion in the OR prior to collagenase digestion.

Our established SOPs for sample collection do not allow to obtain both samples simultaneously at present and amending them was not possible during the timeframe of this revision. Therefore, it was not feasible to collect both types of samples from the same donor at once during this revision. To avoid any confusion, we have revised the figure legend and updated the Discussion section (Lines 286–287 and 529-532) to clarify that our model captures different types of BC structures observed across patients, rather than directly matching tissue and organoid from the same individual.

(Line 286-287): “suggesting that our model could capture different types of BC structures that are observed in patient tissue”.

(Line 529-532): “We observed variability in bile canaliculi (BC) morphology among organoids derived from different donors. However, whether this reflects true patient-to-patient differences remains to be determined and will require further investigation”.

Nonetheless, to address the excellent point raised by the Reviewer regarding how well each individual set of organoids recapitulate in vivo liver tissue from the same donor, we have extended our previous transcriptomic profiling (RNA-seq) to include additional samples from patients recruited during this round of revision. We compared organoids to their source hepatocytes and observed that organoids derived from different donors maintained distinct molecular signatures that reflect inter-patient variability.

Notably, many patient-specific genes expressed both in the organoids and their corresponding source cells have been previously associated with liver-related pathologies, including susceptibility to hepatitis virus infection (e.g., IL1RL1, ERAP2), liver cancer (e.g., GPC3), and cholestasis during pregnancy (e.g., GABRP). Additionally, several donor-specific genes were involved in key metabolic pathways, such as

glutathione (*GSTM3*, *GSTM1*), lactate (*LDHC*), and lipid (*APOA4*, *FAR2*, *ACSM1*) metabolism, among others (Fig. 3d and Supplementary Table 4).

These results suggest that *h-HepOrgs* retain stable, donor-specific molecular characteristics over time in culture, at least up to passage 7, for the samples analysed. These findings are presented in New Fig. 3d, Extended Data Fig. 4e, and Supplementary Table 4, and are discussed in the Results section (Line 289-303). For the Reviewer's convenience, we have also pasted the relevant figures and data below.

Figure R2.7: Hepatocyte organoids retain donor-specific transcriptional signatures that recapitulate inter-donor variability observed in primary hepatocytes. **a.** Heatmap showing scaled expression of donor-specific genes computed across primary human hepatocyte (PHH) samples and hepatocyte organoids cultured in differentiation medium (*h-HepOrg-DM*). Columns represent individual donors, and hierarchical clustering was performed on both samples and genes. Each row corresponds to a donor-specific gene identified by differential expression analysis. The expression profiles of *h-HepOrg-DM* (yellow/brown) samples exhibit high concordance with the inter-donor variability observed in PHH samples (purple), indicating that hepatocyte organoids retain donor-specific transcriptional signatures. **b.** Spearman correlation heatmap showing pairwise similarities between primary hepatocyte samples and differentiated hepatocyte organoids. Donor-specific genes were first identified from fresh primary hepatocytes, and these genes were then used to calculate pairwise correlations with gene expression profiles from organoids. Warmer colours indicate higher correlation coefficients, representing stronger transcriptomic similarity.

12. Fig 3d: The authors state 'Differentiated *h-HepOrg* presented mature hepatic functions similar to freshly isolated hepatocytes, including albumin secretion and, to a lesser extent, cytochrome p450 activity (Fig.3d-e)'. However, the CYP2C9 and CYP3A4 activity presented (Fig. 3d) is statistically significantly different between PHHs and *h-HepOrg-DM* - please clarify.

A: We thank the Reviewer for pointing out this inconsistency in our manuscript. As part of the review process, we have now directly compared the CYP activity between h-HepOrg and matched primary human hepatocytes (PHHs) cultured in standard 2D conditions for 1 or 7 days.

Our analyses revealed that h-HepOrg differentiated in DM secreted comparable amounts of Albumin to matched PHHs cultured for either 1 or 7 days. Confirming our previous results. Similarly, differentiated h-HepOrg showed CYP2C9 activity levels equivalent to 7-day PHHs, and slightly lower CYP3A4 activity, although 1-day fresh PHH cultures outperformed both for these activities.

Therefore, we have now corrected this statement in the manuscript (lines 305-310, pasted below for the Reviewer) to better reflect our findings and included these results in the revised manuscript (revised Fig. 3e-f, also below Figure R2.8 for the Reviewer).

(Line 305-310): "Next, we compared the functional performance of differentiated h-HepOrg to primary human hepatocytes (PHHs). HepOrg differentiated in DM exhibited mature hepatic functions, including robust albumin secretion and, moderate cytochrome P450 activity, comparable to 7-day PHHs (Fig. 3e-f). Specifically, differentiated h-HepOrg displayed CYP2C9 activity equivalent to that of 7-day PHHs, and modestly reduced CYP3A4 activity, while 1-day PHHs demonstrated superior activity for both enzymes".

Figure R2.8: Albumin secretion and cytochrome activity assays on human hepatocyte organoids and primary hepatocytes. a-b. Human hepatocyte organoids expanded in EM2 (h-HepOrg-EM2, dark purple) and differentiated in DM (h-HepOrg-DM, light purple) were tested for functional cytochrome activity (a) and albumin secretion (b) and compared to human liver cholangiocyte organoids (h-CholOrg, green) and fresh isolated primary human hepatocytes (PHH) cultured as standard 2D monolayers for 1 day (1d-PHH, pink) or 7 days (7d-PHH, orange). Graphs represent mean \pm SEM for $n=4-7$ donors from $n=3$ independent experiments. Results are expressed as RLU (a) or ng/ml (b) normalized by the total cell count. Two-way ANOVA with Tukey test for multiple comparisons.

Additionally, we would like to highlight for the Reviewer new data added during the revision on the function of cytochrome p450 in our model. We have now expanded the p450-activity assays to directly compare p450-mediated drug metabolism between h-HepOrg and matched primary human

hepatocytes (PHHs). We selected the drug Verapamil, a L-type calcium channel blocker with antiarrhythmic, antianginal, and antihypertensive activity, because it is extensively metabolized by the liver (up to 80%) via the cytochrome P450 enzyme system. Norverapamil, the major in vivo metabolite, is the result of verapamil's N-demethylation via CYP2C8, CYP3A4, and CYP3A5, and carries approximately 20% of the cardiovascular activity of its parent drug. CYP2D6 and CYP2E1 have also been implicated in the metabolic pathway of verapamil, albeit to a minor extent⁶. Remarkably, mass spectrometry analysis demonstrated that our differentiated h-HepOrg significantly outperformed PHHs from the same donors in their p450-mediated drug metabolizing capacity for Verapamil (new Fig. 3g). These functional results align with our transcriptomic analyses, which indicated that differentiated h-HepOrg and 1-day cultured PHHs express similar levels of key metabolizing enzymes, including CYP2E1, CYP2C9, CYP1A2, CYP2C8, CYP3A4, and CYP3A5—the latter three being responsible for the N-demethylation of Verapamil into Norverapamil (revised Fig. 2d, and Extended Data Fig. 3g and 3i).

These results are presented in the New Fig. 3g and revised Fig. 2d, and in Extended Data Fig. 3g and 3i and are also provided below for the Reviewer's convenience (Figure R2.9).

Figure R2.9: Functional and transcriptional characterization of human hepatocyte organoids cultured in differentiation medium. **a.** The production rate of norverapamil reflects the xenobiotic metabolism capacity of Hepatocytes in HepOrgs. The rate (in pmol/h per 10^3 cells) was determined by mass spectrometry for h-HepOrg and primary human hepatocytes (PHHs) obtained from 3 donors. For h-CholOrg (used as a negative control), the production rate was below the detection limit. Bars represent mean values with SD ($n = 3$). Statistical significance between PHHs and h-HepOrg was assessed using an unpaired two-tailed Student's t-test with Welch's correction,

comparing biological replicates (n = 3 donors per group). b. Heatmap representing the scaled expression of the indicated genes for known hepatocyte and cholangiocyte (Chol.) markers. Their region-specific expression (pericentral or portal) or function is indicated in different colours. c. Heatmap showing the expression of the full list of genes from the drug metabolism datasets from the GSEA analysis between h-HepOrg grown in DM compared to EM2 medium, and their expression in fresh isolated hepatocytes (Primary, PHHs) and in cholangiocyte organoids (h-CholOrg). Note the similarity between DM and primary hepatocytes. Column, donor. d. CPM values from the RNAseq for the hepatocyte marker Albumin (ALB), the mature hepatocyte marker and metabolizing enzyme CYP3A4, CYP3A5, CYP2C8 and CYP2C9, the bile salt transporter MRP2, the cholangiocyte marker KRT19 and the hepatoblast marker AFP. Note that upon DM the expression of mature hepatocyte markers increases while the expression of the cholangiocyte and foetal marker decreases. Graphs present the CPM values mean +/- SEM from n=3 different donors. Dot, independent donor.

13. The authors state that the assembloids mimic human portal architecture, however much of the data shows immunofluorescence staining of central hepatocyte markers, and many central hepatocyte genes are present in the RNA sequencing data. The authors need to further characterise the assembloids in terms of portal vs central hepatocyte ratios / marker expression to investigate whether the assembloids are enriched for portal hepatocytes, in line with trying to mimic portal tissue architecture and function.

For example, in Fig 6c, it is interesting to note that most of the assembloid cells express central hepatocyte genes (Glul/Cyp2e1) despite the biliary context put forward by the authors. As mentioned above, this requires further investigation - if the authors seek to mimic the portal tract it would be important to characterise in-depth whether there are many peri-portal hepatocytes in the assembloids (markers such as HAL could be used to assess this with I.F. staining) and do these cells exhibit portal hepatocyte functions (assessed using assays beyond gene-level data).

A: We would like to thank Reviewer for these very relevant comments on the characterization of our assembloids, which has significantly helped strengthening the scope, impact and quality of our manuscript. This point contains 2 different questions: (i) expression of central and portal markers in assembloids and (ii) portal hepatocyte functions in assembloids.

In response to this concern, we would like to highlight that, as shown in revised Fig. 5h, we indeed observed the expression of several central hepatocyte markers (such as GLUL and CYP2E1), but also of classical portal hepatocyte markers (such as FASN, APOE, SAA1, SAA2 and APOA). Nonetheless, to address this very important point, we have now performed several additional analyses to investigate the zonation identity of our assembloids, as we detail below. These include:

(i) expanding the gene expression analysis of the assembloids and comparing them with matching hepatocyte organoids

(ii) performed staining for portal zonated markers in assembloids

(iii) performed functional tests for hepatocytes from the portal zone

As part of the review process, we expanded our scRNAseq analysis of assembloids with 2 additional patients (n=4 total for the scRNAseq). This provided enough statistical power to re-analyse the data by comparing the gene expression profiles of the hepatocytes from the assembloids (treated as pseudo-bulk data) and hepatocytes from hepatocyte organoids, both cultured under the same DM conditions. Our analysis revealed that hepatocytes within assembloids exhibited higher expression of periportal markers, including SAA1, SAA2, HAMP, and APOA1, while showed reduced expression of pericentral genes such as CYP2E1, CYP3A4, and GLUL, compared to h-HepOrg cultured alone. (new Fig.5i).

These results were in agreement with the staining for the periportal hepatocyte markers SAA1 and SAA2, which confirmed the spatially confined expression of these portal markers in the hepatocytes located within cholangiocyte-rich, ECAD⁺ regions, with portal Mesenchyme (PFs-nRFP) in close proximity (new Extended Data Fig. 9a). This spatial distribution supports the presence of a periportal-like domain within the assembloids and strengthens our interpretation of the model as reflecting portal zonation.

Additionally, and as per the Reviewer's request, we have strengthened the functional data by performing functional gluconeogenesis and urea synthesis assays, which are key metabolic hallmarks of periportal hepatocytes, and we compared these to standard PHH cultures. Periportal assembloids exhibited significantly higher periportal hepatocyte functions, urea production and gluconeogenesis, compared to h-HepOrg cultured under the same DM conditions, while the drug-metabolizing capacity associated with pericentral hepatocytes was significantly less pronounced, in line with their portal-like nature and the transcriptional analysis results detailed above.

These results are now included in the revised manuscript, lines 414-438 and the new Fig. 5i-k, and Extended Data Fig. 9a, b, and methods, and are also pasted below for the Reviewer (Figure R2.10).

All combined, these results indicated that assembloids do, in fact, exhibit an enrichment of portal-specific genes compared to the hepatocyte organoids, which we believe is aligned with our goal of mimicking the portal region of the tissue.

We believe these additions substantially strengthen the functional characterization of our periportal assembloid model and demonstrate, to our knowledge, it is the first in vitro system combining three major human periportal cell types to recreate the key cell-cell interactions and functions of the periportal zone.

(Line 414-438): “Interestingly, we observed heterogeneous expression of classical zoned hepatocyte markers, with a fraction of hepatocytes expressing periportal markers (SAA1 and SAA2, APOA1) and others expressing pericentral markers such as CYP2E1 (Fig. 5h Extended Data Fig. 8b). To investigate whether the periportal assembloid microenvironment and the interaction with portal ductal and mesenchymal populations could promote a more portalized hepatocyte identity, we compared the gene expression profiles of hepatocytes from HepOrg cultured in DM with those from assembloids (also cultured in DM). Notably, hepatocytes within assembloids exhibited higher expression of periportal markers, including SAA1, SAA2, HAMP, and APOA1, while showing reduced expression of pericentral genes such as CYP2E1, CYP3A4, and GLUL, compared to HepOrg cultured alone. (Fig. 5i).

Staining for the periportal hepatocyte markers SAA1 and SAA2 confirmed the spatially confined expression of these portal markers in the hepatocytes located within cholangiocyte-rich, ECAD⁺ regions, with portal Mesenchyme (PFs-nRFP) in close proximity, further supporting the establishment of a periportal-like Hep domain in the assembloids (Extended Data Fig. 9a).

Notably, periportal assembloids exhibited enhanced functional specialization characteristic of periportal hepatocytes. They outperformed h-HepOrg, cultured in the same conditions, in both urea production and gluconeogenesis (both portal functions), while the drug-metabolizing capacity associated with pericentral hepatocytes was less pronounced compared to hepatocyte organoids, in line with their more portal-like nature. As expected, periportal assembloids retained core hepatocyte functions, with albumin secretion increasing over time to levels matching hepatocyte organoids and exceeding 2D primary hepatocyte cultures. (Fig.5j-k and Extended Data Fig. 9b).

These findings suggested that the periportal microenvironment within the assembloids could promote the acquisition of a more portal-like hepatocyte phenotype”.

Figure R2.10: Compared to Primary Human Hepatocytes (PHHs), h-HepOrg present superior drug metabolizing capabilities while periportal assembloids exhibit enhanced periportal function. **a.** Heatmap showing expression of liver zonation genes in h-HepOrg (h-HepOrg-DM) and in hepatocytes from assembloids. Note that both, h-

HepOrg and Assembloids are cultured in DM medium and while h-HepOrg present enhanced expression of central vein markers, including several p450-family drug metabolizing enzymes, hepatocytes from assembloids show clear enrichment of periportal markers. b. Immunofluorescence staining of day 4 periportal assembloids reveals distinct spatial expression of the portally zoned markers SAA1 and SAA2 (top row, first and second panels). Co-staining with DAPI (nuclei), Chol-nGFP, and PFs-nRFP highlights compartmentalization of SAA1/SAA2 expression within cholangiocyte-rich regions (top row, third panel). Zonation is further contextualized by co-localization with ECAD (E-cadherin; epithelial marker), reinforcing epithelial identity of SAA1/SAA2-expressing cells (top and bottom rows, fourth and fifth panels). Additional multiplexed imaging (bottom row) demonstrates spatial relationships among ECAD, Chol-nGFP, PFs-nRFP, and SAA1/SAA2 expression. These results confirm the establishment of periportal-like zonation architecture in assembloids. Scale bar, 100 μ m. c-d, Urea and glucose production in assembloids (brown), hepatocyte organoids (purple), PHHs (pink) and h-CholOrg (green). c. Assembloids cultured for 5 days (brown) were used to analyse urea synthesis compared to hepatocyte organoids cultured in differentiation medium (DM, purple), freshly isolated primary human hepatocytes cultured in monolayer for 24 h (PHHs, pink), and CholOrg (green) as a negative control. Data represent mean \pm SD. from n=3 independent donors. d. Assembloids were cultured for 6 days prior to gluconeogenesis analysis and compared to hepatocyte organoids cultured in differentiation medium (DM), and CholOrg as a negative control. Data represent mean \pm SD. from n=3 independent donors. e. Mass spectrometry analysis was used to detect Norverapamil, the primary metabolite of Verapamil, in hepatocyte organoids compared to assembloids (6-day culture). For h-CholOrg (used as a negative control), the production rate was below the detection limit. Bars represent mean values with SD (n = 3). Statistical significance between PHHs and h-HepOrg was assessed using an unpaired two-tailed Student's t-test with Welch's correction, comparing biological replicates (n = 3 donors per group).

14. Fig 3f: Xenotransplantation of organoids and PHHs into the Tyrosinemia type I liver disease mouse model (Fah^{-/-}Rag2^{-/-}Il2rg^{-/-} mouse):

Only the survival curve data is presented here. The authors should show the following data:

a) the number of mice in each group - I could not find this information in the Results, Methods or Fig legend text.

A: We apologize for the oversight. The number of mice transplanted with PHHs and in the sham-operated group is 7 each, while 3 mice received hHepOrg_EM2 and 4 received hHepOrg-DM. We have now included these group sizes in the Methods section of the revised manuscript.

b) What was the efficiency of engraftment of the organoids and PHHs? An intrasplenic injection was performed, but there is no data presented demonstrating that the hepatic organoids engrafted efficiently into the liver.

A: We thank the Reviewer for pointing out an important aspect of the characterization of the engraftments. To address this point, the mice transplanted with PHHs and organoids were sacrificed around day 90 to assess engraftment efficiency. The results showed that the mice transplanted with organoids expanded in our HepOrg_EM2 medium and treated with DM (h-HepOrg-DM organoids) exhibited an average intrahepatic engraftment efficiency of 9.0%. This engraftment efficiency is similar

to the engraftment efficiency of the positive control group transplanted with primary human hepatocytes (6.5%), indicating that our organoids engrafted efficiently into the recipient mice.

We have now included these additional data and results in the revised version of the manuscript Lines 321-325 and new Extended Data Fig.4f copied below as Figure R2.11 for the Reviewer.

Figure R2.11: Human GAPDH immunohistochemistry in liver tissues from *Fah^{-/-}/Rag2^{-/-}/Il2rg^{-/-}* mice transplanted with PHHs or *h-HepOrg*. Representative images of immunohistochemistry for Human GAPDH on liver tissue sections from (FRG) mice transplanted with fresh isolated primary human hepatocytes (PHH-graft, left) or *h-HepOrg* cultured in DM (*h-HepOrg-DM-graft*) at 90-92 days after transplantation. Scale bar, 1 mm (left); 100 µm (right).

And if the organoids did engraft in the liver, what was the topographical location of the organoids? eg which zone in the liver? This could be assessed with immunofluorescence staining of the livers across the various groups of mice.

A: We thank the Reviewer for this insightful comment. As the organoids were transplanted via splenic injection, the majority of cells engrafted around the portal vein, as confirmed by immunohistochemical staining. However, we also observed some positively engrafted cells in peri-venous regions between vessels, suggesting that engraftment was not confined to a specific zonation but occurred broadly within the tissue parenchyma. See Figure R2.11 above"

In the revised version of the manuscript, we have modified the text to include that point (line320-324).

(Line 321-325): "Remarkably, both expanded and differentiated hepatocyte organoids readily engrafted and maintained their hepatic function *in vivo*, following xenotransplantation in the Tyrosinemia type I liver disease mouse model (*Fah^{-/-}Rag2^{-/-}Il2rg^{-/-}* mouse)⁸, with grafts distributed throughout the liver parenchyma. Importantly, the engrafted cells were able to rescue the otherwise lethal phenotype of the mice (Fig. 3h and Extended Data Fig. 4f)".

15. Line 334, Fig 5c and Ext Fig 7b: authors state low adhesion U-plates readily generated assembloids within 24h, however Ext Fig 7b states D6. Fig 5f has no mention of culture time. Please clarify in results text, figures and legend.

A: We sincerely appreciate the Reviewer's careful attention to these technical details. We apologize for the confusion caused by the inconsistent description in the text and figures.

The Statistics on aggregation efficiency shown in Fig. 5c were indeed taken after 24h of aggregation, either in U-Plate or Aggrewell plates. In contrast, the images presented in Fig. 5b, Extended Data Fig 7b and 7c correspond to assembloids maintained in culture until day 6 (D6) prior to downstream analyses.

We apologize for the confusion caused by the inconsistent figure referencing in the text and we sincerely appreciate the Reviewer's careful attention to this detail.

To address this, we have now revised the text and the legends of Fig. 5 and Extended Data Fig. 7 to explicitly state the timeline of assembloid formation and culture duration to avoid ambiguity (Lines 375-380, results section).

(Line 375-380): "Of all the approaches tested, mixing one h-HepOrg structure with a defined number of portal mesenchymal and ductal cells (derived from h-CholOrg) in 96-well low adhesion U-plates readily generated assembloids within 24h (Extended Data Fig.7d). From all the cellular ratios tested, the ratio 1 HepOrg structure: 25 MSC and 100 Chol better captured the tissue cell ratios at day 6 post-assembloid formation, and was selected for further experiments from hereon (Extended Data Fig. 7b-c)".

16. Extended data 5d: The Thy-1 staining presented is not convincing.

A: The staining images in Extended Data Fig. 5d were direct micrographs retrieved from Human Protein Atlas database. Upon careful re-evaluation, we acknowledge that the THY-1 staining signal in the original Extended Data Fig. 5d was weak and could lead to ambiguity.

As we already provide representative immunofluorescence staining for THY-1 (CD90) in Fig. 4c and in the top panel of Fig. 5d (tissue panel), both of which more clearly illustrate the THY-1 expression pattern in the portal region, we have decided to remove the THY-1 staining from Extended Data Fig. 5d in the revised manuscript. We hope this clarification addresses the Reviewer's concern, and we sincerely thank them for their careful assessment.

17. Fig 5d: The assembloid images in this figure and multiple subsequent figures (eg Extended Fig 7f; Extended Fig 8 b,d) are not convincing in terms of recapitulating the structure of in vivo portal tracts, ie cholangiocytes clearly forming central lumens and mesenchymal cells surrounding the bile duct structures. Many of the mesenchymal cells are distal to cholangiocytes in the assembloids.

A: We believe the Reviewer may have overlooked relevant data presented in Fig. 5d, Fig. 5f, and Extended Data Fig. 7g in the previous version of the manuscript. These three panels show assembloid images where portal mesenchymal cells (nuclear-RFP-tagged, and additionally VIMENTIN-stained in Fig. 5f and Extended Data Fig. 7g) are in very close proximity to, and in occasions surround, cholangiocytes (nuclear-GFP-tagged, h-Chol-GFP).

The Reviewer cites Figures where mesenchymal cells were only nuclear-tagged (nRFP) but not stained for membrane/cytoplasmic markers (e.g., Extended Data Fig. 7f; Extended Data Fig. 8b, d (now moved to Extended Data Fig. 7h, j)). We agree this could have caused confusion, as mesenchymal cells in vivo exhibit long protrusions capable of spanning considerable distances to interact with cholangiocytes, even when the somas are spatially separated in 3D.

The images in Fig. 5f, and Extended Data Fig. 7g, where Msc cells are stained with Vimentin, from the first version of the manuscript, revealed that in periportal assembloids mesenchymal cells are in close proximity to the ducts and also extend long cellular processes that reach toward and physically interact with the basal side of cholangiocyte structures, reminiscent of the in vivo in human tissue. To further illustrate this for the Reviewer and strengthen this observation, in the revised version of the manuscript, we have included additional examples: revised Extended Data Fig. 7g as well as replaced the original image (middle pannel) with a more representative example that better illustrates the spatial relationship between portal fibroblasts (PFs) and cholangiocytes (revised Extended Data. 7g, copied below as Figure R2.12a). We also provide additional immunofluorescence staining, (Figure R2.12b, for Reviewer reference only), which consistently confirm the presence of mesenchymal cells next to the ducts and show the cell extensions in contact with cholangiocytes despite distal soma positioning.

We believe these additions substantially strengthen the functional characterization of our periportal assembloid model and demonstrate that human periportal assembloids recreate the key cell–cell interactions of the periportal zone. We acknowledge, though, that our in vitro assembloids do not fully replicate the compact anatomical arrangement of in vivo portal tracts with the Msc in tight arrangement wrapping the duct. Accordingly, we have moderated our statements regarding architectural fidelity. Nonetheless, we believe that the observed cellular interactions reflect a meaningful degree of structural and functional organization, especially considering the complexity of modelling the human liver microenvironment.

These findings are now indicated in the revised manuscript (Results, lines 391-395 and Discussion, lines 548-550 also copied below for the Reviewer). Additionally, the updated figure panel (revised Extended Data. 7g, middle panel) and the Figure for Reviewer only are both pasted below in Figure R2.12 for the Reviewer's assessment. We trust these provide clearer evidence of the Msc co-localization with cholangiocytes and hope these clarifications address the Reviewer's concern.

(Line 391-395): "Vimentin staining confirmed that portal mesenchymal cells (nuclear-RFP) were integrated within the assembloid structure and in close proximity to cholangiocytes (nuclear-GFP). Msc consistently extended long cellular processes that reached toward and physically interacted with the basal side of cholangiocytes, reminiscent of the interactions observed *in vivo* in human tissue (VIMENTIN panel in Fig. 5f and Extended Data Fig. 7g), although not completely wrapping the cholangiocytes as in the portal tracts *in vivo*".

(Line 548-550): "These structures recapitulated key aspects of native tissue architecture *in vitro*, with mesenchymal cells positioned in close proximity to, and establishing basal contacts with, cholangiocytes, both embedded within a hepatocyte parenchyma".

Figure R2.12: In assembloids, periportal fibroblasts extend long cellular processes that reach toward and interact with the basal side of cholangiocyte structures. **a.** Representative confocal images of periportal assembloids containing human portal fibroblasts (nuclear magenta) and human cholangiocytes from h-CholOrg (nuclear green) cultured for 3 days and stained for hepatocyte (HNF4a, yellow) markers. Nuclei are stained with DAPI (blue) and membranes with Phalloidin (white). Scale bars, 50 μm. Note that cholangiocytes (green) are embedded in the hepatocyte structure and contain portal mesenchyme (magenta) in close proximity **b.** Representative confocal images of periportal assembloids stained for the PFs marker vimentin (white). Nuclei and membranes (left, middle) are stained with DAPI (blue) and phalloidin (magenta), respectively. Scale bars, 50 μm. Note that PFs present long cellular projections that span several cell distances.

Furthermore, in Ext data 7e the phalloidin staining is picking out many lumens which have no associated cholangiocytes,

A: We apologize for the lack of clarity on our part and thank the Reviewer for raising this point, which has significantly helped improving the accessibility of our presentation. Extended Data Fig. 7e (now moved to Extended Data Fig. 7d) illustrates an assembloid at the 24-hour time point in AggreWell cultures. As discussed earlier, 24 hours was the earliest time point we analyzed to assess initial cell integration within the structures. We believe that the luminal structures observed without associated cholangiocytes at this early 24-hour stage— but not at later time points (Days 3 and 6)— reflect early-stage reorganization of the hepatocytes in the aggregates. As the assembloids mature, these lumens get reduced and instead the lumens detected all become consistently associated with cholangiocytes, supporting the dynamic and progressive nature of this morphogenetic process.

To better illustrate this dynamic cell re-organization during assembloid formation, we have now arranged the panels to illustrate this cellular dynamic behaviour (Extended Data Fig. 7d, new Extended Data Fig. 7e and revised Figure Extended Data Fig. 7f) and have clarified that also in the text in lines 383-389.

(Line 383-389): “Remarkably, from day 3-onwards, the periportal assembloids recapitulated key architectural features of the *in vivo* tissue, with ductal cells (KRT19+ and nGFP+) forming bile duct-like structures containing open lumina, with portal mesenchymal cells (nuclear-RFP) in close proximity, and embedded within the hepatocyte (HNF4a+) parenchyma (Fig.5d, f and Extended Data Fig. 7e-g)”.

Figure R2.13: Time-dependent morphogenesis and cellular re-organization during human periportal assembloids formation. **a.** Representative confocal images of assembloids after 24 hours in AggreWell demonstrating that h-PFs (Red) and h-Chol (Green) have already assembled with h-HepOrg (not labelled, DAPI, blue) to generate composite structures. Scale bars, 100 μm (d) and 50 μm (e). **b.** Representative confocal images of periportal assembloids cultured for 3 days. Nuclei are stained with DAPI (blue); cell membranes are visualized using Phalloidin (white). Scale bars, 50 μm . **c.** Representative confocal images of assembloids after 6 days in culture. Nuclei are stained with DAPI (blue); cell membranes are visualized using Phalloidin (white). Scale bars, 50 μm .

and in Ext data 7g PFs (portal fibroblasts) rarely co-locate with cholangiocytes.

As discussed above, the new images in Extended Figure. 7e and the images in panels Fig. 5f, and Extended Data Fig. 7g, where Msc cells are stained with Vimentin, reveal that in periportal assembloids

mesenchymal cells are in close proximity, in occasions surrounding the cholangiocytes, and extend long cellular processes that reach toward and physically interact with the basal side of cholangiocyte structures, reminiscent of the *in vivo* in human tissue. In the new version of the manuscript, for the revised Figure Extended Data Fig. 7g, we have replaced the original image with a more representative example that better illustrates the spatial relationship between portal fibroblasts (PFs) and cholangiocytes. The updated figure (also pasted above as Figure R2.12a, for the Reviewer's assessment) now provides clearer evidence of their co-localization. We hope these clarifications address the Reviewer's concern.

18. Fig 5f: there are no scale bars on the lower images.

A: We apologize for this oversight and thank the Reviewer for his/her meticulous attention to detail. We have now added scale bars to the lower images in Fig. 5f.

Further, in panel 4 the phalloidin stain looks nuclear and overlaps with HNF4a. Why is this different to the rest of the manuscript and the expected expression of phalloidin?

A: We thank the Reviewer for this valuable comment, which made us realize that the presentation of these panels was confusing. We respectfully clarify that in panel 4 of Fig. 5f, the white signal is not derived from phalloidin staining, as **phalloidin was not included** in the overlay shown in this panel.

To explain further: in the previous version of Fig. 5f, panels 3 and 4 presented multiplex staining across five channels (DAPI, blue; h-PFs, magenta; cholangiocytes, green; HNF4A, yellow; phalloidin, white). To facilitate visualization, we split the data into two separate panels:

Panel 3 displayed DAPI (blue), h-PFs (magenta), cholangiocytes (green), and phalloidin (white).

Panel 4 displayed DAPI (blue), h-PFs (magenta), cholangiocytes (green), and HNF4A (yellow)—**without phalloidin**.

Thus, the "white nuclei" observed in panel 4 resulted from the overlap of the DAPI (blue) and HNF4A (yellow) signals, which when merge appear in white in the composite image.

We acknowledge that the inclusion of the DAPI channel in panel 4 may have contributed to the confusion. To address this, we have now removed the DAPI channel from that panel in the revised Fig. 5f (middle panel) to avoid misinterpretation. For the Reviewer's convenience, the corrected figure is also provided below.

Figure R2.14: Representative confocal images of periportal assembloids containing h-Chol (green), portal mesenchyme (magenta) and stained for the hepatocyte marker HNF4a (yellow). Nuclei and membranes (left, middle) are stained with DAPI (blue) and phalloidin (green, left, grey, middle), respectively. Scale bars, 50 μm .

19. Line 344: the authors state periportal assembloids retain hepatocyte functions, however only Alb secretion is assessed. Further types of hepatocyte function need to be assessed. The authors also state that secretion of Alb is comparable to hepatocyte organoids alone, however is that with or without DM? As well as performing statistical analyses, the authors need to increase the 'n' for the assembloids.

A: We thank the Reviewer for pointing out the need to further assesses hepatocyte function within the periportal assembloids, which has helped us increase the quality of the manuscript. Here the Reviewer raises two points:

(i): the need to increase the n-number and clarification of the Albumin secretion data for assembloids presented in the previous version of the manuscript (Extended Data Fig 8, now moved to Fig. 5i).

(ii): the need for assessing additional hepatocyte functions in assembloids, beyond Albumin

We answer both points together below:

In response to the Albumin data concerns, we have now expanded the Albumin secretion analysis to increase the number of biological replicates (n= 4-7 donors, in at least 3 independent experiments) and time points of analysis to 1, 6, and 10 day-assembloids, and compared the Albumin secretion to matching HepOrg cultured in EM2 or DM and matching (when possible) 1-day and 7-day PHH cultures. We have also included the statistical analysis as requested. Our results confirm our previous observation that 6 and 10 day-assembloids secrete similar levels of Albumin as matching HepOrg in DM and 7-day PHHs in sandwich culture, and significantly higher levels than 1-day PHHs in monolayer.

We have now included these results in the revised version of the manuscript (Lines 433-436) and the revised Extended Data Fig.8a (now moved to Fig. 5I), and also pasted them below for the Reviewer's convenience (Figure R2.15).

Figure R2.15: Human albumin secretion by assembloids cultured for 1, 6, or 10 days compared to h-HepOrg alone and to PHHs. Data from hepatocyte organoids and primary human hepatocytes are reproduced from Fig. 3f for comparison. Albumin levels were normalized to total cell number. Data are mean +/-SEM from n = 4–7 independent donors. Two-way ANOVA with Tukey test for multiple comparisons. Note that periportal assembloids show comparable levels of Albumin secretion than h-HepOrg and standard 7-day PHH cultures.

In response to the need for assessing additional functions, in this revised version of the manuscript, we have strengthened the functional data by performing gluconeogenesis and urea synthesis assays, which are key metabolic hallmarks of periportal hepatocytes, and we compared these to standard PHH cultures. As indicated in the response to point#13 of this same Reviewer, periportal assembloids exhibited significantly higher periportal hepatocyte functions, urea production and gluconeogenesis, compared to h-HepOrg cultured under the same DM conditions, while the drug-metabolizing capacity associated with pericentral hepatocytes was significantly less pronounced, in line with their portal-like nature and the transcriptional analysis that indicates a more portalized expression pattern, compared to HepOrg in the same DM medium.

These results are now included in the revised manuscript, lines 429-436 and the new Fig. 5j-k, and Extended Data Fig. 9b, and methods, and are also pasted below for the Reviewer (Figure R2.16).

We believe these additions strengthen the functional characterization of our periportal assembloid model and address the Reviewer's concerns.

(Line 429-436): “Notably, periportal assembloids exhibited enhanced functional specialization characteristic of periportal hepatocytes. They outperformed h-HepOrg, cultured in the same conditions, in both urea production and gluconeogenesis (both portal functions), while the drug-metabolizing capacity associated with pericentral hepatocytes was less pronounced compared to hepatocyte organoids, in line with their more portal-like nature. As expected, periportal assembloids retained core hepatocyte functions, with albumin secretion increasing over time to levels matching hepatocyte organoids and exceeding 2D primary hepatocyte cultures. (Fig.5j-l and Extended Data Fig. 9b)”.

Figure R2.16: Compared to Primary Human Hepatocytes (PHHs), h-HepOrg present superior drug metabolizing capabilities while periportal assembloids exhibit enhanced periportal function. **a-b**, Urea and glucose production in assembloids (brown), hepatocyte organoids (purple), PHHs (pink) and h-CholOrg (green). **a**. Assembloids cultured for 5 days (brown) were used to analyse urea synthesis compared to hepatocyte organoids cultured in differentiation medium (DM, purple), freshly isolated primary human hepatocytes cultured in monolayer for 24 h (PHHs, pink), and CholOrg (green) as a negative control. Data represent mean \pm SD. from $n=3$ independent donors. **b**. Assembloids were cultured for 6 days prior to gluconeogenesis analysis and compared to hepatocyte organoids cultured in differentiation medium (DM), and CholOrg as a negative control. Data represent mean \pm SD. from $n=3$ independent donors. **c**. Mass spectrometry analysis was used to detect Norverapamil, the primary metabolite of Verapamil, in hepatocyte organoids compared to assembloids (6-day culture). For h-CholOrg (used as a negative control), the production rate was below the detection limit. Bars represent mean values with SD ($n = 3$). Statistical significance between PHHs and h-HepOrg was assessed using an unpaired two-tailed Student’s t-test with Welch’s correction, comparing biological replicates ($n = 3$ donors per group).

Line 348 short term culture in Ext Fig 7f: length is not stated here – what is meant by short term?

A: We thank the Reviewer for pointing out the need to clarify this point. We defined "short-term" as a culture duration of <6 days, as opposed to long-term cultures (up to a month, as shown in Extended Data Fig 11). Specifically, the duration of the culture described in Extended Data Fig. 7f was 6 days. We agree that this nomenclature is ambiguous and have removed it now from the manuscript. Instead, we just state the time in culture in text or Figure legends and, only for the fibrosis model, we use long-term to refer to >2 weeks-old assembloids. We have clarified that in the text in lines 383 of the main text and legends.

20. Ext Fig 8a: bars show assembloids at day 1 and 6, however the figure legend states day 1 and 7.

A: We apologise for this inconsistency. The previous Extended Data Fig 8 contained Albumin secretion in assembloids. As explained above, through the review process we have expanded the Albumin secretion analysis increasing the number of time points of analysis to 1, 6, and 10 days, and the revised Extended Data Fig.8a has now moved to Fig. 5i. We have carefully reviewed the legend to ensure accuracy and clarity, and edited the figure legend to reflect the correct time points.

21. Ext Fig 8 is largely descriptive. Please quantify cell death and proliferation. And to allow comparison with the results presented in Ext Fig 7f, the authors need to provide quantitation comparing the 'short term' and 'longer term' cultures.

A: We have now performed comprehensive quantifications of both cell death and proliferation rates, as shown in the new Extended Data Fig. 7k, including statistical comparisons between short-term (day 6) and long-term (day 16) cultures. These analyses revealed no significant differences in either proliferation or apoptosis between the two time points, indicating that the assembloid cultures maintain a stable cellular state over time. The corresponding quantification has been incorporated in the revised manuscript text and new Extended Data Fig. 7k and are also provided below for the Reviewer's assessment.

Figure R2.17: Quantification of KI67+ and Cleaved Caspase 3+ cells in assembloids cultured for 6 or 16 days. Results are presented as mean \pm SD of $n = 3$ donors, with two replicates per donor (total of 6 data points per group). P values were calculated using two-sided unpaired Student's t-test.

What do the authors mean by 'architectural disorganisation'? Showing brightfield images does not make this clear. Perhaps F-actin or phalloidin staining over the time course would clarify this?

A: In the previous version of the manuscript, this terminology appeared in the sentence:

Line 345-347: "Under these conditions, the assembloids could be maintained in vitro for at least two weeks without showing any evidence of cell death or architectural disorganization."

Our intention was to convey that the assembloids retained their 3D structure and organization over culture. However, we appreciate the Reviewer's point that the phrasing was ambiguous. To address this, we have revised the manuscript to more precisely describe the findings (revised text in Lines 396–398, pasted below for the Reviewer's convenience).

(Lines 396–398): “Under these conditions, the assembloids could be maintained for at least two weeks in culture, maintaining their cellular states, with no evidence of increased cell death or proliferation over time (Extended Data Fig. 8j–k)”.

Furthermore, the authors should quantify assembloid function over time eg Alb secretion and other markers of healthy organoids.

A: This point is related to Points #13 and #19 from the same Reviewer, regarding the need to further demonstrate hepatocyte function within the assembloids. As detailed above, we have expanded the functional characterization of our model by measuring gluconeogenesis and urea synthesis, in addition to repeating the albumin secretion analysis with multiple donors and time points, as requested. These assays—particularly gluconeogenesis and urea production, which are portal-associated functions—demonstrate that hepatocytes within the assembloids are functional and healthy. Furthermore, we highlight that assembloids exhibit reduced, but detectable, pericentral drug-metabolizing activity, consistent with their acquisition of a more portal identity.

We believe these additions (that are presented above in point #13 and #19, Figure R2.10 and R2.15) further strengthen the manuscript and highlight the dynamic functional capabilities of the model.

22. Extended Fig 8a: The 'n' is very low for the assembloids (day 1 and day 6) in this figure (n=2) - more donor assembloids should be assessed in this dataset, ie the 'n' should be increased in these groups.

A: We believe this is the same point as Point #19 where the Reviewer asked “The authors also state that secretion of Alb is comparable to hepatocyte organoids alone, however is that with or without DM? As well as performing statistical analyses, the authors need to increase the ‘n’ for the assembloids.”

As detailed above, we have now expanded the Albumin secretion assay to increase the sample size and time points of analysis. These results are detailed in the response to Point #19 and the associated Figure R2.15. We refer here the Reviewer to the response to this point above.

23. Extended Data Fig 8f (upper panel): the number of bile canaliculi per assembloid, across donors should be quantitated.

A: The Reviewer requested quantification of the number of bile canaliculi per assembloid based on Extended Data Fig. 8f (now moved to Extended Data Fig. 9d). However, we respectfully clarify that in that figure—and elsewhere in the manuscript—we had not performed bile canaliculi staining in assembloids, only in HepOrg.

Nonetheless, in response to this comment, we have now stained assembloids for bile canaliculi to demonstrate the presence of bile canaliculi structures within our assembloids. However, we have not undertaken detailed quantification of bile canaliculi in assembloids, as such quantitative analysis is not possible with the current pipeline due to technical challenges with the bile canaliculi segmentation that was developed for hepatocyte organoids. The current pipeline relies on the overlap of the phalloidin staining with the bile canaliculi staining (CD13) for the segmentation step. Because in assembloids, contrary to HepOrg, we have the immunofluorescently-labelled cholangiocytes and PFs incorporated within the structure, we have only a limited number of channels to use for the bile canaliculi staining and phalloidin. Basically, we have DAPI for the nuclei to visualize the unlabelled hepatocytes, green channel for the ductal cells (cholangiocytes), red for the portal fibroblasts— and finally, CD13 for bile canaliculi in the far red. Due to the limitation in channels, we included phalloidin in the same channel as for the portal fibroblasts. To perform a quantitative analysis of the bile canaliculi, we need to take the entire structure of the assembloids into account. Unfortunately, we were not able to segment only bile canaliculi, as there was a hinderance in accurate segmentation and consequentially correct analysis when phalloidin signals were in the same channel where we have the other cell-types. Given that in the assembloids section of the manuscript we are not presenting bile canaliculi data neither we discuss bile canaliculi structure in the assembloids, we have only included qualitative representative images below for the Reviewer only. Here, the images are maximum intensity projections of specific z-stacks where we do not have overlapping signals from the other cell-types, as we had to put the

staining of two markers in the same channel, and reduced the number of channels displayed for clearer visualization of the bile canaliculi.

Figure R2.18 for Reviewer only: *Bile canaliculi staining in assembloids:* Representative images of human assembloids stained for CD13, a marker for bile canaliculi (yellow). Ductal cells (DC) were labelled in green and nuclei were stained with DAPI (cyan). Scale bars, 100 µm (left) and 50 µm (right, magnified regions).

24. Fig 7h, Ext Data fig 10c - Ki67 immunofluorescence staining of mesenchymal cells and cholangiocytes should be performed and quantitated.

A: Here we believe the Reviewer refers to Fig. 7e (Ki67 and caspase3 immunofluorescence staining, now moved to Extended Data Fig 10a), and not Fig 7h (now moved to Fig. 6h) nor Extended Data Fig 10c (now moved to Extended Data Fig. 10f), which refers to the GSEA of Cholangiocytes (Go-term analysis and Reactome).

As response to this point, we have now quantified the number of Ki67+ in both cholangiocytes and mesenchymal cells and compared it to homeostatic assembloids, as requested. Additionally, we have also quantified the number of caspase3+ hepatocytes. We observed significantly higher proliferation of cholangiocytes under fibrosis-like conditions, compared to homeostatic-like conditions, early after assembly (day 0, aggregation after 24 hours), although there was high variation across the donors (Figure R2.19a). This became not significant at day 7. Mesenchymal cells showed a trend toward higher proliferation that was not significant, in agreement with publicly available liver cell atlases of human fibrotic tissue that shows that these cells do not proliferate in patients^{9,10}.

We also observed significantly increased cleaved caspase 3 staining in hepatocytes, suggesting hepatocyte death, which could explain the reduced number of hepatocytes in the fibrotic-like structures (Figure R2.19b).

These results have been now included in the revised version of the manuscript (Lines 457-461) and Extended Data Fig 10b-c, and are also pasted below for the Reviewer.

(Lines 457-461): “Ki-67 staining indicated that cholangiocytes exhibited early proliferative responses to fibrotic cues, while cleaved-caspase 3 staining revealed that the reduction in hepatocyte numbers was associated with increased hepatocyte death, at least in part, through apoptosis (Extended Data Fig. 10a-c). This finding is consistent with our unpublished observations in mouse assembloids, suggesting a conserved mechanism across species”.

Figure R2.19: Cholangiocytes and mesenchymal cell proliferation and cell death under homeostatic and fibrotic conditions. **a.** Statistical analysis of day 0 (following 24 hours of aggregation in the Aggrewell system) and day 7 revealed significantly higher proliferation of cholangiocytes under fibrosis-like conditions compared to homeostatic-like conditions at day 0. Additionally, mesenchymal cells showed a trend toward higher proliferation under fibrosis-like conditions at this early time point. These findings are consistent with the dynamic nature of the culture conditions and provide important insights into the early response of these cell types to fibrotic cues. By day 7, proliferation rates decreased and showed no significant differences between conditions. Results are presented as mean +/-SD of n=3 independent donors. **b.** Quantification of cleaved Caspase-3+ cells as a percentage of total cells in day 7 (d7) Homeostatic-like and Fibrosis-like assembloids. Each dot represents one technical replicate, with two replicates per donor (n = 3 donors). Bars represent mean +/-SD. P-value was calculated using a two-sided paired Student's t-test.

Referee #3 (Remarks to the Author):

The manuscript by Lei Yuan et al. presents a novel method for developing human periportal liver assembloids. The systematic exploration of pathway activators and inhibitors, supported by pathway analysis (IPA), is unbiased. The authors combined Wnt and Hippo signaling modulation in addition to ingredients from past methods to generate human hepatocyte organoid culture successfully. They established a living biobank and provided a detailed characterization of these organoids. Additionally, the authors developed methods to culture portal mesenchyme and integrated it with human liver organoids and cholangiocyte organoids. These efforts culminated in a platform capable of modeling certain aspects of biliary fibrosis. The statistics are used appropriately, and the conclusions are robust. The manuscript is written very well and easy to follow.

A: We sincerely thank the Reviewer for the positive evaluation and encouraging comments on our work. We greatly appreciate the recognition of the novelty of our periportal liver assembloid model. We are especially grateful for the Reviewer's comments on the clarity of the manuscript, the robustness of our conclusions, and the appropriateness of the statistical analyses.

In response to the Reviewer's valuable feedback, we have further refined our manuscript. We believe these revisions have enhanced the clarity and impact of our work.

Below, we provide a detailed point-by-point response to the Reviewer's comments.

My concerns are below:

1- Long-term expansion of adult human hepatocytes as organoids (h-HepOrg) holds significant potential for modeling liver tissue and advancing regenerative medicine.

A: We thank the Reviewer for appreciating the impact of our work and the potential to advance regenerative medicine.

However, it remains unclear how the current study addresses persistent challenges in this field and substantially advances our understanding of the underlying factors. Metrics such as high albumin levels or Apo proteins, while exciting, have been almost achieved in previous studies. The full maturation spectrum of hepatocytes still has not been reached in this work which has been a main target for many past studies. These are metrics such as P450 family enzymes including master drug

metabolism enzyme CYP3A4. The level that is on par with primary human hepatocytes still not achieved as it is shown in the study.

A: We thank the Reviewer for this comment that has helped us improve the quality of the manuscript. The reviewer is concerned on the p450 activity of our h-HepOrg and its comparative to standard Primary Human Hepatocytes (PHHs), specially taking into account the importance of p450 enzymes in drug metabolism.

To address this point, we have now expanded the p450-activity assays to directly compare p450-mediated drug metabolism between h-HepOrg and matched primary human hepatocytes (PHHs). We selected the drug Verapamil, a L-type calcium channel blocker with antiarrhythmic, antianginal, and antihypertensive activity, because it is extensively metabolized by the liver (up to 80%) via the cytochrome P450 enzyme system. Norverapamil, the major in vivo metabolite, is the result of verapamil's N-demethylation via CYP2C8, CYP3A4, and CYP3A5, and carries approximately 20% of the cardiovascular activity of its parent drug. CYP2D6 and CYP2E1 have also been implicated in the metabolic pathway of verapamil, albeit to a minor extent⁶. Remarkably, mass spectrometry analysis demonstrated that our differentiated h-HepOrg significantly outperformed PHHs from the same donors in their p450-mediated drug metabolizing capacity for Verapamil (revised Fig. 3e, f and new Fig. 3g). These functional results align with our transcriptomic analyses, which indicated that differentiated h-HepOrg and 1-day cultured PHHs express similar levels of key metabolizing enzymes, including CYP2E1, CYP2C9, CYP1A2, CYP2C8, CYP3A4, and CYP3A5—the latter three being responsible for the N-demethylation of Verapamil into Norverapamil (revised Fig. 2d, and Extended Data Fig. 3g and 3i).

These results are now presented in the revised manuscript (revised Fig. 3e, f, new Fig. 3g, revised Fig. 2d, and Extended Data Fig. 3g and 3i), and for the Reviewer's convenience, we have also included them below (Figure R3.1).

Figure R3.1: Functional and transcriptional characterization of human hepatocyte organoids cultured in differentiation medium. **a-b.** Human hepatocyte organoids expanded in EM2 (passage 1-7) and differentiated in DM were tested for functional cytochrome activity (**a**) and albumin secretion (**b**) and compared to human liver cholangiocyte organoids (**h-CholOrg**) and fresh isolated primary human hepatocytes (**PHH**) cultured as standard 2D monolayers for 1 or 7 days. Graphs represent mean \pm SEM for $n=4-7$ donors from $n=3$ independent experiments. Results are expressed as RLU (**a**) or ng/ml (**b**) normalized by the total cell count. Two-way ANOVA with Tukey test for multiple comparisons was used. **c.** The production rate of norverapamil reflects the xenobiotic metabolism capacity of Hepatocytes in HepOrgs. The rate (in pmol/h per 10^3 cells) was determined by mass spectrometry for **h-HepOrg** and primary human hepatocytes (**PHHs**) obtained from 3 donors. For **h-CholOrg** (used as a negative control), the production rate was below the detection limit. Bars represent mean values with SD ($n=3$). Statistical significance between **PHHs** and **h-HepOrg** was assessed using an unpaired two-tailed Student's *t*-test with Welch's correction, comparing biological replicates ($n=3$ donors per group). **d.** Heatmap representing the scaled expression of the indicated genes for known hepatocyte and cholangiocyte (**Chol.**) markers. Their region-specific expression (pericentral or portal) or function is indicated in different colours. **e.** Heatmap showing

the expression of the full list of genes from the drug metabolism datasets from the GSEA analysis between h-HepOrg grown in DM compared to EM2 medium, and their expression in fresh isolated hepatocytes (Primary, PHHs) and in cholangiocyte organoids (h-CholOrg). Note the similarity between DM and primary hepatocytes. Column, donor. f. CPM values from the RNAseq for the hepatocyte marker Albumin (ALB), the mature hepatocyte marker and metabolizing enzyme CYP3A4, CYP3A5, CYP2C8 and CYP2C9, the bile salt transporter MRP2, the cholangiocyte marker KRT19 and the hepatoblast marker AFP. Note that upon DM the expression of mature hepatocyte markers increases while the expression of the cholangiocyte and foetal marker decreases. Graphs present the CPM values mean +/- SEM from n=3 different donors. Dot, independent donor.

Additionally, we expanded on the p450 activity levels for CYP2C9 by comparing differentiated h-HepOrg and matching PHHs cultured for 1 day or 7 days in sandwich culture (in our previous version these were not matching cells). Our results showed CYP2C9 activity levels equivalent to 7-day PHHs, and slightly lower CYP3A4 activity, although 1-day fresh PHH cultures outperformed both for these activities. These results are now presented in the revised manuscript (revised Fig. 3e), and for the Reviewer's convenience, we have also included them below (Figure R3.2).

Figure R3.2: Cytochrome activity assays on human hepatocyte organoids and primary hepatocytes. a-b. Human hepatocyte organoids expanded in EM2 (h-HepOrg-EM2, dark purple) and differentiated in DM (h-HepOrg-DM, light purple) were tested for functional cytochrome activity (a) and albumin secretion (b) and compared to human liver cholangiocyte organoids (h-CholOrg, green) and fresh isolated primary human hepatocytes (PHH) cultured as standard 2D monolayers for 1 day (1d-PHH, pink) or 7 days (7d-PHH, orange). Graphs represent mean +/- SEM for n=4-7 donors from n= 3 independent experiments. Results are expressed as RLU (a) or ng/ml (b) normalized by the total cell count. Two-way ANOVA with Tukey test for multiple comparisons.

2- Other studies have also attempted to tackle these challenges by using primary hepatocytes to expand culture methods, and they have also shown limited success. An experimental comparison with prior methods, such as those by Peng *et al.*, 2018¹ and Hu *et al.*, 2018³, would have provided a helpful reference point to know how much improvement in this study could be achieved. This is at the level of hepatocyte organoids as well as organoids with other Non-parenchymal cells (PFs). To my assessment

the current system demonstrates some of the same limitations, including a degree of immaturity in drug metabolism / maturation markers like CYP3A4. However, it shows improvements in factors such as albumin production and cholestatic mechanisms.

A: We thank the Reviewer for raising this point regarding the advances of our model over existing methods, particularly in terms of hepatocyte functionality and drug metabolism. This point is connected to point 1 above.

*To address the Reviewer’s questions, we conducted new comparative analyses of expansion potential, hepatocyte function, drug metabolism, and gene expression between our h-HepOrg model and (i) primary human hepatocytes (PHHs), as well as (ii) the previously published mouse hepatocyte organoid model¹ and human fetal hepatocyte organoid model³ cited by the Reviewer. Our findings, summarized in the **new Supplementary Table 3** and detailed below, highlight the key advancements of our system:*

1) Enhanced Expansion Potential and Biobanking Capacity

*We directly compared the expansion potential of h-HepOrg cultures under our EM1 and EM2 conditions with protocols described by Hu et al., 2018³ and Peng et al., 2018¹. As shown in **new Extended Data Fig. 1j**, our h-HepOrg cultures expanded in EM2 exhibited robust (17-fold) expansion within the first week, while the other methods showed minimal or no growth. This capacity enabled us to establish a living biobank of 28 donor-derived organoid lines that we can expand for >3months and can be frozen and thawed efficiently — a major advantage over previous models. These results, presented in the **new Extended Data Fig. 1j** are pasted below for the Reviewer’s convenience (**Figure R3.3**)*

Figure R3.3: Cell expansion comparison between h-HepOrg grown in our h-HepOrg-EM2 medium and previously reported hepatocyte cultures. Following hepatocyte isolation, a total of 12,500 cells were seeded in Matrigel and cultured in our h-HepOrg-EM2 or EM1 medium or in the medium published by Peng et al., or Hu et al., or in our MM medium supplemented or not with Wnt, TRULI. We also used as controls the conditions where the preparation was not MACS or the tissue was not perfused, as well as the EpCAM+ cholangiocyte fraction cultured in our cholangiocyte medium. The number of cells was counted on an automatic cell counter at the

indicated time points (day 0, 10, and 15). We only observed robust cell expansion in either *h*-HepOrg grown in our EM2 medium or in the *h*-CholOrg grown in cholangiocyte medium from Huch et al 2015. Graph represents the mean +/- SD of *n*=3 independent donors with *n*=3 technical replicates from *n*=3 independent experiments. Statistical significance was determined by two-way ANOVA with Dunnett's test for multiple comparisons was used, comparing each condition to the *h*-HepOrg-EM2 group at the last time point (day15). The colour coding indicates the comparison of our *h*-HepOrgEM2 medium to the respective medium in the legend.

2) Gene expression closely resemble PHHs

As part of the revision process, we expanded our RNA-seq analyses to include *h*-HepOrg, freshly isolated PHHs, and 1-day cultured PHHs. Transcriptomic profiling revealed that *h*-HepOrg cultured in DM closely match the global gene expression of freshly isolated PHHs, though 1-day PHHs remain slightly closer. Importantly, for key functional categories such as drug and cholesterol metabolism, *h*-HepOrg show comparable expression to PHHs. These results are presented now in the revised Extended Data Figure 3c-h and also pasted below for the Reviewer's convenience (Figure R3.4)

Figure R3.4: Differentiated human hepatocyte organoids retain gene expression of mature in vivo hepatocytes. a-f. *h*-HepOrg organoids were cultured for several passages in *h*-HepOrg-EM2 medium and then differentiated in DM as indicated in methods. Seven days after the start of the differentiation protocol, the organoids were

harvested and processed for RNAseq analysis. **a.** PCA analysis showing the PC1 and PC2 components from the RNAseq analysis from fresh isolated primary hepatocytes (primary, pink), cholangiocyte organoids (h-CholOrg, green) and h-HepOrg in EM2 (dark purple) or DM (light purple). **b.** Heatmap showing all DEG between h-HepOrg grown in DM compared to EM2 medium, and their expression in fresh isolated hepatocytes (Primary, PHHs) and in cholangiocyte organoids (h-CholOrg). Note the similarity between DM and primary hepatocytes. Column, donor. **c.** Selected list of gene sets from the GO-terms database that are significantly enriched in h-HepOrg under DM compared to EM2. The full list is presented in Supplementary Dataset_2. The results are presented as a dot plot where the dot colour denotes the adjusted p-value. Note that many of positively enriched gene sets are related to classical hepatocyte functions such as lipid and drug metabolism and bile secretion and transport. Conversely, negatively enriched gene sets are related to cell cycle. **d-e.** Heatmap showing the expression of the full list of genes from the cholesterol dataset (e) and drug metabolism (f) datasets from the GSEA analysis between h-HepOrg grown in DM compared to EM2 medium, and their expression in fresh isolated hepatocytes (Primary, PHHs) and in cholangiocyte organoids (h-CholOrg). Note the similarity between DM and primary hepatocytes. Column, donor. **f.** Heatmap showing the expression of DEG between DM and EM2 that intersect with the gene list of pericentrally or periportally zoned genes from Yu et al., 2022¹¹, Brazonvskaja et al. 2024¹², Yakubovski et al., 2025¹³ and Watson et al. 2025¹⁴.

In parallel, we compared the gene expression profiles of our expanded (h-HepOrg-EM2) and differentiated (h-HepOrg-DM) HepOrg with publicly available datasets. To facilitate this comparison, we leveraged datasets already compiled in Ardismita et al., 2022⁴ that include the human fetal HepOrg models from the Clevers laboratory, as cited by the Reviewer, as well as other hepatocyte culture systems and fresh isolated PHHs. As shown in Figure R3.5, we found by PCA that differentiated h-HepOrg samples (h-HepOrg-DM, red circles) and h-HepOrg in expansion (EM1/EM2, green circles) cluster closely with PHHs from other studies (turquoise triangles) as well as with fetal (purple triangles) and PSC-derived hepatic cells (pink triangles) from Hu et al 2018³ and Wang et al., 2020⁵ respectively.

However, because these comparisons were not performed on samples from the same donors, we cannot fully exclude that donor variation may influence these results. Additionally, due to too strong batch effects, we could not use all the samples from our manuscript, mostly only the samples from the first sequencing batch, hence why there is only 2 samples in DM condition. Therefore, due to these 2 caveats, we have opted to present these results below for the Reviewer's consideration only (Figure R3.5, for Reviewer only). Should the Reviewer or Editor feel that inclusion of these findings would further strengthen the manuscript, we would be pleased to incorporate them accordingly.

Figure R3.5, for the Reviewer only: Comparison between our *h-HepOrg* models and published human culture systems. PCA was performed on gene expression profiles from our study (circles), including *h-HepOrg* in expansion medium (*h-HepOrg-EM*, green) or DM (*h-HepOrg-DM*, red) or *h-CholOrg* cultured in our cholangiocyte medium (grey), and in the publicly available datasets compiled in Ardisasmita et al., 2022 (triangles), which include studies containing primary human hepatocytes (PHHs, turquoise), Common Bile Duct tissue (CBD, light yellow), Liver tissue (liver, dark yellow), Fetal hepatocytes (Fetal_Hep, light green) and pluripotent stem cells (PSC, dark purple) and corresponding hepatic cultures of these (HLC, hepatic liver cells) including Cholangiocyte organoids (Chol_HLC, blue), Fetal hepatocyte organoids (FHep_HLC, light purple), hepatocytes in monolayer (Hep_HLC, grey), HepG2 cells (orange) and PSC-derived hepatic liver cells (PSC_HLC, pink). Each point represents an individual sample in each study. Colours indicate different conditions and can include more than one study. Note that many samples from our study, both in EM and DM, cluster close to the PHHs (turquoise) from many studies, as well within the Fetal hepatocyte organoids (FHep-HLC, light purple) and the PSC-derived HLC from Wang et al., 2020⁵. Due to too strong batch effects, we could only use the samples from our first sequencing batch, hence why there is only 2 samples in DM condition. P, passage.

3) Similar Hepatic function and enhanced drug metabolizing capacity:

As shown above, in the response to point 1, we have now directly compared hepatocyte function (Albumin secretion, CYP activity, and drug metabolism) between *h-HepOrg* and matched primary human hepatocytes (PHHs) cultured for 1 or 7 days. A direct comparison with the published mouse (Peng et al., 2018)¹ and human fetal (Hu et al., 2018)³ *HepOrg* models was not possible, as we were unable to expand adult human hepatocytes under the reported conditions.

Our analyses revealed that *h-HepOrg* differentiated in DM secreted comparable amounts of Albumin to matched PHHs cultured for either 1 or 7 days. Similarly, differentiated *h-HepOrg* showed CYP2C9

activity levels equivalent to 7-day PHHs, and slightly lower CYP3A4 activity, although 1-day fresh PHH cultures outperformed both for these activities. Importantly, mass spectrometry analysis demonstrated that our differentiated h-HepOrg significantly outperformed PHHs from the same donors in their drug metabolizing capacity for Verapamil (revised Fig. 3e, f and new Fig. 3g).

These new results are now presented in the revised manuscript (New Fig. 3e–g, revised Fig. 2d, and Extended Data Fig. 3g and 3i), and for the Reviewer’s convenience, we have also included them above, in the response to point 1 (Figure R3.1).

4) Patient-specific fidelity

As part of the review process, we extended our previous transcriptomic profiling (RNA-seq) to include additional samples from patients recruited during this round of revision. We compared organoids to their source hepatocytes and observed that organoids derived from different donors maintained distinct molecular signatures that reflect inter-patient variability.

Notably, many patient-specific genes expressed both in the organoids and their corresponding source cells have been previously associated with liver-related pathologies, including susceptibility to hepatitis virus infection (e.g., *IL1RL1*, *ERAP2*), liver cancer (e.g., *GPC3*), and cholestasis during pregnancy (e.g., *GABRP*). Additionally, several donor-specific genes were involved in key metabolic pathways, such as glutathione metabolism (*GSTM3*, *GSTM1*), lactate metabolism (*LDHC*), and lipid metabolism (*APOA4*, *FAR2*, *ACSM1*), among others (Fig. 3d and Supplementary Table 4).

These results suggest that h-HepOrgs retain stable, donor-specific molecular characteristics over time in culture, at least up to passage 7. These findings are presented in new Fig. 3d, Extended Data Fig. 4e, and Supplementary Table 4, and are discussed in the Results section (Line 526-529). For the Reviewer’s convenience, we have also pasted the relevant figures and data below.

(Lines 526-529): “Notably, h-HepOrg retained the transcriptomic profiles of their donor cells in a patient-specific manner. The organoids preserved inter-individual variability in metabolic enzymes as well as in genes associated with susceptibility to viral infection, lipid metabolism and liver pathology”.

Figure R3.6: Hepatocyte organoids retain donor-specific transcriptional signatures that recapitulate inter-donor variability observed in primary hepatocytes. **a.** Heatmap showing scaled expression of donor-specific genes computed across primary human hepatocyte (PHH) samples and hepatocyte organoids cultured in differentiation medium (h-HepOrg-DM). Columns represent individual donors, and hierarchical clustering was performed on both samples and genes. Each row corresponds to a donor-specific gene identified by differential expression analysis. The expression profiles of h-HepOrg-DM (yellow/brown) samples exhibit high concordance with the inter-donor variability observed in PHH samples (purple), indicating that hepatocyte organoids retain donor-specific transcriptional signatures. **b.** Spearman correlation heatmap showing pairwise similarities between primary hepatocyte samples and differentiated hepatocyte organoids. Donor-specific genes were first identified from fresh primary hepatocytes, and these genes were then used to calculate pairwise correlations with gene expression profiles from organoids. Warmer colours indicate higher correlation coefficients, representing stronger transcriptomic similarity.

5) Generation of assembloids with portal identity

Because we start with patient-derived material, we have been able to generate not only hepatocyte organoids but also cholangiocyte organoids and portal mesenchymal cell cultures from the same donors, allowing us to assemble liver assembloids containing all three cell types from the same individual. This level of donor-matched multicellular complexity has not been achievable with previous methods.

Furthermore, new analyses performed during the revision process show that hepatocytes within the assembloids acquire a more portalized gene expression profile compared to hepatocyte organoids cultured alone. Specifically, assembloid hepatocytes upregulate portal zone markers such as *SAA1/2*, *HAMP*, and *APOE*, and display enhanced functional features associated with the portal region,

including ureagenesis and gluconeogenesis. In contrast, the expression of central vein markers and associated functions is reduced relative to HepOrg alone. These findings suggest that the inclusion of ductal cells and portal mesenchyme creates a microenvironment that promotes a more portal-like hepatic identity.

These results are presented in the new Figure 5i-k, Extended Data Fig. 9a-b, and we also combined them below for the Reviewer (Figure R3.7).

Figure R3.7: Compared to Primary Human Hepatocytes (PHHs), h-HepOrg present superior drug metabolizing capabilities while periportal assembloids exhibit enhanced periportal function. a. Heatmap showing expression of liver zonation genes in h-HepOrg (h-HepOrg-DM) and in hepatocytes from assembloids. Note that both, h-HepOrg and Assembloids are cultured in DM medium and while h-HepOrg present enhanced expression of central vein markers, including several p450-family drug metabolizing enzymes, hepatocytes from assembloids show clear enrichment of periportal markers. **b.** Immunofluorescence staining of day 4 periportal assembloids reveals distinct spatial expression of the portally zoned markers SAA1 and SAA2 (top row, first and second panels). Co-staining with DAPI (nuclei), Chol-nGFP, and Pfs-nRFP highlights compartmentalization of SAA1/SAA2 expression within cholangiocyte-rich regions (top row, third panel). Zonation is further contextualized by co-localization with ECAD (E-cadherin; epithelial marker), reinforcing epithelial identity of SAA1/SAA2-expressing cells (top and bottom rows, fourth and fifth panels). Additional multiplexed imaging (bottom row) demonstrates spatial relationships among ECAD, Chol-nGFP, Pfs-nRFP, and SAA1/SAA2 expression. These results confirm the establishment of periportal-like zonation architecture in assembloids. Scale bar, 100 μm. **c-d,** Urea and glucose production in assembloids (brown), hepatocyte organoids (purple), PHHs (pink) and h-CholOrg (green). **c.** Assembloids cultured for 5 days (brown) were used to analyse urea synthesis compared to hepatocyte organoids cultured in differentiation medium (DM, purple), freshly isolated primary human hepatocytes cultured in monolayer for 24 h (PHHs, pink), and CholOrg (green) as a negative control. Data represent mean ± SD, from n=3 independent donors. **d.** Assembloids were cultured for 6 days prior to gluconeogenesis analysis and compared to hepatocyte organoids cultured in differentiation medium (DM), and CholOrg as a negative control. Data represent mean ± SD, from n=3 independent donors. **e.** Mass spectrometry analysis was used to detect Norverapamil, the

primary metabolite of Verapamil, in hepatocyte organoids compared to assembloids (6-day culture). For h-CholOrg (used as a negative control), the production rate was below the detection limit. Bars represent mean values with SD (n = 3). Statistical significance between PHHs and h-HepOrg was assessed using an unpaired two-tailed Student's t-test with Welch's correction, comparing biological replicates (n = 3 donors per group).

Taken together, our results indicate that while the h-HepOrg model closely resembles existing methods in terms of gene expression and CYP function (with the exception of CYP3A4 levels at day 1 in PHHs), it notably outperforms previous models in several key aspects. These include enhanced expansion potential and biobanking capacity, superior drug-metabolizing ability (at least for the drug tested), preservation of patient-to-patient variability, and the capacity to generate composite assembloid structures with portalized functions.

We have summarized this side-by-side comparison of our model with previously published systems in the new Supplementary Table 3, which highlights the unique strengths and innovations of our platform, along with a transparent discussion of its current limitations.

*please also make sure that Peng et al is cited.

A: We cite Peng, et al., 2018¹, as detailed above (new Reference #34 in the revised version of the manuscript).

3- What is the starting hepatocyte population, and would the initial zonation of hepatocyte impact the outcome?

A: We thank the Reviewer for raising this important point. As described in the Methods section, the primary hepatocytes were derived from patient tissue via enzymatic digestion and subsequent MACS sorting, without further stratification based on pericentrally or periportal zonation subpopulations. Thus, the starting hepatocyte population likely represents a heterogeneous mix of cells from across the liver lobule. Importantly, our new analyses (New Fig. 5i and functional data Fig. 5j-k) show that periportal features in our model emerge through the controlled co-culture conditions and cell-cell interactions within the assembloid environment, rather than being pre-determined by the initial zonation of hepatocytes (detailed in the response to point #2 above, "Generation of assembloids with portal identity"). These findings suggest that the microenvironment plays a dominant role in shaping the zonation-like features observed in our system.

Nonetheless, we fully acknowledge the Reviewer's valid point that we cannot completely exclude the possibility that a subpopulation of hepatocytes with a specific initial zonation bias may preferentially

contribute to organoid formation and expansion. Unfortunately, we were unable to directly address this possibility, as current technical limitations preclude reliable separation of human hepatocytes based on zonation-specific surface markers without substantial loss of cell viability. In our experience, FACS sorting (unlike MACS) significantly compromises viability of patient-derived hepatocytes, and as we previously demonstrated, viability >50% is critical for successful organoid generation (Supplementary Table 1). Optimizing this process would require substantial additional efforts beyond the scope of the current revision.

We have now explicitly discussed this limitation in the revised manuscript (Lines 556–558).

(Line 556-558): “Likewise, we cannot fully exclude the possibility that distinct hepatocyte subpopulations at the onset of culture influence differential responses to microenvironmental cues”.

4- The addition of portal mesenchyme is a novel aspect. However, it is not clear if the presented outcomes could have been achieved with any other cells with myofibroblasts or fibroblasts origin. For example, they could use stellate cells with hepatocyte organoids as control and compare what would be special about this combination and how it can shed light on specific periportal-related phenotypes that are beyond a simple wound healing response. Hence, unfortunately while promising it still is unclear whether the model demonstrates specific events beyond a generic desmoplastic inflammatory reaction that could be achieved with other cells.

A: The Reviewer raises an important point about whether the observed outcomes could be replicated by other mesenchymal cells, such as hepatic stellate cells (HSCs). To address this, we explicitly compared the effects of human HSCs (hHSCs) with human portal fibroblasts (hPFs) in co-culture with hepatocyte and cholangiocyte organoids. Our data reveal striking differences in cellular behavior and fibrogenic impact, supporting the unique role of hPFs in modeling periportal-specific phenotypes.

Specifically, we observed that hHSCs, when introduced in excess to induce fibrosis, formed pronounced spiky, stellate-like structures which seemed to condense and pull matrix on both hepatocytes and cholangiocytes. This aligns with known HSC-driven fibrotic mechanisms, where activated HSCs exhibit a highly contractile, myofibroblastic phenotype (Friedman et al., 2008¹⁵). In contrast, hPFs induced a distinct fibrotic response characterized by a more diffuse extracellular matrix deposition, reminiscent of periportal fibrosis patterns (Mederacke et al., 2013¹⁶).

Further, staining for proliferation and apoptosis markers on assembloids where portal fibroblasts drive fibrosis presented a significant 15-fold increase in KI67+ cells (suggestive of compensatory proliferation), whereas assembloids with hHSCs trended toward elevated Cleaved Caspase-3 (though

not statistically significant, $P=0.0068$). This implies divergent cellular stress responses: hPFs may promote a regenerative microenvironment, while hHSCs might favour a pro-apoptotic milieu, consistent with their roles in parenchymal vs. periportal fibrogenesis¹⁷.

Thus, while both cell types elicit fibrotic reactions, their morphological, biomechanical, and functional outcomes differ markedly. Our hPF-based model uniquely captures periportal-specific features—such as ductular reaction and portal-inflammatory crosstalk—that cannot be fully recapitulated by hHSCs, which are more representative of parenchymal scarring. We acknowledge that further investigation is needed to elucidate the precise mechanisms by which each mesenchymal cell type contributes to the distinct phenotypes observed. However, we believe that such mechanistic studies constitute a substantial research project in their own right and are beyond the scope of the current manuscript.

Given the limited time frame of this revision and the fact that hepatic stellate cells (HSCs) are not primary drivers of biliary fibrosis, we have chosen not to include these preliminary findings in the manuscript and are instead sharing them here exclusively for the Reviewer's consideration. That said, should the Reviewer or Editor feel that inclusion of this data would enhance the manuscript, we would be happy to revise accordingly and follow their guidance.

[Figure Redacted]

5- A parallel study is under review, almost the same title in mice from the same team and reported in the references #39. While this study focuses on the human model, the differences between the mouse

model and the human model reported here could be minimal and can take away from innovation of this study.

*A: We thank the Reviewer for raising this important point. While reference #39 (now Ref 41) reports a parallel study using a mouse-based periportal liver model, we would like to emphasize that the human assembloid model presented in our current study offers distinct and complementary advantages. To clarify the differences between the two models, we have included a detailed comparison in **new Supplementary Table 3**, highlighting species-specific features and applications.*

Notably, our human model robustly allows the expansion of hepatocytes, cholangiocytes and portal mesenchyme from patient tissue, and, as unique feature, from the same patient, and has allowed us to generate a living biobank of 28 donors to date. Thanks to the Reviewer's suggestions, we have now expanded on the characterization of our systems, both the hepatocyte organoids and the assembloids, and show that our human model robustly captures inter-donor variability, which allows for the study of patient-specific phenotypes and facilitates applications in precision medicine and functional diagnostics—capabilities that are inherently limited in mouse-based models.

We also show that it is an invaluable resource for human drug toxicity studies and drug metabolism, which are inherently different between rodents and humans. Here, we demonstrate that our human liver organoid exhibit clinically relevant drug metabolism capacity and the assembloid system recapitulates periportal-related functions (gluconeogenesis and ureagenesis), which are essential for modeling human-specific pathophysiology and therapeutic responses.

*Together, these features underscore the translational potential and unique utility of **the human periportal assembloid system, which complements but does not duplicate the findings in the mouse study**.*

6- The MACS sorting of EpCAM-positive cells for the depletion of these precursors of biliary organoids should be validated through FACS analysis to ensure accuracy and reproducibility and remove any possibility of residual cells used for making organoids.

*A: We thank the Reviewer for this important and constructive suggestion. As recommended, we have performed additional flow cytometry (FACS) analysis on both pre-sorted and post-sorted hepatic cell populations to rigorously evaluate the purity and efficiency of our MACS-based isolation protocol. As shown in the **newly added Extended Data Fig. 1c**, the EpCAM-positive population was effectively removed following MACS sorting, resulting in a highly pure EpCAM-negative hepatocyte fraction. These results confirm the efficacy and reproducibility of our MACS-sorting strategy and rule out the*

presence of residual EpCAM-positive precursor cells that could contribute to biliary organoid formation. We have updated the manuscript to include this validation and referenced the new figure accordingly (see Methods, page 30, and Results, page 3) which we also include below for the Reviewer's consideration. Thank you again for this constructive critique, which has strengthened our study.

(Lines 125-126): "as we confirmed by FACS (Extended Data Fig. 1c)".

Figure R3.9: After MACS, the EpCAM^{neg} fraction containing Primary Human Hepatocytes is efficiently depleted from EpCAM⁺ cells. **a.** Following collagenase perfusion, Flow cytometry analysis was performed to evaluate the presence of EpCAM⁺ cells in human liver single-cell suspensions before and after MACS-based removal of EpCAM⁺ cells. Note that post-MACS sorting, a highly enriched EpCAM⁻ hepatocyte fraction is obtained. **b.** Quantification of EpCAM⁻ cell percentages before and after MACS. Data are represented as mean \pm SD ($n = 3$ donors).

7- The efficiency of the method and inter-individual variations should be reported to provide a clearer understanding of the robustness and applicability of this approach.

A: We thank the Reviewer for raising this important point.

Regarding efficiency, it is unclear whether the Reviewer refers to the generation of hepatocyte organoids or the formation of assembloids. In both cases, the relevant data were included in the previous version of the manuscript—specifically in Supplementary Table 2 (for organoid derivation efficiency) and Fig. 5c (for assembloid formation efficiency). It is possible this information may have been overlooked by the Reviewer.

To briefly summarize: in Supplementary Table 2, we showed that hepatocyte organoids could be derived from 100% of patient samples (18 out of 18) when cell viability exceeded 50% and cells were cultured in our optimized HepOrg_EM2 medium. Since the initial submission, this number has increased to 28 donors, with derivation efficiency consistently maintained at 100%. For assembloid formation, the efficiency is approximately 80%, as stated in the main text of the first submission (line

338) and shown in Fig. 5c. To avoid confusion, we have now explicitly clarified this in the revised manuscript lines 176 and 382.

As for inter-individual variability, we would like to highlight that also this information was already included in Supplementary Table 2. This table summarizes key characteristics of the donor samples, including age, sex, diagnosed disease, isolation method, cell viability, culture conditions (media composition), and passage number. This table underscores the robustness and reproducibility of our system across the 28 independent patient-derived samples.

We have now also emphasized this citation more clearly in the main text to support our discussion on the model's consistency and translational relevance. We believe these data provide a comprehensive overview of the efficiency and inter-individual consistency of our system.

References

- 1 Peng, W. C. *et al.* Inflammatory Cytokine TNF α Promotes the Long-Term Expansion of Primary Hepatocytes in 3D Culture. *Cell* **175**, 1607-1619 e1615 (2018). <https://doi.org:10.1016/j.cell.2018.11.012>
- 2 Zhang, K. *et al.* In Vitro Expansion of Primary Human Hepatocytes with Efficient Liver Repopulation Capacity. *Cell Stem Cell* **23**, 806-819 e804 (2018). <https://doi.org:10.1016/j.stem.2018.10.018>
- 3 Hu, H. *et al.* Long-Term Expansion of Functional Mouse and Human Hepatocytes as 3D Organoids. *Cell* **175**, 1591-1606 e1519 (2018). <https://doi.org:10.1016/j.cell.2018.11.013>
- 4 Ardisasmita, A. I. *et al.* A comprehensive transcriptomic comparison of hepatocyte model systems improves selection of models for experimental use. *Commun Biol* **5**, 1094 (2022). <https://doi.org:10.1038/s42003-022-04046-9>
- 5 Wang, Q. *et al.* Generation of human hepatocytes from extended pluripotent stem cells. *Cell Res* **30**, 810-813 (2020). <https://doi.org:10.1038/s41422-020-0293-x>
- 6 Zhou, S. F., Zhou, Z. W., Yang, L. P. & Cai, J. P. Substrates, inducers, inhibitors and structure-activity relationships of human Cytochrome P450 2C9 and implications in drug development. *Curr Med Chem* **16**, 3480-3675 (2009). <https://doi.org:10.2174/092986709789057635>
- 7 Segovia-Miranda, F. *et al.* Three-dimensional spatially resolved geometrical and functional models of human liver tissue reveal new aspects of NAFLD progression. *Nat Med* **25**, 1885-1893 (2019). <https://doi.org:10.1038/s41591-019-0660-7>
- 8 Azuma, H. *et al.* Robust expansion of human hepatocytes in Fah^{-/-}/Rag2^{-/-}/Il2rg^{-/-} mice. *Nat Biotechnol* **25**, 903-910 (2007). <https://doi.org:10.1038/nbt1326>
- 9 Dobie, R. *et al.* Single-Cell Transcriptomics Uncovers Zonation of Function in the Mesenchyme during Liver Fibrosis. *Cell Rep* **29**, 1832-1847 e1838 (2019). <https://doi.org:10.1016/j.celrep.2019.10.024>
- 10 Guilliams, M. *et al.* Spatial proteogenomics reveals distinct and evolutionarily conserved hepatic macrophage niches. *Cell* **185**, 379-396 e338 (2022). <https://doi.org:10.1016/j.cell.2021.12.018>
- 11 Yu, S. *et al.* Spatial transcriptome profiling of normal human liver. *Sci Data* **9**, 633 (2022). <https://doi.org:10.1038/s41597-022-01676-w>
- 12 Brazovskaja, A. *et al.* Cell atlas of the regenerating human liver after portal vein embolization. *Nat Commun* **15**, 5827 (2024). <https://doi.org:10.1038/s41467-024-49236-7>
- 13 Yakubovsky, O. *et al.* A spatial transcriptomics atlas of live donors reveals unique zonation patterns in the healthy human liver. *bioRxiv*, 2025.2002.2022.639181 (2025). <https://doi.org:10.1101/2025.02.22.639181>
- 14 Watson, B. R. *et al.* Spatial transcriptomics of healthy and fibrotic human liver at single-cell resolution. *Nat Commun* **16**, 319 (2025). <https://doi.org:10.1038/s41467-024-55325-4>
- 15 Friedman, S. L. Mechanisms of hepatic fibrogenesis. *Gastroenterology* **134**, 1655-1669 (2008). <https://doi.org:10.1053/j.gastro.2008.03.003>

- 16 Mederacke, I. *et al.* Fate tracing reveals hepatic stellate cells as dominant contributors to liver fibrosis independent of its aetiology. *Nat Commun* **4**, 2823 (2013). <https://doi.org:10.1038/ncomms3823>
- 17 Kisseleva, T. & Brenner, D. Molecular and cellular mechanisms of liver fibrosis and its regression. *Nat Rev Gastroenterol Hepatol* **18**, 151-166 (2021). <https://doi.org:10.1038/s41575-020-00372-7>

Point by point response letter to the Reviewers:

We appreciate the editor and Reviewers for their continued time and effort in evaluating our manuscript. We are grateful for the insightful and constructive comments provided during this second round of review, which have allowed us to further improve the clarity and robustness of our study.

We have addressed all points raised and revised the manuscript accordingly. A point-by-point response is provided below, with Reviewer comments in black and our responses in blue and *italics*.

All textual revisions are **tracked and highlighted in yellow** in the revised manuscript.

Referee #1 (Remarks to the Author):

The authors have addressed all my points. Well done!

A: We thank the Reviewer for the positive comments. Much appreciated.

Referee #2 (Remarks to the Author):

The authors have comprehensively addressed my comments, I only have one further minor comment (listed below). My congratulations to the authors on an outstanding body of work.

A: We thank the Reviewer for the positive comments. Much appreciated.

Minor comment:

Fig 2f: In the h-HepOrg-DM images, the upper and lower panels appear to be from different samples - it would be helpful to the reader to show combined (dual) immunofluorescence staining of GS and HAL in the h-HepOrg-DM, thereby highlighting the heterogeneity and spatial distribution of hepatocyte function within the same h-HepOrg-DM.

A: We appreciate the reviewer's suggestion and fully agree that showing combined immunofluorescence staining of GS and HAL within the same h-HepOrg-DM would more clearly illustrate the spatial distribution of hepatocyte function. Unfortunately, the antibodies used for GS (Anti-Glutamine Synthetase antibody produced in rabbit, Sigma-Aldrich (Merck), #G2781) and HAL (Anti-HAL antibody produced in rabbit, Sigma-Aldrich (Merck), #HPA038547) are raised in the same species, which precluded dual staining on the same organoid structure. As a result, we presented images from separate but equivalent organoids derived from the same donor.

To achieve this dual staining, we would need to obtain, validate and optimize organoid staining for alternative antibodies from different host species, which is unfortunately not feasible within the current revision timeline.

Nonetheless, to provide an answer to the reviewer, we performed dual staining for CYP2E1 and ECAD to assess spatial heterogeneity within individual h-HepOrg-DM. CYP2E1 is a well-established pericentral hepatocyte marker, and in the previous version of the manuscript we had already shown it is heterogeneously expressed in h-HepOrg in DM (previous Extended Data Fig. 3j). ECAD, on the other hand, although broadly expressed in hepatocytes throughout the liver lobule, has been shown to exhibit stronger expression in the periportal region in mouse liver (Hempel, Schmitz et al. 2015) and recent spatial transcriptomics data indicates that it is moderately enriched in the portal region in human liver (Yakubovsky, Afriat et al. 2025).

Our results show that while all cells within the same h-HepOrg-DM express ECAD, a subset exhibits weaker ECAD staining and is positive for CYP2E1, whereas strongly ECAD+ cells are negative for CYP2E1. This reciprocal expression pattern within the same organoid is consistent with zoned hepatocyte function and supports the presence of intra-organoid spatial heterogeneity.

Therefore, we have now revised the previous Extended Data Fig 3j that contained CYP2E1 only, for a revised Figure with the dual ECAD/ CYP2E1 staining (revised Extended Data Fig 3j). However, since ECAD is not a definitive periportal marker in human tissue (contrary to mouse) but it is just enriched in periportal zone, to avoid potential overinterpretation, we have refrained from claiming liver zonation in hepatocyte organoids, instead we have limited the results to the exact interpretation of the data being. Below we copy the revised Extended Data Fig.3j and the accompanying text edits for the Reviewer's convenience.

Line 215-218: Dual immunofluorescence staining for CYP2E1 (pericentral marker) and ECAD (enriched in periportal region) highlighted the heterogeneity and spatial distribution of hepatocyte function within the same h-HepOrg, at least for those genes tested (Extended Data Fig.3j).

Figure R2.1_Revised Extended Data Fig. 3j: Dual ECAD/CYP2E1 staining in h-HepOrgs in differentiation medium (h-HepOrg-DM). Representative immunofluorescent images from n=3 independent experiments of zonation markers in h-HepOrg-DM - pericentral zone marker, CYP2E1 (magenta) and periportal zone marker, ECAD (green) in the same organoid. Both markers depicted in Fire LUT for enhanced visualization. Scale bar, 50 μm.

Referee #3 (Remarks to the Author):

While the manuscript has improved by addressing the comments, I have some reservations about the novelty and groundbreaking aspects of the work and some parts are not well developed yet.

1) there are similar studies in the literature, including the examples below that used a version to achieve similar goal and would be hard to substantiate the differences or superiority. Unfortunately this leaves me with dampened enthusiasm on innovation and significance.

Concerning Hep. expansion:

1. Hui et al., Cell Stem Cell (2018): “In Vitro Expansion of Primary Human Hepatocytes with Efficient Liver Repopulation” — Developed a defined medium to expand primary human hepatocytes to large quantities and proliferating hepatocytes exhibited both hepatocyte and progenitor features, could revert to mature phenotypes in organoid culture, and notably repopulated mouse livers with high efficiency.....

A: Here we believe the Reviewer meant the Zhang et al., 2018 Cell Stem Cell paper from the Hui lab “In Vitro Expansion of Primary Human Hepatocytes with Efficient Liver Repopulation Capacity PMID: 30416071”

We would like to emphasize that the manuscript cited by the Reviewer from the Hui lab (Zhang et al. 2018) was cited in our previous version of the manuscript (ref 18, lines 62-63 of the previous version, now ref18 lines 58-59) and also included in the Supplementary Table 3, where we made a comparative assessment of existing hepatocyte culture systems. Briefly, while Hui’s lab ProlHH method reported expansion of 2D-hepatocytes with high repopulation capacity, these cells predominantly exhibited progenitor-like features during expansion and required further 3D culture conditions to reacquire differentiated hepatocyte phenotypes. In contrast, in our model we expand human hepatocytes as Hepatocyte Organoids (3D) that retain mature hepatocyte features, namely hepatocyte expression, bile canaliculi formation and function and further use them to generate assembloids. Additionally, we directly measured CYP-mediated drug metabolism using matched-donor PHHs, showing that h-HepOrg can equal or exceed PHHs in functional metabolism of Verapamil via CYP3A4/3A5/2C8 (Fig. 3g). This level of donor-matched functional comparison using LC-MS/MS is not included in Zhang et al. 2018 (Hui’s lab paper) or the referenced preprints below.

Additionally, our h-HepOrg system supports expansion while preserving both mature hepatocyte identity and inter-individual functional heterogeneity—capturing patient specific features including the expression of disease-relevant genes (e.g., IL1RL1, ERAP2; hepatitis or GPC3, liver cancer, among

others) thereby enabling robust applications in personalized medicine (Fig. 3d, Supplementary Table 4). To our knowledge, this is the first adult hepatocyte culture system reported to capture this degree of patient-specific variability.

Combined our results highlight **a key distinction between our platform and Hui's lab system**: while the Hui lab in Zhang et al. 2018 focused on reprogramming adult hepatocytes to a "progenitor-like hepatocytes state", our model maintains a more mature hepatic identity both functionally and transcriptionally.

2. <https://www.biorxiv.org/content/10.1101/2024.06.10.598262v1.full.pdf>

"Mass Generation and Long-term Expansion of Hepatobiliary Organoids from Adult Primary Human Hepatocytes" by Spee, Bart et al.

3. www.biorxiv.org/content/10.1101/2024.12.28.630269v1.full.pdf

"Expandable, Functional Hepatocytes Derived from Primary Cells Enable Liver Therapeutics"

A: We thank the Reviewer for pointing us to the recent preprints:

(2) from the Spee lab (Marsee, Ritchie et al. 2024), "Mass Generation and Long-term Expansion of Hepatobiliary Organoids from Adult Primary Human Hepatocytes"

and

(3) the Chhabra lab (Mallanna, Karanth et al. 2024), "Expandable, Functional Hepatocytes Derived from Primary Cells Enable Liver Therapeutics".

We fully acknowledge the importance of situating our work within the context of ongoing efforts in the field, and the need to clearly distinguish our h-HepOrg system from these models. However, since the medium composition for Mallana et al. is not disclosed and these unpublished studies are not yet peer-reviewed, we believe it would be premature and unfair to both our work and to those authors to attempt a direct comparison at this time. Instead, to appropriately recognize these contributions we have cited both preprints in the revised discussion of the manuscript, to underscore the ongoing challenges in expanding adult human hepatocytes and the importance of this research area.

To reiterate the novelty and impact of our h-HepOrg system, we highlight the following key features and advances of our model:

1) Robust Expansion and Biobanking from Adult Human Tissue

Our system supports the serial and exponential expansion of adult primary human hepatocytes over more than 3 months (1:2 split /week = doubling population time ~7 days), while preserving hepatocyte

identity (Fig. 1 and Extended Data Fig. 1). This has enabled the establishment of a living biobank from 28 distinct adult donors/patients—a scale and robustness not previously demonstrated.

2) Preservation of Donor-Specific Gene Signatures

We show that h-HepOrgs maintain donor-specific transcriptional profiles over long-term culture (up to passage 7), including expression of disease-relevant and polymorphic genes such as GPC3, IL1RL1, APOA4, and FAR2 (Fig. 3d, Supplementary Table 4). This fidelity is crucial for applications in precision medicine, pharmacogenomics, and disease modeling, and distinguishes our model from progenitor-focused or dedifferentiated systems.

3) In Vivo-like Tissue Architecture and Zonation Heterogeneity

Differentiated h-HepOrgs exhibit spatial heterogeneity in gene expression, recapitulating some features of hepatic zonation with coexisting periportal and pericentral marker expression (Fig. 2d, Extended Data Fig. 3h). Moreover, bile canaliculi in h-HepOrgs formed under differentiation medium display thickness and 3D network topology closely resembling those in native tissue (Fig. 3b-c, Extended Data Fig. 4a-d), further emphasizing their architectural and functional maturity.

4) Functional Competence and Regenerative Capacity

h-HepOrgs express high levels of mature hepatocyte markers and secrete albumin at levels comparable to matching 2D- hepatocytes (both 1day and 7day primary hepatocytes)—the current gold standard. They also display comparable cytochrome P450 enzyme activity as 7day-PHH (but not 1day-PHH), outperform matching 2D-PHH cultures in drug metabolism capacity, and can repopulate the liver in a mouse model of hepatic injury (Fig. 3e-h, Extended Data Fig. 4f).

We have now revised the discussion of the manuscript (lines 439-447, copied below for the Reviewer's convenience) to further emphasize these points and, as said above, cited both preprints to highlight the longstanding challenge of expanding adult human hepatocytes and the broader interest in developing clinically and scientifically relevant culture systems. We hope this clearly demonstrates how **our system represents a conceptual and technical advancement in the field of human hepatocyte culture and organoid models.**

We are grateful for the Reviewer's comment, which has helped us improve the clarity and context of our manuscript.

Lines 439-447: Recent advances in human liver models underscore the ongoing efforts and the broad interest in developing physiologically relevant in vitro systems. These include: iPSC-derived hepatocyte organoids exhibiting dual zonation⁶⁸, functional hepatocyte organoids derived from cryopreserved hepatocytes⁶⁹, mass-generation of hepatobiliary organoids⁷⁰, co-

cultures of dermal fibroblasts with hepatocyte spheroids⁷¹ or mouse fibroblasts aggregated with hepatocyte spheroids and cholangiocyte organoids⁷². However, a model capable of recapitulating the multicellular periportal liver tissue organization and cellular interactions *ex vivo*—while enabling inter-individual comparative studies and investigation of patient-specific disease traits—has not yet been developed.

Concerning assembloids :

1. Nature (2025): “Mouse liver assembloids model periportal architecture and biliary fibrosis” (DOI: 10.1038/s41586-025-09183-9) — The authors of the current manuscript highlight some differences in Table 3, but in my view, these are limited, and the fundamental innovation of the model remains modest. While species-specific differences are important and justify developing human models, the claimed advances here appear incremental as the principle used to generate the self-organizing structure is similar via mixing the cell types in defined ratios.

*A: We thank the Reviewer for referencing our mouse study (Dowbaj, Sljukic et al. 2025) as an important point of comparison. However, we respectfully submit that our human assembloid study constitutes **a conceptually and functionally distinct advance** with significant translational and biological implications that extend well beyond the scope of the previous mouse model. Please allow us to delineate the fundamental differences:*

*While the mouse study allowed us to uncover mechanistic insights into liver architecture and fibrosis in a genetically consistent model, the current **human** system enables **long-term co-culture and expansion** of primary human liver lineages, captures **donor-specific hepatic** features, and recapitulates functional **human drug metabolic** liver activity *in vitro* — all of which were not possible in the prior mouse work. Key distinctions between the two systems include:*

1. Fundamental differences in application scope:

The mouse study primarily focused on: (i) Structural reconstruction, such as the bile canaliculi–bile duct connection and (ii) Mechanistic insights into fibrogenesis and epithelial-mesenchymal interactions.

Rodent models are ideal for identifying and dissecting universal mechanisms due to their little genetic variability, and ease of manipulation. However, similar studies in human tissue are challenging owing to the high genetic variability of human samples that require having large sample sizes.

Human models instead are essential for translational relevance, especially in the context of: Drug metabolism and hepatotoxicity; Human-specific gene expression and regulatory pathways; Patient-specific disease modelling and Inter-individual variability in hepatic function.

Our human assembloid **model addresses these needs** by faithfully reconstructing patient-specific, multicellular liver tissue that exhibits hepatocyte and metabolic function, features that are not captured in the mouse system. It uniquely **supports applications in drug metabolism, toxicity testing, human-specific pathophysiology, and personalized diagnostics.**

2. Species-Specific Applications Not Possible in Mouse Models

Key features of our human model that go beyond the mouse system include:

- Expression and regulation of human-specific drug-metabolizing enzymes (e.g., CYP3A4)
- More accurate toxicology profiling relevant to human clinical outcomes
- Engraftment and functional repopulation *in vivo* using human hepatocytes

These applications are inherently species-dependent and critical for the translation of liver biology into clinical research, drug testing, and personalized medicine.

3. Patient-Specific Modeling and New Experimental Possibilities

Unlike the mouse model, our human **assembloids enable Patient-specific features and can be leveraged** for:

- (i) Modeling of rare genetic and metabolic liver disorders on a patient-specific basis
- (ii) Pharmacogenomic analysis, allowing donor-matched evaluation of drug metabolism and response
- (iii) Preservation of donor-specific gene signatures, including polymorphic or disease-linked genes such as APOA4, GPC3, and IL1RL1 (see Fig. 3d, Supplementary Table 4)

In addition, **our system is modular and expandable.** By isolating and biobanking the three major hepatic cell types (hepatocytes, portal fibroblasts, and cholangiocytes) from different donors, we can mix and match combinations — akin to a “Lego-like” assembly — enabling comparative studies of inter-individual differences in cellular behavior or drug response. This modularity and patient specificity are not achievable in murine or human- iPSC-derived systems.

Finally, we acknowledge that a recent preprint from the 14th July from the Bhatia lab (Westerfield, Grzelak et al. 2025) “A 3D *in vitro* model of the human hepatobiliary junction” has presented a model where human hepatocytes are mixed with human cholangiocyte organoids and fibroblasts. However, that system uses mouse fibroblasts, and their hepatocytes are cultured as non-expanding spheroids. In contrast, our system supports long-term expansion, cryopreservation, and modular reconstruction from fully human, donor-matched components.

*In summary, while the underlying assembly principles may overlap conceptually, the goals, capabilities, and translational applications of **the human assembloid model are fundamentally different and advance the field in significant new directions.***

We have discussed these differences with our Dowbaj, Slijkic et al., recent paper as well as cited the Bathia's lab recent preprint to further clarify the unique contribution of our work (lines 428-437 and 439-447).

We thank the Reviewer for this comment, which has helped us better articulate the novelty and impact of our study.

*lines 428-437: **While reductionist by nature, ex-vivo systems, offer powerful tools to dissect disease mechanisms, particularly, the relative contributions of the distinct intrinsic cellular programs and microenvironmental cues—including cell-cell interactions—to disease initiation and progression. We recently showed that mouse periportal assembloids retain key architectural features, such as the reconstruction of the bile canaliculi-bile duct connection and can serve as a tractable and modular in vitro model to investigate universal principles of bile canaliculi formation, bile flow, cholestatic injury and biliary fibrogenesis⁶⁴. However, species-specific differences in drug metabolism, toxicity profiles or liver pathophysiology, necessitate the development of complementary human models that capture patient-specific features to better understand diseases mechanisms, identify therapeutic strategies, or screen for therapeutic compounds.***

And

*Lines 439-447: **Recent advances in human liver models underscore the ongoing efforts and the broad interest in developing physiologically relevant in vitro systems. These include: iPSC-derived hepatocyte organoids exhibiting dual zonation⁶⁸, functional hepatocyte organoids derived from cryopreserved hepatocytes⁶⁹, mass-generation of hepatobiliary organoids⁷⁰, co-cultures of dermal fibroblasts with hepatocyte spheroids⁷¹ or mouse fibroblasts aggregated with hepatocyte spheroids and cholangiocyte organoids⁷². However, a model capable of recapitulating the multicellular periportal liver tissue organization and cellular interactions ex vivo—while enabling inter-individual comparative studies and investigation of patient-specific disease traits—has not yet been developed.***

2) The term “periportal triad” refers to a group of three key structures located at the periphery of hepatic lobules in the liver: the portal vein, hepatic artery, and bile duct with oxygenated blood. In the current work, the development of self-organizing periportal-like structures notably lacks endothelial vasculature and limited control on oxygen. These limitations need to be acknowledged in the Discussion.

A: We fully agree with the Reviewer that the canonical “periportal triad” comprises three essential anatomical structures: the portal vein, hepatic artery, and bile duct — all of which are embedded within a vascularized microenvironment enriched in oxygenated blood. In our previous version we had already acknowledged and discussed the absence of other stromal elements, including endothelial cells, in our model:

lines 570-573 of previous version of the manuscript: One caveat of our model, though, is that it still lacks the other stromal components, mainly the other mesenchymal cells (HSC and VSCM) as well as the endothelium and immune cells. Incorporating these to generate more complex models will be crucial to reproduce all aspects of liver disease.

However, the Reviewer is right in pointing out that we had not specifically mentioned that one additional limitation of our model is the lack of control over oxygen gradient or modulation. Therefore, we have now revised the Discussion section to explicitly acknowledge this point by adding the following paragraph:

lines 477–481 in the revised manuscript: One caveat of the model, though, is that it still lacks the other stromal components, mainly the other mesenchymal cells (HSC and VSCM) as well as the endothelium and immune cells. The lack of portal vasculature (portal vein and hepatic artery) limits the formation of a true periportal triad, as endothelial networks are essential for oxygen delivery and spatial patterning.

We hope this addition addresses the Reviewer’s concern.

3) A more critical issue is that the frequency and spatial organization of the (chol-PF-Hep) structures are not well characterized. The authors mention that “mesenchymal cells are closely associated with cholangiocytes,” but are they really? From the images provided, the positioning of mesenchymal and Chol cells appears limited, and in some cases they happen to fall next to each other. Can the authors quantify how many bona fide periportal structures are formed and what proportion of them show the correct configuration? The current data shows a weak association.

A: We thank the Reviewer for this point regarding the spatial organization and frequency of periportal-like structures within our assembloids, which made us realize that we had not explicitly defined what we refer to as an “assembloid” in the manuscript.

In our study, we use the term “assembloids” strictly to describe structures in which all three cell types—cholangiocytes, portal fibroblasts (PFs), and hepatocytes—are present, with cholangiocytes and PFs embedded within the hepatocyte organoid (HepOrg) structure. This distinction is crucial, as we only quantify and analyze those composite structures.

As shown in our previously submitted Figure 5c (now updated as Figure 4c), ~ 80% of the total structures we generate meet this criterion (80% assembloids efficiency). We have now quantified how many of these form “correct” organizations. We find that **~80% show correct** periportal-like spatial configuration, which we define as: structures where the **ductal cells (marked by nuclear-GFP) form an apical lumen and are basally contacted by mesenchymal cells (nuclear RFP or nuclear RFP + Vimentin stained) and surrounded by hepatocytes**. These results substantiate that our assembloids reliably recreate key organizational features of the native periportal region.

Now, in the revised version of the manuscript, we have updated the Results section to reflect this more precise characterization and to ensure that the spatial arrangement and frequency of bona fide periportal-like structures are transparently presented. See **revised Extended Data Fig 7e (Figure R3.1b)** and revised lines in 302-323, also pasted below for the Reviewer convenience. Additionally, for transparency, below we provide two confocal examples (for **Reviewer only**) of what we quantified as incorrect periportal configuration (Figure R3.1a, top panel) and what we quantified as correct configuration (Figure R3.1a, bottom panel), next to the quantification results (Figure R3.1b). The quantification results have been included in the **revised Extended Data Fig. 7e** and are also provided in full in the Revised Source Data file.

We appreciate the reviewer’s suggestion, which has helped us improving the presentation of our findings.

Figure R3.1: Quantification of correctly organized periportal-like spatial configurations in assembloids. **a**, Representative immunofluorescence images for the Reviewer only of hepatic assembloids displaying either incorrect (top) or correct (bottom) periportal-like configurations, to illustrate the correct spatial configuration defined as: mesenchymal cells (nRFP) in close proximity to organized, lumen-forming, cholangiocyte ducts (nGFP). Nuclei are counterstained with DAPI (purple).

Asterisks indicate lumen structures. Scale bars, 50 μ m. **b**, Donut chart illustrating the proportion of correctly (blue) and incorrectly (grey) organized periportal-like structures amongst assembloids ($n = 20$). Approximately 80% of structures exhibited this correct arrangement.

Lines 302-323: To induce the self-assembly of the three cell populations into a single structure that would recapitulate periportal spatial organization, we tested several approaches to aggregate h-HepOrgs with dissociated portal liver mesenchyme and cholangiocytes from h-CholOrgs from the same donor, when possible. Of all the approaches tested, mixing one h-HepOrg structure with a defined number of portal mesenchymal and ductal cells (from h-CholOrgs) in 96-well low adhesion U-plates readily generated structures where the 3 cell types were together with Cholangiocyte and MSC cells embedded inside the HepOrg structure. We called these structures periportal assembloids (Fig. 4a, b). The ratio 1 HepOrg: 25 MSC and 100 Chol better captured the tissue cell ratios at day 6 post-assembloid formation, and was selected for further experiments (Extended Data Fig. 7b, c). To increase the number of assembloids generated, we used Aggrewell™ plates (Fig. 4b and Extended Data Fig. 7d). Notably, both methods generated assembloids with high efficiency (~80% efficiency) (Fig. 4c) and closely maintained the cellular composition and proportions of the tissue (Fig. 4e). Therefore, we only used Aggrewell from hereon. Remarkably, from day 3 onwards, the periportal assembloids recapitulated key architectural features of the *in vivo* tissue, with ductal cells (KRT19⁺ and nGFP⁺) forming bile duct-like structures containing open lumina, with portal mesenchymal cells (nuclear-RFP) in close proximity, and embedded within the hepatocyte (HNF4A⁺) parenchyma. This architectural organization, where ductal cells are forming an apical lumen, basally contacted by mesenchymal cells and embedded in the hepatocyte structure, was observed in the majority (80%) of the assembloids and across donors. These results were independent of the patient source for the healthy portal liver mesenchyme, indicating minimal impact of patient-origin under healthy conditions (Fig. 4d-f and Extended Data Fig. 7d-h).

Perhaps given that single-cell data are available, the authors could also use their cell-cell communication to determine whether there is any biological basis or signaling logic suggesting that mesenchymal cells are recruited toward biliary lumens to improve the model rationale.

A: We thank the reviewer for this suggestion to explore the biological basis of mesenchymal cell recruitment in our model. We agree that understanding the cellular interactions between

cholangiocytes, mesenchyme, and hepatocytes could explain the biological basis not only as to how mesenchymal cells are recruited towards biliary structures but could elucidate the mechanisms of portal tract formation in humans.

Our central hypothesis is that the use of primary cells—cholangiocytes, mesenchymal cells (portal fibroblasts), and hepatocytes—directly derived from human liver tissue allows these cells to retain tissue-specific identity, even after in vitro expansion. This is supported by our observation that cells from different donors exhibit transcriptional specificity despite being cultured under identical conditions (Fig. 3d and Extended Data Fig. 4e and Supplementary Table 4), suggesting retention of tissue-of-origin identity.

Moreover, tissue-specific characteristics of fibroblasts, like those of epithelial and endothelial cells, are increasingly recognized in the literature (see recent Liu et al., 2025). These features influence the expression of ligand–receptor pairs essential for establishing tissue-specific cellular interactions. In line with this, our previous published work (see Figure S4L from Cordero-Espinoza, Dowbaj et al. 2021 Cell Stem Cell) demonstrated that co-culture of mouse cholangiocytes with mouse liver portal mesenchyme supported organoid formation, whereas co-culture with non-liver-derived fibroblasts (e.g., MEFs) did not—further supporting the importance of tissue-specific mesenchymal identity in proper structural assembly.

To explore the reviewer’s question more directly, we have now performed in silico cell–cell communication analysis using our scRNA-seq data from the assembloids. This analysis revealed a total of 4313 significant ($p < 0.01$) ligand–receptor interactions among the three cell types, and 280 significant ($p < 0.01$) ligand–receptor interactions with cholangiocytes as the sender cells and mesenchymal cells as the receiver cells. Focusing on signaling pathways known to mediate mesenchymal growth and recruitment—such as PDGF, TGF- β , and WNT—we identified several ligand–receptor pairs in which cholangiocytes or hepatocytes act as ligand-producing (sender) cells and portal fibroblasts/mesenchyme as the receptor (receivers) cells. Notably:

(i) Cholangiocytes displayed higher expression levels, compared to hepatocytes, of mesenchyme-recruiting ligands such as PDGFB, TGFB1, and several WNTs (WNT7A, 7B, 10A and 11) with corresponding receptors enriched in the mesenchymal population.

(ii) Certain interactions, particularly those involving WNT signaling, were found to be exclusive to cholangiocyte–mesenchyme communication, and although the average expression levels were modest, their specificity scores were high—suggesting targeted interactions rather than non-specific signaling.

(iii) Amongst the cholangiocyte-mesenchyme interactions, we found *IL6ST* signaling and *EFNA1-EPHA3*, both reported in recent spatial transcriptomics datasets from human liver tissue (Andrews et al. (2024) and Brazovskaja et al. (2024)) to mediate Cholangiocyte-mesenchyme interactions, thus supporting the validity of our model to reproduce meaningful *in vivo* interactions.

While these *in silico* results suggest a biological rationale whereby cholangiocytes may secrete ligands that recruit or spatially organize mesenchymal cells, we acknowledge that these findings remain predictive and would require experimental validation to draw definitive conclusions. Moreover, hepatocytes and mesenchymal cells themselves likely contribute to the overall spatial organization, making this a complex and multifactorial process.

Given the scope and focus of the current study, we have chosen not to include this preliminary analysis in the manuscript to avoid overextending its claims but we are just providing them here for the **Reviewer only**. We would gladly include them in the supplementary materials should the reviewer or editor deem this important for supporting the concept of the manuscript.

Figure R3.2 for Reviewer only: Ligand to receptor interactions among the 3 cell-types depicted as a dotplot. Source or senders are labelled at the top and Target or receivers are labelled at the bottom. Size of the dot represents the specificity of the interactions and the colour represents the mean expression levels of the ligand-receptor pair. p -value <0.01 for the depicted ligand-receptor pairs.

Furthermore, what percentage of the Chol cells actually form biliary lumens and what percent are left in parenchymal scattered? Addressing these points is very useful to evaluate whether the model partly includes the hallmark periportal-like and adds to the rigor of the model for people to depend and use.

A: We believe this point refers once more to the structural organization of the assembloids. To address this, we quantitatively assessed the structural organization of cholangiocytes within assembloids. We categorized them as either forming organized biliary lumens or remaining scattered/unstructured.

From the total initial number of cholangiocytes in an assembloids, a significant majority (~70%) of initial cholangiocytes exhibited organized architecture, forming distinct biliary-like lumens.

These results indicate that cholangiocyte organization within our assembloids predominantly recapitulates hallmark features of periportal identity, reinforcing the structural and functional relevance of the model for studying portal tract biology.

*We have explained these results in the revised **Periportal assembloids Methods** section of the manuscript, and Source data file which is associated with the revised Extended Data Fig. 7e. Below, we provide a graph presenting the summary of these results for the **Reviewer's only**.*

Figure R3.3 for Reviewer only: Structural organization of cholangiocytes within periportal-like regions. Quantification of cholangiocyte morphology in 20 assembloids. Cells were classified as either forming biliary lumens (dark grey) or scattered/unstructured (light grey). A significantly greater proportion of cholangiocytes formed organized lumens (mean ~70%) compared to those that remained scattered (~30%) (unpaired two-tailed t-test, $p < 0.0001$). Each dot represents an individual assembloid.

Lines 1136-1139: . **Under these conditions, 70% of the initial Cholangiocytes formed a lumen.** Raw data were incorporated into the quantification of periportal-like spatial organization in assembloids (Source data file from Extended Data Fig. 7e).

4) Authors mention “Staining for the periportal hepatocyte markers SAA1 and SAA2 confirmed the spatially confined expression of these portal markers in the hepatocytes located within cholangiocyte rich, ECAD⁺ regions, with portal Mesenchyme (PFs-nRFP) in close proximity, further supporting the establishment of a periportal-like domain in the assembloids (Extended Data Fig. 9a)” .

>>> However in Fig 9a control staining of hepatocyte organoids are missing without which it is hard to recognize the true signal. Also the tissue patterning as they state is weak.

A: We respectfully disagree with the reviewer's assessment that the antibody does not recognize the true signal. We have carefully evaluated the staining and observed both positive and negative cells within the same structure. This result is consistent with the scRNAseq data presented in the previous version of the manuscript (previous Figure 5h, now Figure 4h, copied below as Figure_R3.4c for the Reviewer convenience), showing that some hepatocytes within assembloids express both SAA1 and SAA2 at the RNA level, corroborating the immunofluorescence and suggesting real biological heterogeneity. Furthermore, the staining co-localizes with the high expression of ECAD in the structure, which has been previously reported to be enriched in periportal regions of the liver. Thus, we don't have reasons to believe that the staining is not “true signal”.

*Regarding the request for a negative control staining in hepatocyte organoids alone, we appreciate the suggestion, but **this would not address the reviewer's concern, as also HepOrg express SAA1 and SAA2.** As shown in Extended Data Fig. 3h of the previous version, hepatocyte organoids under differentiation conditions express both periportal and pericentral markers, including SAA1 and SAA2. We hypothesize that this might occur due to gradients of Wnt signaling and probably also oxygen within the organoids, which could induce periportal marker expression and lead to some cells acquiring periportal-like gene expression. However, we believe that testing this hypothesis would fall outside of the scope of the current manuscript and would shift the focus from the assembloids we presented.*

To address the reviewer's concern, we have now revised this figure to include higher-magnification images and the FIRE-LUT channel, to better visualize differences in staining intensities. As shown in revised Extended Data Fig. 9a (also copied below as Figure R3.4a for the Reviewer's convenience), the individual channels and the magnifications on inserts 1 and 2 indicate the presence of both positive and negative hepatocytes in different cells within a the same assembloid. This is in agreement with our scRNASeq data showing expression of SAA1/SAA2 in a subset of cells (Figure 4h of the manuscript), which we also paste below for the reviewer (Figure 3.4c).

Figure R3.4: Validation of SAA1/SAA2 periportal marker expression in hepatocytes within assembloids by scRNA-seq and immunofluorescence. **a-b**, Immunofluorescence staining of day 4 periportal assembloids for SAA1/SAA2, ECAD. **a**, FIRE-LUT panels illustrate the different intensity of the corresponding markers. **b**, SAA1 and SAA2 (yellow), epithelial marker ECAD (white), cholangiocytes (Chol-nGFP, green), and portal fibroblasts (PFs-nRFP, magenta) panels are shown. Robust SAA1/SAA2 staining is observed in hepatocytes adjacent to ECAD⁺. Hepatocytes with positive (white arrow, region 1) and negative (magenta arrow, region 2) SAA1/SAA2 staining are shown. Scale bar, 50 μ m. **c**, Dot plot showing the expression of periportal markers SAA1 and SAA2 in assembloids, based on scRNA-seq analysis.

Also the tissue patterning as they state is weak.

We thank the reviewer for this point. We agree that while we observe spatially restricted expression of periportal markers, we do not demonstrate that the expression of periportal markers is due to the formation of a periportal domain. An alternative explanation could be that the periportal domain is located inside the structure, rather than on the surface, where factors like reduced Wnt exposure or

oxygen gradients may influence gene expression. In light of this, we have revised the manuscript to **remove the sentence referring to periportal domain establishment**, and we now refrain from making claims about zoned architecture. See revised lines 351-354, which we also provide below for the reviewer's convenience:

Lines 351-354: Staining for the periportal hepatocyte markers SAA1 and SAA2 confirmed the **spatially heterogenous** expression of these portal markers, **with the positive cells overlapping with regions of ECAD⁺ high cells** (Extended Data Fig. 9a), and in agreement with our **scRNAseq results (Fig. 4 h)**.

We believe this explanation clarifies this Reviewer's concern.

5) It appears that when authors use 1D-Hep as the control, the medium lacks ITS (Insulin, transferrin, selenium widely used) and Dexamethasone. This importantly can compromise the quality of the control group and can affect the reported differences. A high-quality control is essential to demonstrate the CYP3A4 (or other cyps) differences convincingly.

A: We agree with the reviewer that the quality of the control group is essential and we appreciate the reviewer's attention to this detail.

*To clarify, **when culturing primary human hepatocytes (PHHs) we supplement the medium with Dexamethasone, for 7-day 2D cultures we utilize CM4000 supplement, which contains both ITS and Dexamethasone.** During the 6h of the CYP assay the medium does not contain ITS or Dexamethasone to avoid potential interference during the measurement. However, we ensure comparability of the assay across different samples by (i) maintaining the same assay medium composition across all samples and conditions (1D-PHH, HepOrg) and (ii) normalizing the assay results for viable cell numbers. We acknowledge that this was not clearly stated in the original manuscript and apologize for the mistake.*

*We have **now corrected and clarified the Methods** section in the revised version of the manuscript (**functional assays section**). Now, we accurately reflect the medium used for culturing and CYP assay and additionally clarify that the medium used is the same across all the compared groups (1D-PHH, 7D-PHH and HepOrg).*

Regarding ITS, for the 7-day PHHs control, ITS and Dexamethasone are included in the culture conditions until the assay. For the 1-day PHH control, the assay starts 18h after isolation and seeding, and in this period, we add Dexamethasone but not the CM4000 (ITS supplement). While we acknowledge that ITS is not present during these 18h, we believe this does not impact the conclusions of our results. In the previous version of the manuscript, we had been transparent and already mentioned (see lines 309-310 from previous version, new lines 253-254 copied below for the Reviewer's convenience) that 1day PHHs were superior in CYP activity compared to our h-HepOrg-DM, and

provided statistical significance (Figure 3e). Furthermore, as mentioned above, the CYP assays are all performed in the same medium for all the conditions and the results are normalized for viable cell numbers, accounting for potential differences affecting viability in the different conditions.

Therefore, even if the addition of ITS during the starting 18-hour of culture were to further enhance CYP activity in 1-day PHHs, it would only strengthen this already acknowledged superiority — not alter the interpretation or conclusions of our data. **Given that this control is already outperforming the experimental conditions, and our main findings would remain unchanged even with an increased superiority of 1-day PHHs, we do not believe it is necessary to repeat the experiment.**

Lines 309-310 from previous version, new lines 253-254 (in yellow highlighted the sentence where we refer to the superiority of the 1d-PHHs on CYP assays: Next, we compared the functional performance of differentiated h-HepOrgs to primary human hepatocytes (PHHs). HepOrg differentiated in DM exhibited mature hepatic functions, including robust albumin secretion and, moderate cytochrome P450 activity, comparable to 7-day PHHs (Fig. 3e–f). Specifically, differentiated h-HepOrgs displayed CYP2C9 activity equivalent to that of 7-day PHHs, and modestly reduced CYP3A4 activity, while 1-day PHHs demonstrated superior activity for both enzymes.

Additionally, there seems to be a discrepancy between the drug testing data and the CYP luc assay as one suggests the system performs on par with the control, while the other does not. I could not find an explanation for this inconsistency. Improving and clarifying this part of the data would make the findings more reliable.

A: We appreciate the reviewer's observation regarding the apparent discrepancy between the CYP luciferase reporter assays (Figure 3e) and the Norverapamil metabolism data (Figure 3g), particularly concerning the comparison between 1d-PHH and h-HepOrg-DM.

We respectfully clarify that this is not a direct contradiction but rather a reflection of the different scope and resolution of the two assays:

1. CYP luciferase assays (Figure 3e) measure the activity of individual cytochrome P450 isoforms, specifically CYP2C9 and CYP3A4, using luciferase-based reporter systems. These assays provide a focused readout of the activity of a single enzyme, which is valuable for assessing isoform-specific induction or suppression.
2. In contrast, the Norverapamil drug metabolism assay (Figure 3g) measures the net metabolic capacity of the system toward verapamil, which is metabolized via multiple cytochrome P450 isoforms, including CYP2C8, CYP3A4, and CYP3A5 (as previously reported in: Tracy, Korzekwa et al. (1999). Therefore, this assay reflects the combined activity of several CYPs rather than any single enzyme.

Consequently, while 1d-PHH may show higher CYP2C9 or CYP3A4 luciferase activity individually, the h-HepOrg-DM system demonstrates superior overall metabolic performance toward Norverapamil, likely due to more robust or sustained expression and coordination among multiple CYP enzymes relevant to verapamil metabolism.

We have now included an explicit explanation of this distinction in the revised Results section (lines 257-259) and added a clarifying sentence in the figure legend to guide readers. We hope this clarification addresses the reviewer's concern and strengthens the reliability of our findings.

Lines 257-259: **This superior overall metabolic performance toward Norverapamil suggests a more robust or sustained expression and coordination among multiple CYP enzymes relevant to Verapamil metabolism including the** metabolizing enzymes *CYP2C8*, *CYP3A4*, and *CYP3A5*, all responsible for Verapamil N-demethylation and highly expressed in h-HepOrgs in differentiation medium (Extended Data Fig. 3g, i).

--

1) Minor comment:

The authors state :” For our goal to reconstruct the physiological cell-cell interactions of the periportal region of the liver lobule it is a prerequisite that the hepatocytes are mature enough to interact and establish connections with the neighbouring ductal epithelium”.

>> this is an assumption that is not backed up by data or literature.

A: We thank the reviewer for this comment and apologize for not appropriately referencing this point in the initial submission. It is well established that mature hepatocytes form an interconnected bile canaliculi (BC) network with long, thin luminal structures essential for bile transport(Gissen and Arias 2015). These BC structures connect apically with bile ducts in the structure known as the Canal of Hering. This anatomical and functional connection ensures directional bile flow and requires preserved polarity in both hepatocytes and cholangiocytes (e.g. Gissen and Arias (2015), Asensio, Ortiz-Rivero et al. (2022)). Given that immature hepatocytes do not generate a fully developed BC network, the maturation state is indeed critical for establishing such physiological interactions.

*However, in our manuscript, we do not directly investigate BC-BD connections or their dependence on hepatocyte maturity, and hence, this statement is not essential to support our conclusions. To avoid potential overinterpretation, **we have removed the sentence** from the revised manuscript.*

References:

Andrews, T. S., D. Nakib, C. T. Perciani, X. Z. Ma, L. Liu, E. Winter, D. Camat, S. W. Chung, P. Lumanto, J. Manuel, S. Mangroo, B. Hansen, B. Arpinder, C. Thoeni, B. Sayed, J. Feld, A. Gehring, A. Gulamhusein, G. M. Hirschfield, A. Ricciuto, G. D. Bader, I. D. McGilvray and S. MacParland (2024). "Single-cell, single-nucleus, and spatial transcriptomics characterization of the immunological landscape in the healthy and PSC human liver." *J Hepatol* **80**(5): 730-743.

Ardisasmita, A. I., I. F. Schene, I. P. Joore, G. Kok, D. Hendriks, B. Artegiani, M. Mokry, E. E. S. Nieuwenhuis and S. A. Fuchs (2022). "A comprehensive transcriptomic comparison of hepatocyte model systems improves selection of models for experimental use." *Commun Biol* **5**(1): 1094.

Asensio, M., S. Ortiz-Rivero, A. Morente-Carrasco and J. J. G. Marin (2022). "Etiopathogenesis and pathophysiology of Cholestasis." *Exploration of Digestive Diseases* **1**(2): 97-117.

Brazovskaja, A., T. Gomes, R. Holtackers, P. Wahle, C. Korner, Z. He, T. Schaffer, J. C. Eckel, R. Hansel, M. Santel, M. Seimiya, T. Denecke, M. Dannemann, M. Brosch, J. Hampe, D. Seehofer, G. Damm, J. G. Camp and B. Treutlein (2024). "Cell atlas of the regenerating human liver after portal vein embolization." *Nat Commun* **15**(1): 5827.

Cordero-Espinoza, L., A. M. Dowbaj, T. N. Kohler, B. Strauss, O. Sarlidou, G. Belenguer, C. Pacini, N. P. Martins, R. Dobie, J. R. Wilson-Kanamori, R. Butler, N. Prior, P. Serup, F. Jug, N. C. Henderson, F. Hollfelder and M. Huch (2021). "Dynamic cell contacts between periportal mesenchyme and ductal epithelium act as a rheostat for liver cell proliferation." *Cell Stem Cell* **28**(11): 1907-1921 e1908.

Dowbaj, A. M., A. Sljukic, A. Niksic, C. Landerer, J. Delpierre, H. Yang, A. Lahree, A. C. Kuhn, D. Beers, H. M. Byrne, S. Seifert, H. A. Harrington, M. Zerial and M. Huch (2025). "Mouse liver assembloids model periportal architecture and biliary fibrosis." *Nature*.

Gissen, P. and I. M. Arias (2015). "Structural and functional hepatocyte polarity and liver disease." *J Hepatol* **63**(4): 1023-1037.

Halpern, K. B., R. Shenhav, O. Matcovitch-Natan, B. Toth, D. Lemze, M. Golan, E. E. Massasa, S. Baydatch, S. Landen, A. E. Moor, A. Brandis, A. Giladi, A. S. Avihail, E. David, I. Amit and S. Itzkovitz (2017). "Single-cell spatial reconstruction reveals global division of labour in the mammalian liver." *Nature* **542**(7641): 352-356.

Hempel, M., A. Schmitz, S. Winkler, O. Kucukoglu, S. Bruckner, C. Niessen and B. Christ (2015). "Pathological implications of cadherin zonation in mouse liver." *Cell Mol Life Sci* **72**(13): 2599-2612.

Mallanna, S. K., S. S. Karanth, J. E. Marturano, A. K. Kudva, M. Lehmann, J. K. Morse, M. Jamiel, T. Norman, C. Wilson, F. Munarin, D. Broderick, M. V. Buskirk, E. Uddin, M. Ret, C. Steele, M. Cheema, J. Black, E. Vanderploeg, C. Chen, S. Bhatia, A. Rezaia, T. J. Lowery, S. Cazanave and

A. Chhabra (2024). "Expandable, Functional Hepatocytes Derived from Primary Cells Enable Liver Therapeutics." bioRxiv: 2024.2012.2028.630269.

Marsee, A., A. Ritchie, A. Myszczyzyn, S. Ye, J.-C. Chang, A. I. Ardismita, I. P. Joore, J. Castro-Alpizar, S. A. Fuchs, K. Schneeberger and B. Spee (2024). "Mass Generation and Long-term Expansion of Hepatobiliary Organoids from Adult Primary Human Hepatocytes." bioRxiv: 2024.2006.2010.598262.

Tracy, T. S., K. R. Korzekwa, F. J. Gonzalez and I. W. Wainer (1999). "Cytochrome P450 isoforms involved in metabolism of the enantiomers of verapamil and norverapamil." British Journal of Clinical Pharmacology **47**(5): 545-552.

Westerfield, A. D., K. A. Grzelak, K. Katsuyama, V. Kumar, B. M. Miller, J. Yun, J. Kirkpatrick, D. Mankus, M. E. Bisher, A. K. R. Lytton-Jean, Z. G. Jiang, D. D. Lee, C. S. Chen and S. N. Bhatia (2025). "A 3D *in vitro* model of the human hepatobiliary junction." bioRxiv: 2025.2007.2011.664464.

Yakubovsky, O., A. Afriat, A. Egozi, K. B. Halpern, T. Barkai, Y. Harnik, Y. K. Kohanim, R. Novoselsky, O. Golani, I. Goliand, Y. Addadi, M. Kedmi, H. Keren-Shaul, L. Fellus-Alyagor, D. Hirsch, C. Mayer, R. Pery, N. Pencovich, T. Taner, I. Nachmany and S. Itzkovitz (2025). "A spatial transcriptomics atlas of live donors reveals unique zonation patterns in the healthy human liver." bioRxiv: 2025.2002.2022.639181.

Zhang, K., L. Zhang, W. Liu, X. Ma, J. Cen, Z. Sun, C. Wang, S. Feng, Z. Zhang, L. Yue, L. Sun, Z. Zhu, X. Chen, A. Feng, J. Wu, Z. Jiang, P. Li, X. Cheng, D. Gao, L. Peng and L. Hui (2018). "In Vitro Expansion of Primary Human Hepatocytes with Efficient Liver Repopulation Capacity." Cell Stem Cell **23**(6): 806-819 e804.

Point-by-point Response to Editorial Requests and Reviewer Comments

We thank the editors and reviewers for their constructive feedback. We have carefully revised the manuscript and the figures according to the editorial requirements, and we are pleased that the reviewer is satisfied with our previous revisions.

We have addressed all points raised and revised the manuscript accordingly. A point-by-point response is provided below, with Editor or Reviewer comments in black and our responses in **blue** and *italics*.

Referee #3 (Remarks to the Author):

I congratulate the authors as they have been able to provide new data and discussion and satisfactory addressed my earlier concerns about the remaining parts of the manuscript.

A: We sincerely thank the reviewer for the positive and encouraging feedback. We are pleased that our additional data and discussion have satisfactorily addressed the previous concerns.